# Conditionally Whitened Generative Models for Probabilistic Time Series Forecasting

**Yanfeng Yang**[*]
Graduate University of Advanced Studies & The Institute of Statistical Mathematics
Tokyo, Japan
yanfengyang0316@gmail.com

**Siwei Chen, Pingping Hu, Zhaotong Shen, Yingjie Zhang, Zhuoran Sun, Shuai Li, Ziqi Chen**[*]
East China Normal University,
Shanghai, China
zqchen@fem.ecnu.edu.cn

**Kenji Fukumizu**[*]
The Institute of Statistical Mathematics
Tokyo, Japan
fukumizu@ism.ac.jp

## Abstract

Probabilistic forecasting of multivariate time series is challenging due to non-stationarity, inter-variable dependencies, and distribution shifts. While recent diffusion and flow matching models have shown promise, they often ignore informative priors such as conditional means and covariances. In this work, we propose Conditionally Whitened Generative Models (CW-Gen), a framework that incorporates prior information through conditional whitening. Theoretically, we establish sufficient conditions under which replacing the traditional terminal distribution of diffusion models, namely the standard multivariate normal, with a multivariate normal distribution parameterized by estimators of the conditional mean and covariance improves sample quality. Guided by this analysis, we design a novel Joint Mean-Covariance Estimator (JMCE) that simultaneously learns the conditional mean and sliding-window covariance. Building on JMCE, we introduce Conditionally Whitened Diffusion Models (CW-Diff) and extend them to Conditionally Whitened Flow Matching (CW-Flow). Experiments on five real-world datasets with six state-of-the-art generative models demonstrate that CW-Gen consistently enhances predictive performance, capturing non-stationary dynamics and inter-variable correlations more effectively than prior-free approaches. Empirical results further demonstrate that CW-Gen can effectively mitigate the effects of distribution shift.

## 1 Introduction

Time series analysis has a long history, with classical approaches such as ARIMA, state-space models, and vector autoregressions (VAR) (Box & Jenkins, 1976; Durbin & Koopman, 2012; Lütkepohl, 2007). Although these methods have been widely applied, they often struggle with high-dimensionality and complex data structures that arise in modern applications. More recently, neural architectures have demonstrated superior predictive accuracy, such as recurrent neural networks (RNN), Long Short-Term Memory (LSTM), and Transformers (Sherstinsky, 2020; Hochreiter & Schmidhuber, 1997; Vaswani et al., 2017). However, these neural models primarily focus on forecasting the conditional mean of future sequences given historical observations, while providing

---

All authors contributed equally. [*]Corresponding authors.

little to uncertainty quantification. These limitations have motivated the development of probabilistic forecasting, which seeks to model not only point predictions but also the associated uncertainty.

Multivariate time series probabilistic forecasting has recently emerged as a key methodology for quantifying predictive uncertainty, enabling informed decision-making in numerous real-world applications in diverse domains such as finance, healthcare, environmental science, and transportation (Lim & Zohren, 2021). Formally, the task involves learning the probability distribution $P_{\mathbf{X}|\mathbf{C}}$ of a future time series $\mathbf{X}_0 \in \mathbb{R}^{d \times T_f}$ of discrete time conditioned on its corresponding historical observations $\mathbf{C} \in \mathbb{R}^{d \times T_h}$, where the integers $T_f$ and $T_h$ denote the lengths of future and historical time series, respectively, and $d$ represents the dimensionality of each time step. However, this task still remains highly challenging, primarily due to (i) non-stationary characteristics, manifested through long-term trends, seasonal effects, and heteroscedasticity (Li et al., 2024; Ye et al., 2025); (ii) complex inter-variable dependency structures (Yuan & Qiao, 2024); (iii) inherent data uncertainty, such as short-term fluctuations (Ye et al., 2025); and (iv) potential distribution shifts between training and testing data (Kim et al., 2022).

In response to these challenges, recent advances in generative learning, especially diffusion models, focus on accurately estimating the conditional distribution $P_{\mathbf{X}|\mathbf{C}}$. TimeGrad employs a RNN to encode historical observations and generates forecasts autoregressively, but suffers from cumulative errors and slow computation (Rasul et al., 2021). CSDI uses a 2D-Transformer for imputation and forecasting (Tashiro et al., 2021), while SSSD employs a Structured State Space Model to reduce computational cost and emphasize temporal dependence (Alcaraz & Strodthoff, 2023). Nevertheless, CSDI, SSSD, and TimeGrad all struggle with long-term forecasting (Shen & Kwok, 2023). Diffusion-TS leverages a transformer to decompose time series into trend, seasonal, and residual components for generation, whereas FlowTS accelerates generation using rectified flow (Yuan & Qiao, 2024; Hu et al., 2025).

Although the aforementioned generative models have achieved promising performance, they ignore informative priors. Such priors, derived from historical observations or auxiliary models, can substantially improve conditional generative modeling. To the best of our knowledge, CARD is the first model to incorporate prior information into conditional diffusion models (Han et al., 2022). It pretrains a regressor to estimate the conditional mean $\mathbb{E}[\mathbf{X}_0|\mathbf{C}]$ and integrates this regressor into the diffusion process, thereby enhancing conditional generation. In time series forecasting, regressing the conditional mean and incorporating it into diffusion models as a prior has become a common practice, as it alleviates the difficulty of modeling non-stationary distributions. TimeDiff adopts a linear regressor to capture short-term patterns and employs a future mixup strategy during training to mitigate boundary disharmony (Shen & Kwok, 2023). However, its linear design limits the ability to capture complex trends and fluctuations. TMDM addresses this limitation by integrating a nonlinear regressor into the variational inference framework, enabling joint training of the regressor and the diffusion model (Li et al., 2024). The regressor for $\mathbb{E}[\mathbf{X}_0|\mathbf{C}]$ (hereafter referred to as the mean regressor) can capture trends, seasonality, and fluctuations but is vulnerable to heteroscedasticity. Building on this line, NsDiff addresses this by introducing two pretrained models: a mean regressor and a variance regressor, the latter estimating the conditional variance of each variable within a sliding window (Ye et al., 2025). By incorporating both regressors into the diffusion process, NsDiff models heteroscedasticity more effectively. Despite these innovations, the method still suffers from certain limitations, particularly the overly complex reverse process and the neglect of correlations among variables. A detailed discussion of these limitations is provided in Appendix A.1. Beyond diffusion models, S2DBM employs a diffusion bridge variant and incorporates the mean regressor in the same manner as CARD (Yang et al., 2024), which limits its ability to handle heteroscedasticity. TsFlow uses Gaussian Processes (GPs) as both the mean and variance regressors (Kollovieh et al., 2025), but its design is restricted to univariate forecasting with short horizons and inherits the typical drawbacks of GPs, including kernel sensitivity and cubic computational cost.

Building on the preceding literature, it is well established that carefully designed priors can substantially enhance generative models. Yet several key questions remain unresolved: How exactly do priors contribute to these improvements, and how accurate must the mean and variance regressors be to provide tangible benefits? How can such regressors be effectively trained, and are there theoretical guarantees supporting their impact? Most existing approaches incorporate mean and variance regressors into diffusion models by following the designs of CARD and DDPM (Han et al., 2022;

Ho et al., 2020). This raises a further question: is this mechanism redundant or inefficient, and could it be simplified within more flexible diffusion frameworks?

Motivated by these questions, we introduce the **Conditional Whitened Generative Models (CW-Gen)**. Our main contributions are:

- We develop a unified framework for conditional generation, CW-Gen, with two instantiations: the **Conditional Whitened Diffusion Model (CW-Diff)** and the **Conditional Whitened Flow Matching (CW-Flow)**. Several prior methods (Han et al., 2022; Li et al., 2024; Ye et al., 2025) can be viewed as special cases of this framework. Furthermore, CW-Gen allows seamless integration with diverse diffusion models.

- We provide theoretical analysis that establishes sufficient conditions under which CW-Gen improves sample quality, as stated in Theorem 1 and Theorem 2 in Appendix C.

- Motivated by Theorems 1 and 2, we propose a novel joint estimation procedure for the conditional mean and sliding-window covariance of time series. Empirically, it achieves high accuracy while effectively controlling covariance eigenvalues, ensuring stability and robustness in generative modeling.

- We integrate CW-Gen with six state-of-the-art generative models and evaluate them on five real-world datasets. Empirical results show consistent improvements in capturing non-stationarity, inter-variable dependencies, and overall sample quality, while also mitigating distribution shift.

## 2 PRELIMINARIES

### 2.1 DENOISING DIFFUSION PROBABILISTIC MODELS (DDPM)

Most of the diffusion models discussed in Section 1 follow the DDPM framework (Ho et al., 2020), which we review below in a general conditional setting. Let $(X_0, C)$ be a random vector with the joint distribution $P_{X,C}$, where $X_0 \in \mathbb{R}^{d_x}$ and $C \in \mathbb{R}^{d_c}$. The (conditional) DDPM aims to learn the conditional distribution $P_{X|C}$ and generate samples that match this distribution through a forward and a reverse process. In the forward process, Gaussian noises are gradually added into $X_0$ by a stochastic differential equation (SDE):

$$dX_\tau = -\tfrac{1}{2}\beta_\tau X_\tau d\tau + \sqrt{\beta_\tau}dW_\tau, \ \tau \in [0,1], \ X_0 \sim P_{X|C},$$

where $\beta_\tau > 0$ and $W_\tau$ is a Brownian motion in $\mathbb{R}^{d_x}$. We use $\tau$ for the time of diffusion throughout this paper, while $t$ is the index for time series. From the properties of Ornstein–Uhlenbeck-process (OU-process), we derive the marginal distribution of $X_\tau$:

$$X_\tau \overset{d}{=} \alpha_\tau X_0 + \sigma_\tau \epsilon, \ \epsilon \sim N(0, I_{d_x}),$$

where $\alpha_\tau := \exp\left\{-\int_0^\tau \beta_s ds/2\right\}$, $\sigma_\tau^2 := 1 - \alpha_\tau^2$, $\overset{d}{=}$ denotes equality in distribution, and $I_{d_x}$ is the $d_x$-dimensional identity matrix. By construction of $\beta_\tau$, the integral $\int_0^1 \beta_s ds$ is sufficiently large, so the distribution of $X_1$ (the terminal distribution) is well-approximated by $N(0, I_{d_x})$. In the reverse process, a standard Gaussian noise $\overset{\leftarrow}{X}_1$ is gradually denoised by an SDE:

$$d\overset{\leftarrow}{X}_\tau = \left[ -\tfrac{1}{2}\beta_\tau \overset{\leftarrow}{X}_\tau - \beta_\tau \nabla_x \log p_\tau(\overset{\leftarrow}{X}_\tau|C) \right] d\tau + \sqrt{\beta_\tau} d\overset{\leftarrow}{W}_\tau, \tag{1}$$

where $\tau$ starts from $\tau = 1$ and ends at $\tau = \tau_{\min}$, with $\tau_{\min}$ being an early stopping time close to 0, and $\overset{\leftarrow}{W}_\tau$ is a Brownian motion. In (1), $p_\tau(\cdot|C)$ and $\nabla_x \log p_\tau(\cdot|C)$ denote the conditional density and score function of $X_\tau$ given $C$, respectively. Since the conditional score function is intractable, Ho et al. (2020) and Song et al. (2021) proposed approximating it with a neural network $s_\theta$ parameterized by $\theta$, trained by minimizing:

$$\mathbb{E}_{(X_0,C),\tau,\epsilon} \left\| s_\theta \left( \alpha_\tau X_0 + \sigma_\tau \epsilon, C, \tau \right) + \epsilon/\sigma_\tau \right\|^2,$$

where $\tau \sim U(0,1]$ and $\epsilon \sim N(0, I_{d_x})$. Finally, substituting $\nabla_x \log p_\tau(\overset{\leftarrow}{X}_\tau|C)$ in (1) with $s_\theta(\overset{\leftarrow}{X}_\tau, C, \tau)$ yields the reverse process:

$$d\overset{\leftarrow}{X}_\tau = \left[ -\tfrac{1}{2}\beta_\tau \overset{\leftarrow}{X}_\tau - \beta_\tau s_\theta(\overset{\leftarrow}{X}_\tau, C, \tau) \right] d\tau + \sqrt{\beta_\tau} d\overset{\leftarrow}{W}_\tau, \ \tau \in [\tau_{\min}, 1].$$

## 2.2 FLOW MATCHING

Unlike diffusion models based on SDEs, Flow Matching (FM) employs an ordinary differential equation (ODE) to connect Gaussian noise $\epsilon \sim N(0, I_{d_x})$ with the data $X_0 \sim P_{X|C}$ (Lipman et al., 2023):

$$dX_\tau = (\epsilon - X_0)d\tau, \ \tau \in [0,1]. \tag{2}$$

A neural network $v_\psi$, parameterized by $\psi$, learns the vector field of (2) by minimizing:

$$\mathbb{E}_{(X_0,C),\tau,\epsilon}\|\epsilon - X_0 - v_\psi(X_0 + \tau(\epsilon - X_0), C, \tau)\|^2.$$

Given the learned vector field, FM generates samples by solving the ODE:

$$d\overleftarrow{X}_\tau = -v_\psi(\overleftarrow{X}_\tau, C, \tau)d\tau$$

from $\tau = 1$ to $\tau = \tau_{\min}$, where $\overleftarrow{X}_1$ is Gaussian noise. The final state $\overleftarrow{X}_{\tau_{\min}}$ is the generated sample.

# 3 THEORY AND JOINT MEAN–COVARIANCE ESTIMATOR (JMCE)

## 3.1 THEORETICAL FOUNDATION

A key question addressed in this subsection is how modifying the terminal distribution $N(0, I_{d_x})$ can enhance generation quality. The total variation distance between the generated distribution of a diffusion model and the true distribution grows as the convergence error of the forward process increases, where the latter involves the Kullback–Leibler divergence (KLD) between $P_{X|C}$ and the terminal distribution $D_{\mathrm{KL}}\big(P_{X|C} \,\|\, N(0, I_{d_x})\big)$ as a factor in the error (Oko et al., 2023; Chen et al., 2023; Fu et al., 2024). Hence, a smaller value of this KLD leads to samples that better match $P_{X|C}$. This insight motivates replacing the standard terminal distribution $N(0, I_{d_x})$ with $N(\mu_{X|C}, \Sigma_{X|C})$, where $\mu_{X|C}$ and $\Sigma_{X|C}$ are the true conditional mean and covariance of $X$ given $C$. Since these quantities are unknown in practice, they must be estimated by $\widehat{\mu}_{X|C}$ and $\widehat{\Sigma}_{X|C}$. The advantage of this replacement can then be measured by the reduction in

$$D_{\mathrm{KL}}\Big(P_{X|C} \,\|\, N(\widehat{\mu}_{X|C}, \widehat{\Sigma}_{X|C})\Big) \quad \text{relative to} \quad D_{\mathrm{KL}}\big(P_{X|C} \,\|\, N(0, I_{d_x})\big).$$

This raises the fundamental question of when replacing the terminal distribution $N(0, I_{d_x})$ with $N(\widehat{\mu}_{X|C}, \widehat{\Sigma}_{X|C})$ improves generation quality, which the following theorem addresses.

**Theorem 1** *Let $P_{X|C}$ denote the true conditional distribution of $X \in \mathbb{R}^{d_x}$ given $C$, with conditional mean $\mu_{X|C}$ and positive-definite conditional covariance $\Sigma_{X|C}$. Define $Q_0 := N(0, I_{d_x})$ and $\widehat{Q} := N(\widehat{\mu}_{X|C}, \widehat{\Sigma}_{X|C})$, where $\widehat{\mu}_{X|C}$ and $\widehat{\Sigma}_{X|C}$ are estimators of $\mu_{X|C}$ and $\Sigma_{X|C}$, respectively. Let $\widehat{\lambda}_{X|C,i}$ denote the $i$-th eigenvalues of $\widehat{\Sigma}_{X|C}$, for $i = 1, 2, \ldots, d_x$. A sufficient condition ensuring that $D_{\mathrm{KL}}(P_{X|C} \,\|\, \widehat{Q}) \le D_{\mathrm{KL}}(P_{X|C} \,\|\, Q_0)$ is:*

$$\left(\min_{i\in\{1,\ldots,d_x\}} \widehat{\lambda}_{X|C,i}\right)^{-1} \left(\|\mu_{X|C} - \widehat{\mu}_{X|C}\|_2^2 + \|\Sigma_{X|C} - \widehat{\Sigma}_{X|C}\|_N\right)$$
$$+ \sqrt{d_x}\,\|\Sigma_{X|C} - \widehat{\Sigma}_{X|C}\|_F \le \|\mu_{X|C}\|_2^2. \tag{3}$$

*where $\left\|\Sigma_{X|C} - \widehat{\Sigma}_{X|C}\right\|_N = \sum_{i=1}^{d_x} \widetilde{s}_i$ and $\widetilde{s}_i$ is the $i$-th singular value of $\Sigma_{X|C} - \widehat{\Sigma}_{X|C}$.*

Theorem 1 states that when (3) holds, replacing $Q_0$ with $\widehat{Q}$ reduces the KLD between $P_{X|C}$ and the terminal distribution, thereby improving generation quality. Importantly, it provides a foundation for designing loss functions to estimate $\mu_{\mathbf{X}|\mathbf{C}}$ and $\Sigma_{\mathbf{X}|\mathbf{C}}$, as detailed in Equation (4) below. We emphasize that the estimators of $\mu_{X|C}$ and $\Sigma_{X|C}$ are obtained by minimizing the sample counterpart of the left-hand side of (3), as detailed in the next subsection.

In order for (3) to hold, it is necessary to obtain accurate estimators of both $\mu_{X|C}$ and $\Sigma_{X|C}$. The estimation accuracy of $\Sigma_{X|C}$ is measured in terms of both the Frobenius norm and the nuclear norm, with the latter characterized by $\sum_{i=1}^{d_x} \widetilde{s}_i$. We employ a Cholesky decomposition

and introduce a penalty term into the loss function (4) to enforce that the smallest eigenvalue, $\min_{i \in \{1,\ldots,d_x\}} \{\widehat{\lambda}_{X|C,i}\}$, remains strictly positive and bounded away from zero, as detailed in the next subsection. Furthermore, in non-stationary time series, $\mu_{X|C}$ often exhibits sharp variations and thus deviates from zero. Consequently, (3) is more likely to hold when accurate estimators of both $\mu_{X|C}$ and $\Sigma_{X|C}$ are available. Conversely, (3) may fail to hold in unfavorable regimes—for example, when the signal magnitude $\|\mu_{X|C}\|_2^2$ is small, the estimators of $\mu_{X|C}$ and $\Sigma_{X|C}$ are inaccurate, or the inverse of the smallest eigenvalue becomes large. In such cases, incorporating the corresponding prior models can potentially degrade performance. In the next subsection, we design a novel loss function to mitigate this risk. A detailed discussion can be found in Appendix D.

We further identify the scenarios in which our proposed replacement outperforms TMDM and Ns-Diff (Li et al., 2024; Ye et al., 2025), as formally established in Theorem 2 in Appendix C.

## 3.2 JOINT MEAN–COVARIANCE ESTIMATOR (JMCE)

Theorem 1 establishes that accurate estimators of both the conditional mean and covariance can improve the quality of samples generated by diffusion models. Guided by the sufficient conditions (3), we design a novel Joint Mean–Covariance Estimator (JMCE).

In terms of time series, directly estimating the true conditional covariance is extremely challenging, as it is often highly complex and non-smooth, which makes consistent estimation difficult. Instead, the sliding-window covariance is preferable, as it not only offers more accurate approximations but also improves computational efficiency (Iwakura et al., 2008; Chen et al., 2024). Motivated by this, we estimate the sliding-window conditional covariance, rather than the true conditional covariance. Let $\widetilde{\Sigma}_{\mathbf{X}_0,t} \in \mathbb{R}^{d \times d}$ denote the sliding-window covariance at time $t$, and let $\widehat{\Sigma}_{\mathbf{X}_0,t|\mathbf{C}} \in \mathbb{R}^{d \times d}$ be an estimator of $\widetilde{\Sigma}_{\mathbf{X}_0,t}$ for $t = 1, \ldots, T_f$. We design a non-autoregressive model to simultaneously output:

$$\widehat{\mu}_{\mathbf{X}|\mathbf{C}}, \widehat{L}_{1|\mathbf{C}}, \ldots, \widehat{L}_{T_f|\mathbf{C}} = \mathrm{JMCE}(\mathbf{C})$$

with $\widehat{\Sigma}_{\mathbf{X}_0,t|\mathbf{C}} := \widehat{L}_{t|\mathbf{C}} \widehat{L}_{t|\mathbf{C}}^\top$, for $t = 1, \ldots, T_f$ and all $\widehat{L}_{t|\mathbf{C}}$ are lower-triangle matrices. This design, inspired by Cholesky decomposition, guarantees that all $\widehat{\Sigma}_{\mathbf{X}_0,t|\mathbf{C}}$ are positive semi-definite (PSD). The detailed algorithm of $\mathrm{JMCE}(\mathbf{C})$ can be found in Appendix B. In our implementation, we use a Non-stationary Transformer (Liu et al., 2022) as the backbone of JMCE. Based on (3) in Theorem 1, we construct the trainning loss in JMCE by combining three components: $\mathcal{L}_2 := \mathbb{E}_{(\mathbf{X}_0,\mathbf{C})} \left\| \mathbf{X}_0 - \widehat{\mu}_{\mathbf{X}|\mathbf{C}} \right\|_2^2$, $\mathcal{L}_F := \mathbb{E}_{(\mathbf{X}_0,\mathbf{C})} \sum_{t=1}^{T_f} \left\| \widetilde{\Sigma}_{\mathbf{X}_0,t} - \widehat{\Sigma}_{\mathbf{X}_0,t|\mathbf{C}} \right\|_F$, and $\mathcal{L}_{\mathrm{SVD}} := \mathbb{E}_{(\mathbf{X}_0,\mathbf{C})} \sum_{t=1}^{T_f} \left\| \widetilde{\Sigma}_{\mathbf{X}_0,t} - \widehat{\Sigma}_{\mathbf{X}_0,t|\mathbf{C}} \right\|_N$. The smallest eigenvalues of $\widehat{\Sigma}_{\mathbf{X}_0,t|\mathbf{C}}$ have a crucial impact on the magnitude of the left-hand side of inequality (3). We thus introduce a regularization term that enforces the smallest eigenvalues of $\widehat{\Sigma}_{\mathbf{X}_0,t|\mathbf{C}}$ to remain strictly positive and bounded away from zero, thereby avoiding numerical instability and rank deficiency. Let $\lambda_{\min}$ be a tunable hyperparameter. The penalty term is defined as:

$$\mathcal{R}_{\lambda_{\min}}\left(\widehat{\Sigma}_{\mathbf{X}_0,t|\mathbf{C}}\right) := \sum_{i=1}^{d} \mathrm{ReLU}\left(\lambda_{\min} - \widehat{\lambda}_{\widehat{\Sigma}_{\mathbf{X}_0,t|\mathbf{C}},i}\right),$$

where $\widehat{\lambda}_{\widehat{\Sigma}_{\mathbf{X}_0,t|\mathbf{C}},i}(i = 1, \ldots, d)$ denote the eigenvalues of $\widehat{\Sigma}_{\mathbf{X}_0,t|\mathbf{C}}$, and $\mathrm{ReLU}(x) = \max\{x, 0\}$. It is indicated that any eigenvalue smaller than $\lambda_{\min}$ will be penalized. The overall training loss in JMCE for the conditional mean and covariance is defined as:

$$\mathcal{L}_{\mathrm{JMCE}} = \mathcal{L}_2 + \mathcal{L}_{\mathrm{SVD}} + \lambda_{\min} \sqrt{d \cdot T_f} \mathcal{L}_F + w_{\mathrm{Eigen}} \cdot \sum_{t=1}^{T_f} \mathcal{R}_{\lambda_{\min}}\left(\widehat{\Sigma}_{\mathbf{X}_0,t|\mathbf{C}}\right), \tag{4}$$

where $w_{\mathrm{Eigen}}$ is a hyperparameter that controls the strength of the penalty. Empirically, we choose $w_{\mathrm{Eigen}} \sim \mathcal{O}(\lambda_{\min}^{-1})$. It is important to note that (4) is specifically designed to ensure that (3) holds.

The algorithm of the joint estimator can be found in Appendix B. JMCE excels at estimating the conditional mean and covariance while controlling the minimal eigenvalue. We conduct a substantial ablation study to show the advantages, and discuss them in Appendix E.

# 4 CONDITIONAL WHITENED GENERATIVE MODELS (CW-GEN)

In this section, we propose Conditionally whitened diffusion models (CW-Diff) and Conditionally whitened flow matching (CW-Flow). Together, we call them Conditionally Whitened Generative Models (CW-Gen).

## 4.1 CONDITIONALLY WHITENED DIFFUSION MODELS (CW-DIFF)

Our JMCE outputs $\widehat{\mu}_{\mathbf{X}|\mathbf{C}} \in \mathbb{R}^{d \times T_f}$ and $\widehat{\Sigma}_{\mathbf{X}_0|\mathbf{C}} := [\widehat{\Sigma}_{\mathbf{X}_0,1|\mathbf{C}}, \ldots, \widehat{\Sigma}_{\mathbf{X}_0,T_f|\mathbf{C}}] \in \mathbb{R}^{d \times d \times T_f}$. Since all $\widehat{\Sigma}_{\mathbf{X}_0,t|\mathbf{C}}$ are positive definite, we can compute $\widehat{\Sigma}_{\mathbf{X}_0|\mathbf{C}}^k := [\widehat{\Sigma}_{\mathbf{X}_0,1|\mathbf{C}}^k, \ldots, \widehat{\Sigma}_{\mathbf{X}_0,T_f|\mathbf{C}}^k] \in \mathbb{R}^{d \times d \times T_f}$ for $k \in \{-0.5, 0.5\}$ via eigen-decomposition. Let $\boldsymbol{\epsilon} := [\epsilon_1, \ldots, \epsilon_{T_f}] \in \mathbb{R}^{d \times T_f}$, where each column $\epsilon_t \sim N(0, I_d)$ and the columns $\epsilon_1, \ldots, \epsilon_{T_f}$ are mutually independent. We define the tensor operation

$$\widehat{\Sigma}_{\mathbf{X}_0|\mathbf{C}}^{0.5} \circ \boldsymbol{\epsilon} := [\widehat{\Sigma}_{\mathbf{X}_0,1|\mathbf{C}}^{0.5} \cdot \epsilon_1, \ldots, \widehat{\Sigma}_{\mathbf{X}_0,T_f|\mathbf{C}}^{0.5} \cdot \epsilon_{T_f}] \in \mathbb{R}^{d \times T_f}. \tag{5}$$

Accordingly, we say that a tensor follows $\mathcal{N}(\widehat{\mu}_{\mathbf{X}|\mathbf{C}}, \widehat{\Sigma}_{\mathbf{X}_0|\mathbf{C}})$ if it has the same distribution as $\widehat{\Sigma}_{\mathbf{X}_0|\mathbf{C}}^{0.5} \circ \boldsymbol{\epsilon} + \widehat{\mu}_{\mathbf{X}|\mathbf{C}}$. With this formulation, we define the forward process:

$$d\big(\mathbf{X}_\tau - \widehat{\mu}_{\mathbf{X}|\mathbf{C}}\big) = -\tfrac{1}{2}\beta_\tau\big(\mathbf{X}_\tau - \widehat{\mu}_{\mathbf{X}|\mathbf{C}}\big)d\tau + \sqrt{\beta_\tau} \cdot \widehat{\Sigma}_{\mathbf{X}_0|\mathbf{C}}^{0.5} \circ d\mathbf{W}_\tau, \ \tau \in [0,1], \ \mathbf{X}_0 \sim P_{\mathbf{X}|\mathbf{C}}, \tag{6}$$

where $\mathbf{W}_\tau$ is a Brownian motion in $\mathbb{R}^{d \times T_f}$. By the property of the OU-process, the terminal distribution of $\mathbf{X}_1$ is close to $\mathcal{N}(\widehat{\mu}_{\mathbf{X}|\mathbf{C}}, \widehat{\Sigma}_{\mathbf{X}_0|\mathbf{C}})$. A formal proof of the terminal distribution of (6) is provided in Appendix C. Furthermore, the following SDE is equivalent to (6):

$$d\,\widehat{\Sigma}_{\mathbf{X}_0|\mathbf{C}}^{-0.5} \circ \big(\mathbf{X}_\tau - \widehat{\mu}_{\mathbf{X}|\mathbf{C}}\big) = -\tfrac{1}{2}\beta_\tau \cdot \widehat{\Sigma}_{\mathbf{X}_0|\mathbf{C}}^{-0.5} \circ \big(\mathbf{X}_\tau - \widehat{\mu}_{\mathbf{X}|\mathbf{C}}\big)d\tau + \sqrt{\beta_\tau}d\mathbf{W}_\tau, \ \tau \in [0,1],$$

which implies that the diffusion processes can be directly performed on $\mathbf{X}_0^{\mathbf{CW}} := \widehat{\Sigma}_{\mathbf{X}_0|\mathbf{C}}^{-0.5} \circ \big(\mathbf{X}_0 - \widehat{\mu}_{\mathbf{X}|\mathbf{C}}\big)$. We call this operation conditional whitening (CW). Subtracting $\widehat{\mu}_{\mathbf{X}|\mathbf{C}}$ removes the non-stationary trends and seasonal effects in $\mathbf{X}_0$, while being operated by $\widehat{\Sigma}_{\mathbf{X}_0|\mathbf{C}}^{-0.5}$ addresses heteroscedasticity and mitigates linear correlations among features. The CW operation thus renders the data as stationary as possible and enables diffusion models to more effectively capture temporal and higher-order dependencies. Moreover, since it is a full-rank linear transformation, CW is entirely invertible. Building on these properties, we now formally write the forward process of the Conditional Whitened Diffusion Model (CW-Diff) as follows:

$$d\mathbf{X}_\tau^{\mathbf{CW}} = -\tfrac{1}{2}\beta_\tau \mathbf{X}_\tau^{\mathbf{CW}}d\tau + \sqrt{\beta_\tau}d\mathbf{W}_\tau, \ \tau \in [0,1], \tag{7}$$

with the initial state $\mathbf{X}_0^{\mathbf{CW}}$ satisfying $\big(\widehat{\Sigma}_{\mathbf{X}_0|\mathbf{C}}^{0.5} \circ \mathbf{X}_0^{\mathbf{CW}} + \widehat{\mu}_{\mathbf{X}|\mathbf{C}}\big) \sim P_{\mathbf{X}|\mathbf{C}}$. Correspondingly, we use a neural network $s_\theta^{\mathbf{CW}}$ to learn the score function of $\mathbf{X}_\tau^{\mathbf{CW}}$ given $\mathbf{C}$ by minimizing the following loss function:

$$\mathbb{E}_{(\mathbf{X}_0^{\mathbf{CW}}, \mathbf{C}), \tau, \boldsymbol{\epsilon}} \| s_\theta^{\mathbf{CW}}\big(\alpha_\tau \mathbf{X}_0^{\mathbf{CW}} + \sigma_\tau \boldsymbol{\epsilon}, \mathbf{C}, \tau\big) + \boldsymbol{\epsilon}/\sigma_\tau \|^2.$$

Let $\overleftarrow{\mathbf{X}}_1^{\mathbf{CW}} \sim \mathcal{N}(0, I_{d \times d \times T_f})$, where $I_{d \times d \times T_f} := [I_d, \ldots, I_d] \in \mathbb{R}^{d \times d \times T_f}$. Then, the reverse process of CW-Diff is given by:

$$d\overleftarrow{\mathbf{X}}_\tau^{\mathbf{CW}} = \Big[-\tfrac{1}{2}\beta_\tau \overleftarrow{\mathbf{X}}_\tau^{\mathbf{CW}} - \beta_\tau s_\theta^{\mathbf{CW}}(\overleftarrow{\mathbf{X}}_\tau^{\mathbf{CW}}, \mathbf{C}, \tau)\Big] d\tau + \sqrt{\beta_\tau}d\overleftarrow{\mathbf{W}}_\tau,$$

where $\tau$ decreases from 1 to $\tau_{\min}$, with $\tau_{\min}$ being an early stopping time close to 0. Finally, we obtain

$$\overleftarrow{\mathbf{X}}_{\tau_{\min}} = \widehat{\Sigma}_{\mathbf{X}_0|\mathbf{C}}^{0.5} \circ \overleftarrow{\mathbf{X}}_{\tau_{\min}}^{\mathbf{CW}} + \widehat{\mu}_{\mathbf{X}|\mathbf{C}}$$

by inverting the original CW operation. $\overleftarrow{\mathbf{X}}_{\tau_{\min}}$ is the final sample generated by CW-Diff approximating $P_{\mathbf{X}|\mathbf{C}}$.

The forward process in Equation (7) is consistent with that of DDPM. Furthermore, CW-Diff is readily extendable to TMDM, NsDiff, and other diffusion models. This extension is accomplished by replacing the initial variable $\mathbf{X}_0$ with its CW-transformed form $\mathbf{X}_0^{\mathbf{CW}}$. Within this framework, the task of learning the mean and sliding-window covariance in $\mathbf{X}_0^{\mathbf{CW}}$ may be interpreted as a form of residual learning, analogous to the mechanisms used in GBDT and XGBoost (Chen & Guestrin, 2016).

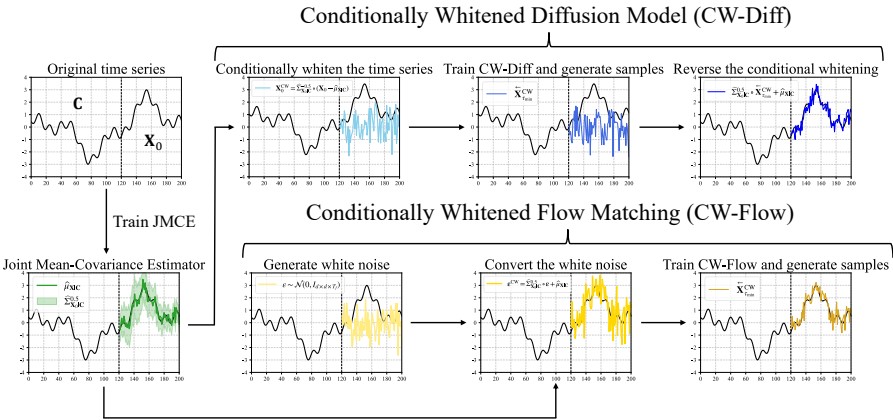

Figure 1: The flow chat of JMCE, CW-Diff and CW-Flow.

## 4.2 CONDITIONALLY WHITENED FLOW MATCHING (CW-FLOW)

In CW-Diff, the inverse matrices of $\widehat{\Sigma}_{\mathbf{X}_0,t|\mathbf{C}}$ are computed via eigen-decomposition, which requires a computational complexity of $\mathcal{O}(d^3 T_f)$. To reduce this cost and improve efficiency, we transition to the FM framework introduced in Section 2.2, where the estimated mean and covariance can be incorporated in a more efficient way.

The Conditional Whitened Flow Matching (CW-Flow) model employs an ODE to connect $\mathbf{X}_0 \sim P_{\mathbf{X}|\mathbf{C}}$ with a noise $\boldsymbol{\epsilon}^{\mathrm{CW}} \sim \mathcal{N}(\widehat{\mu}_{\mathbf{X}|\mathbf{C}}, \widehat{\Sigma}_{\mathbf{X}_0|\mathbf{C}})$:

$$d\mathbf{X}_\tau^{\mathrm{CW}} = \left(\boldsymbol{\epsilon}^{\mathrm{CW}} - \mathbf{X}_0\right)d\tau, \ \tau \in [0, 1].$$

Accordingly, the CW-Flow network $v_\psi^{\mathrm{CW}}$ is trained by minimizing:

$$\mathbb{E}_{(\mathbf{X}_0,\mathbf{C}),\tau,\boldsymbol{\epsilon}^{\mathrm{CW}}} \left\| \boldsymbol{\epsilon}^{\mathrm{CW}} - \mathbf{X}_0 - v_\psi^{\mathrm{CW}}(\mathbf{X}_0 + \tau(\boldsymbol{\epsilon}^{\mathrm{CW}} - \mathbf{X}_0), \mathbf{C}, \tau) \right\|^2.$$

CW-Flow then generates samples by solving the following ODE:

$$d\overleftarrow{\mathbf{X}}_\tau^{\mathrm{CW}} = -v_\psi^{\mathrm{CW}}(\overleftarrow{\mathbf{X}}_\tau^{\mathrm{CW}}, \mathbf{C}, \tau)d\tau, \ \overleftarrow{\mathbf{X}}_1^{\mathrm{CW}} \sim \mathcal{N}(\widehat{\mu}_{\mathbf{X}|\mathbf{C}}, \widehat{\Sigma}_{\mathbf{X}_0|\mathbf{C}}),$$

where $\tau$ starts from $\tau = 1$ and ends at $\tau = \tau_{\min}$. $\overleftarrow{\mathbf{X}}_{\tau_{\min}}^{\mathrm{CW}}$ is the final sample generated by CW-Flow approximating $P_{\mathbf{X}|\mathbf{C}}$. Compared with CW-Diff, CW-Flow does not require computing inverse matrices or reversing the CW operation of the final sample $\overleftarrow{\mathbf{X}}_{\tau_{\min}}^{\mathrm{CW}}$. The algorithms of CW-Diff and CW-Flow are provided in Appendix B. The flow chart of CW-Diff and CW-Flow can be found in Figure 1.

## 5 EXPERIMENTS

**Datasets:** We evaluate CW-Gen on five representative time series datasets—ETTh1, ETTh2, ILI, Weather, and Solar Energy—spanning various domains and temporal resolutions. Further details of the datasets can be found in Appendix E.1. For the ETT datasets, the training/validation/test split follows a 3:1:1 ratio, while for the other datasets we adopt a 7:1:2 ratio. Table 1 presents the dataset properties and the win rate of CW-Gen, computed as the proportion of cases where CW-Gen outperforms competing methods, based on the results in Tables 2-6.

**Baselines:** We evaluate five diffusion models and one flow matching model for time series forecasting (denoted as Raw), and further integrate all six generative models with our CW-Diff and CW-Flow approaches (denoted as CW). Specifically, the baselines include TimeDiff (Shen & Kwok, 2023), SSSD (Alcaraz & Strodthoff, 2023), Diffusion-TS (Yuan & Qiao, 2024), TMDM (Li et al., 2024), NsDiff (Li et al., 2024), and FlowTS (Hu et al., 2025). Among them, TimeDiff, TMDM, and NsDiff are prior-informed methods.

**Metrics:** We evaluate the predictive performance with six metrics: Continuous Ranked Probability Score (CRPS) (Matheson & Winkler, 1976), Quantile Interval Coverage Error (QICE) (Han et al., 2022), Probabilistic Correlation score (ProbCorr), Conditional Context Fréchet Inception Distance (Conditional FID) (Yue et al., 2022), Probabilistic mean square error (ProbMSE), and Probabilistic mean average error (ProbMAE). Formal definitions can be found in Appendix E.2. We also provide the results for ProbMSE and ProbMAE in Tables 7 and 8 in Appendix E.3.

**Settings:** During evaluation, $\mathbf{X}_0$ and $\mathbf{C}$ refers to non-overlapping subsequences drawn from the test set, where $\mathbf{C}$ denotes the historical observations and $\mathbf{X}_0$ the corresponding future series. We adopt the widely used long-term forecasting setting with a historical length of 168 and a future horizon of 192 (Shen & Kwok, 2023; Ye et al., 2025). Inspired by NsDiff, The sliding-window covariance is computed with a window size of 95, except for ILI, where it is set to 15 (Ye et al., 2025). In the JMCE loss (4), $\lambda_{\min}$ is fixed at 0.1, and the penalty weight $w_{\text{Eigen}}$ is set to 50. All diffusion models follow their default diffusion schedules, and the number of sampling steps is set to 50 (20 for NsDiff). We train JMCE and CW-Gen on the training set, select the model checkpoint with the lowest loss on validation set, and then perform evaluation on the test set. Each model generates 100 samples for evaluation. On each dataset, we train every model 10 times with different random seeds and report the mean and one standard deviation of the four metrics. We also conduct extensive ablation studies on JMCE, which can be found in Appendix E.4. The other parameters are provided in Appendix F.

**Results:** As shown in Tables 2-6, CW-Gen reduces CRPS and QICE in a substantial number of cases, indicating improvements in predictive accuracy. Moreover, it consistently lowers ProbCorr and Conditional FID, with only minor exceptions, showing that CW-Gen enables models to better capture feature correlations in time series and to enhance overall sample quality. Moreover, as shown in Tables 7 and 8, our CW-Gen method improves the ProbMSE metric in 76.67% and the ProbMAE metric in 80.00% of the evaluated model–dataset combinations. This demonstrates that, in addition to enhancing probabilistic forecasting ability, CW-Gen also strengthens the point forecasting performance of the models.

**Illustrations:** In Figure 2, we illustrate representative results of representative generative models combined with CW-Gen. Among them, Diffusion-TS serves as a typical diffusion model, NsDiff is a diffusion based model augmented by priors, and FlowTS is based on flow matching. Comparing NsDiff and CW-Gen with the other models, we observe that generative models without priors tend to generate sample with shifted means and variances, which we attribute to distribution shifts between the training and test sets. This observation highlights the benefit of incorporating priors in probabilistic time series forecasting, as they can effectively mitigate such distribution shifts. In contrast, CW-Diffusion-TS and CW-FlowTS, which leverage JMCE as priors, exhibit no noticeable mean shift compared to Diffusion-TS and FlowTS. Moreover, the individual samples generated by CW-Diffusion-TS and CW-FlowTS achieve finer resolution and better capture the peaks in Dimension 1 than their non-CW counterparts. Compared with NsDiff, CW-NsDiff produces more accurate sample means and smaller standard deviations in Dimension 1, which contributes to more reliable uncertainty quantification. More illustrations can be found in Figure 3 in Appendix E.

Table 1: Dataset descriptions, including dimensions $d$, frequencies, total length of time series, length of historical observations $T_h$, length of future time series $T_f$, and win rates of our CW methods. Win rate refers to the proportion that our CW method outperforms original method.

| Dataset | Dimension | Frequency | Total length | $T_h$ | $T_f$ | Win rate of CW-Gen |
|---------|-----------|-----------|--------------|-------|-------|--------------------|
| ETTh1 | 7 | 1 Hour | 14,400 | 168 | 192 | $22/24 \approx 91.67\%$ |
| ETTh2 | 7 | 1 Hour | 14,400 | 168 | 192 | $22/24 \approx 91.67\%$ |
| ILI | 7 | 1 Week | 966 | 52 | 36 | $20/24 \approx 83.33\%$ |
| Weather | 21 | 10 Minutes | 52,696 | 168 | 192 | $22/24 \approx 91.67\%$ |
| Solar Energy | 137 | 10 Minutes | 52,560 | 168 | 192 | $19/24 \approx 79.17\%$ |

Table 2: Metrics for models trained on original ETTh1 (Raw) and conditionally whitened ETTh1 (CW). Each experiment is repeated by 10 times, and standard deviations are provided in brackets. The better results between Raw and CW are underlined. The win rates of every metric of Raw and CW-Gen models are also provided.

| Model | CRPS (↓) | | QICE (↓) | | ProbCorr (↓) | | Conditional FID (↓) | |
|---|---|---|---|---|---|---|---|---|
| (ETTh1) | Raw | CW | Raw | CW | Raw | CW | Raw | CW |
| TimeDiff | 0.787 | 0.505 | 9.038 | 8.821 | 0.320 | 0.243 | 19.008 | 6.788 |
| (2023) | (0.051) | (0.040) | (0.946) | (1.916) | (0.012) | (0.027) | (6.088) | (5.425) |
| SSSD | 0.836 | 0.524 | 11.624 | 4.838 | 0.326 | 0.238 | 40.887 | 9.265 |
| (2023) | (0.153) | (0.085) | (1.312) | (1.921) | (0.032) | (0.024) | (17.601) | (5.003) |
| Diffusion | 0.626 | 0.445 | 3.002 | 2.963 | 0.401 | 0.266 | 81.563 | 7.686 |
| -TS (2024) | (0.027) | (0.024) | (0.838) | (0.887) | (0.017) | (0.012) | (60.905) | (2.751) |
| TMDM | 0.472 | 0.440 | 3.360 | 4.555 | 0.230 | 0.213 | 9.931 | 3.831 |
| (2024) | (0.031) | (0.001) | (1.055) | (0.855) | (0.014) | (0.001) | (4.439) | (0.431) |
| NsDiff | 0.407 | 0.431 | 1.792 | 1.249 | 0.214 | 0.206 | 35.261 | 8.820 |
| (2025) | (0.032) | (0.029) | (0.682) | (0.228) | (0.014) | (0.010) | (7.785) | (1.541) |
| FlowTS | 0.724 | 0.488 | 8.820 | 8.817 | 0.354 | 0.254 | 39.793 | 4.865 |
| (2025) | (0.135) | (0.020) | (2.631) | (0.460) | (0.060) | (0.021) | (24.853) | (0.563) |
| Win rate | 16.7% | 83.3% | 16.7% | 83.3% | 0.0% | 100.0% | 0.0% | 100.0% |

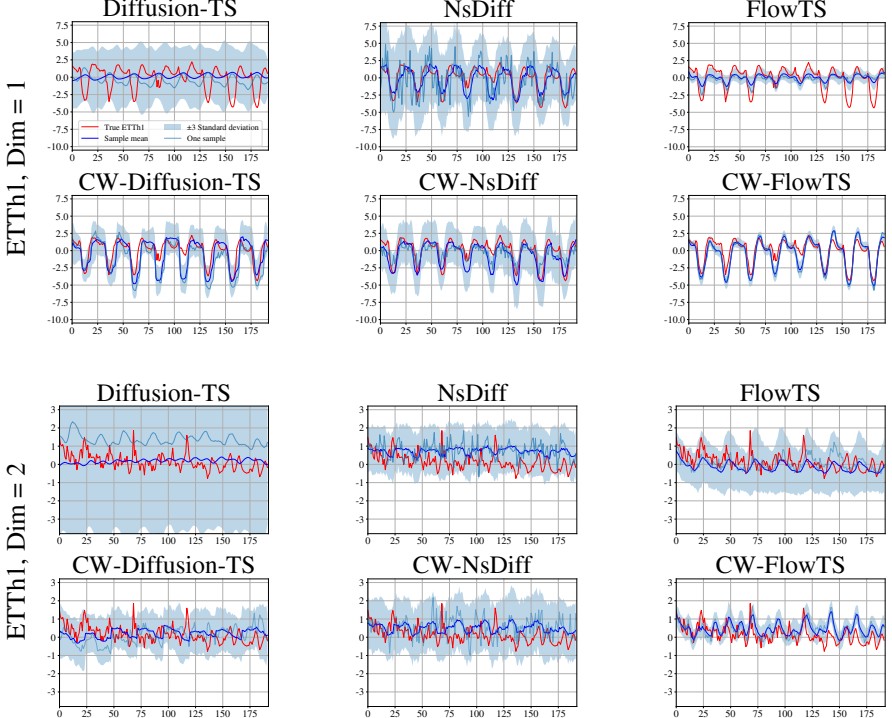

Figure 2: Comparison of Diffusion-TS, NsDiff, FlowTS, and their CW variants on ETTh1 across Dimensions 1 and 2. True ETTh1 means the real time series from ETTh1 dataset. Sample mean and standrad deviation refer to the mean and standrad deviation of 100 samples generated by generative models. One sample refers to a randomly chosen instance among the 100 generated samples.

## 6 CONCLUSION

In this work, we establish for the first time a sufficient condition that reduces the KL divergence between a conditional distribution and the terminal distribution of a diffusion model. By tightening this KL divergence, we obtain a sharper bound on the total variation distance between the generated distribution of the diffusion model and the true distribution. Building on this result, we design the Joint Mean–Covariance Estimator (JMCE), which jointly estimates the conditional mean and the conditional sliding-window covariance while controlling the behavior of the minimal eigenvalue. We then use JMCE as a data-driven prior to conditionally whiten the original data, and train diffusion models on the whitened space, yielding the Conditionally Whitened Diffusion Model (CW-Diff). Similarly, by modifying the terminal distribution of flow matching, we introduce the Conditionally Whitened Flow Model (CW-Flow). Together, we refer to these as CW-Gen. We evaluate CW-Gen on five real-world time series datasets using six generative models and four evaluation metrics. Experimental results demonstrate that CW-Gen consistently improves model performance in most cases.

## 7 REPRODUCIBILITY STATEMENT

Our proposed CW-Gen models are presented in Section 4, and the corresponding algorithms are provided in Section B. Theorem 1 can be found in Section 3, with its proof given in Appendix C.2. In addition, we introduce Theorem 2 in Appendix C, and its proof is provided in Appendix C.3. Detailed descriptions of the datasets, models, and evaluation metrics used in our experiments are included in Section 5, Appendix E and Appendix F. The code is available at: `https://github.com/Yanfeng-Yang-0316/Conditionally_whitened_generative_models`.

## ACKNOWLEDGMENTS

This work has been partially supported by JST CREST JPMJCR2015 and JSPS Grant-in-Aid for Transformative Research Areas (A) 22H05106. Yanfeng Yang is supported by JST SPRING, Japan Grant Number JPMJSP2104. Dr. Ziqi Chen's work was partially supported by National Natural Science Foundation of China (NSFC) (12271167 and 72331005).

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

# A    RELATED WORKS

## A.1    TRANSFORMER-MODULATED DIFFUSION MODELS (TMDM) AND NON-STATIONARY DIFFUSION MODELS (NSDIFF)

Han et al. (2022) incorporated an estimator of the conditional mean into the DDPM framework, naming this approach CARD. TMDM later adopted this framework for time series forecasting (Li et al., 2024). Recall $\widehat{\mu}_{\mathbf{X}|\mathbf{C}} \in \mathbb{R}^{d \times T_f}$ is an estimator of $\mathbf{X}_0$ given $\mathbf{C}$. The discrete forward process of TMDM is:

$$\mathbf{X}_{[n]} = \sqrt{1 - \beta_{[n]}}\mathbf{X}_{[n-1]} + \left(1 - \sqrt{1 - \beta_{[n]}}\right)\widehat{\mu}_{\mathbf{X}|\mathbf{C}} + \sqrt{\beta_{[n]}}\boldsymbol{\epsilon}_{[n]},\ n = 1, \dots, N,$$

where $\mathbf{X}_{[0]} \sim P_{\mathbf{X}|\mathbf{C}}$ and $\boldsymbol{\epsilon}_{[n]} \overset{i.i.d}{\sim} \mathcal{N}(0, I_{d \times d \times T_f})$ for all $n$. $N$ is a sufficient large index. By incorporating $\widehat{\mu}_{\mathbf{X}|\mathbf{C}}$ into the forward process, the model is able to more effectively handle the non-stationary trends and seasonal effects.

To further mitigate heteroscedasticity, Ye et al. (2025) proposed NsDiff, which introduces estimators for the sliding-window variance of the time series. Let $\widetilde{\sigma}^2_{\mathbf{X}_0,t} \in \mathbb{R}^{d \times d}$ denote the diagonal sliding-variance matrix at time $t$. We then define $\widetilde{\sigma}^k_{\mathbf{X}_0} := \left[\widetilde{\sigma}^k_{\mathbf{X}_0,1}, \dots, \widetilde{\sigma}^k_{\mathbf{X}_0,T_f}\right] \in \mathbb{R}^{d \times d \times T_f}$, for $k \in \{1,2\}$. We also introduce $\widehat{\sigma}^2_{\mathbf{X}_0|\mathbf{C}} := \left[\widehat{\sigma}^2_{\mathbf{X}_0,1|\mathbf{C}}, \dots, \widehat{\sigma}^2_{\mathbf{X}_0,T_f|\mathbf{C}}\right] \in \mathbb{R}^{d \times d \times T_f}$ as an estimator of $\widetilde{\sigma}^k_{\mathbf{X}_0}$. The discrete forward processes of NsDiff is:

$$\mathbf{X}_{[n]} = \sqrt{1 - \beta_{[n]}}\mathbf{X}_{[n-1]} + \left(1 - \sqrt{1 - \beta_{[n]}}\right)\widehat{\mu}_{\mathbf{X}|\mathbf{C}} + \left[\beta_{[n]}\widetilde{\sigma}^2_{\mathbf{X}_0} + \beta^2_{[n]}\left(\widehat{\sigma}^2_{\mathbf{X}_0|\mathbf{C}} - \widetilde{\sigma}^2_{\mathbf{X}_0}\right)\right]^{0.5} \circ \boldsymbol{\epsilon}_{[n]}. \quad (8)$$

In the reverse process, $\widetilde{\sigma}^2_{\mathbf{X}_0}$ is unknown; NsDiff estimates it by exploiting both $\mathbf{X}_{[n]}$ and $\widehat{\sigma}^2_{\mathbf{X}_0|\mathbf{C}}$, rather than relying solely on $\widehat{\sigma}^2_{\mathbf{X}_0|\mathbf{C}}$. This yields a more accurate estimate of $\widetilde{\sigma}^2_{\mathbf{X}_0}$ and improves performance.

However, NsDiff also has several limitations. First, as shown in Equation (8), the sliding-variance plays a crucial role, yet its estimator is not effectively exploited. In the reverse process, estimation is carried out by solving $d$ univariate quadratic equations, rendering the sampling procedure unnecessarily complicated. When solving these equations, failures may occur. To mitigate this issue, the sampling steps of reverse process should be set to a relatively small value (e.g., 20) in order to reduce the probability of failure. Second, although NsDiff incorporates the diagonal sliding-variance $\widetilde{\sigma}^2_{\mathbf{X}_0}$, it does not include the covariance, thereby ignoring correlations in multivariate time series. Third, in the reverse process of NsDiff, it begins with a Gaussian noise with variance $\widehat{\sigma}^2_{\mathbf{X}_0|\mathbf{C}}$. This is inconsistent with the terminal distribution whose variance is $\widetilde{\sigma}^2_{\mathbf{X}_0}$.

# B    ALGORITHMS OF JMCE, CW-DIFF AND CW-FLOW

The training procedure of JMCE is summarized in Algorithm 1. The training and sampling routines of CW-Diff are presented in Algorithms 2 and 3, respectively. Similarly, the corresponding training and sampling procedures for CW-Flow are provided in Algorithms 4 and 5.

# C    THEOREMS AND PROOFS

## C.1    THEOREMS

**Theorem 2** *Define the Bregman divergence (Harandi et al., 2014) between two matrices $M_1, M_2$ of the same dimension as $B(M_1, M_2) = D_{\mathrm{KL}}(N(0, M_1) \| N(0, M_2)) = 0.5\left(\mathrm{Tr}(M_2^{-1}M_1 - I_{d_x}) - \log\left|M_2^{-1}M_1\right|\right)$. Let $M_{X|C} \in \mathbb{R}^{d_x \times d_x}$ be a positive-definite matrix and $\widehat{M}_{X|C}$ be a positive-definite estimator of $M_{X|C}$. Let $\widetilde{M}_{X|C} \in \left\{\widehat{M}_{X|C}, M_{X|C}\right\}$. A sufficient condition for the inequality $D_{\mathrm{KL}}(P_{X|C} \| \widehat{Q}) \leq D_{\mathrm{KL}}(P_{X|C} \| N(\widehat{\mu}_{X|C}, \widetilde{M}_{X|C}))$ to hold is*

$$\left\|\mu_{X|C} - \widehat{\mu}_{X|C}\right\|^2_2 \left(\left\|\widehat{\Sigma}^{-1}_{X|C} - \Sigma^{-1}_{X|C}\right\|_2 + \left\|\Sigma^{-1}_{X|C}\right\|_2 + \left\|\widetilde{M}^{-1}_{X|C}\right\|_2\right)$$
$$+ 2B(\Sigma_{X|C}, \widehat{\Sigma}_{X|C}) + d_x \left\|\widetilde{M}^{-1}_{X|C} - M^{-1}_{X|C}\right\|_2 \left\|\Sigma_{X|C} + M_{X|C}\right\|_2 \leq 2B(\Sigma_{X|C}, M_{X|C})$$
$$(9)$$

---

**Algorithm 1** Joint Mean-Covariance Estimator (JMCE)

---

**Input**: $(\mathbf{X}_0, \mathbf{C})$ in training set, hyperparameters $\lambda_{\min}, w_{\text{Eigen}}$.
**Output**: A joint mean-covariance estimator $\text{JMCE}(\cdot)$.

1: Calculate sliding-window covariances $\widetilde{\Sigma}_{\mathbf{X}_0,1}, \ldots, \widetilde{\Sigma}_{\mathbf{X}_0,T_f}$ of $\mathbf{X}_0$
2: Initialize a non-autoregressive model $\text{JMCE}(\cdot)$
3: **while** not converge **do**
4:      Calculate $\widehat{\mu}_{\mathbf{X}|\mathbf{C}}, \widehat{L}_{1|\mathbf{C}}, \ldots, \widehat{L}_{T_f|\mathbf{C}} = \text{JMCE}(\mathbf{C})$
5:      **for** $t = 1, \ldots, T_f$ **do**
6:          Let $\widehat{\Sigma}_{\mathbf{X}_0,t|\mathbf{C}} = \widehat{L}_{t|\mathbf{C}}\widehat{L}_{t|\mathbf{C}}^{\top}$
7:          Perform eigen-decomposition of $\widehat{\Sigma}_{\mathbf{X}_0,t|\mathbf{C}}$ and obtain eigenvalues $\widehat{\lambda}_{\widehat{\Sigma}_{\mathbf{X}_0,t|\mathbf{C}},i}, i = 1, \ldots, d$
8:          Perform singular value decomposition (SVD) of $\widetilde{\Sigma}_{\mathbf{X}_0,t} - \widehat{\Sigma}_{\mathbf{X}_0,t|\mathbf{C}}$ and obtain singular values $\widetilde{s}_{i,t}, i = 1, \ldots, d$
9:      **end for**
10:      Calculate $L_2 = \left\|\mathbf{X}_0 - \widehat{\mu}_{\mathbf{X}|\mathbf{C}}\right\|^2$, $L_F = \sum_{t=1}^{T_f} \left\|\widetilde{\Sigma}_{\mathbf{X}_0,t} - \widehat{\Sigma}_{\mathbf{X}_0,t|\mathbf{C}}\right\|_F$, $L_{\text{SVD}} = \sum_{t=1}^{T_f}\sum_{i=1}^{d} \widetilde{s}_{i,t}$,
11:      $R_{\lambda_{\min}} = \sum_{t=1}^{T_f}\sum_{i=1}^{d} \text{ReLU}(\lambda_{\min} - \widehat{\lambda}_{\widehat{\Sigma}_{\mathbf{X}_0,t|\mathbf{C}},i})$
12:      Calculate $L_{\text{JMCE}} = L_2 + L_{\text{SVD}} + \lambda_{\min}\sqrt{d \cdot T_f}L_F + w_{\text{Eigen}}R_{\lambda_{\min}}$
13:      Calculate $\nabla L_{\text{JMCE}}$ and update the parameters of $\text{JMCE}(\cdot)$
14: **end while**
15: **return** $\text{JMCE}(\cdot)$

---

**Algorithm 2** Train a Conditionally Whitened Diffusion model (CW-Diff)

---

**Input**: $(\mathbf{X}_0, \mathbf{C})$ in training set, diffusion schedule $\beta_\tau, \tau \in [0, 1]$, a JMCE model $\text{JMCE}(\cdot)$.
**Output**: A trained neural network $s_\theta^{\text{CW}}$.

1: Calculate $\widehat{\mu}_{\mathbf{X}|\mathbf{C}}, \widehat{L}_{1|\mathbf{C}}, \ldots, \widehat{L}_{T_f|\mathbf{C}} = \text{JMCE}(\mathbf{C})$
2: Calculate $\widehat{\Sigma}_{\mathbf{X}_0|\mathbf{C}} = [\widehat{L}_{1|\mathbf{C}}\widehat{L}_{1|\mathbf{C}}^{\top}, \ldots, \widehat{L}_{T_f|\mathbf{C}}\widehat{L}_{T_f|\mathbf{C}}^{\top}]$
3: Calculate $\widehat{\Sigma}_{\mathbf{X}_0|\mathbf{C}}^{-0.5} = [(\widehat{L}_{1|\mathbf{C}}\widehat{L}_{1|\mathbf{C}}^{\top})^{-0.5}, \ldots, (\widehat{L}_{T_f|\mathbf{C}}\widehat{L}_{T_f|\mathbf{C}}^{\top})^{-0.5}]$
4: Calculate $\mathbf{X}_0^{\text{CW}} = \widehat{\Sigma}_{\mathbf{X}_0|\mathbf{C}}^{-0.5} \circ (\mathbf{X}_0 - \widehat{\mu}_{\mathbf{X}|\mathbf{C}})$
5: Initialize a neural network $s_\theta^{\text{CW}}$
6: **while** not converge **do**
7:      Draw $\tau \sim U(0, 1]$
8:      Draw $\boldsymbol{\epsilon} \sim \mathcal{N}(0, I_{d \times d \times T_f})$
9:      Calculate $\alpha_\tau = \exp\left\{-\int_0^\tau \beta_s ds/2\right\}$ and $\sigma_\tau^2 = 1 - \alpha_\tau^2$
10:      Calculate $L_{\text{Diff}} = \|s_\theta^{\text{CW}}\left(\alpha_\tau \mathbf{X}_0^{\text{CW}} + \sigma_\tau\boldsymbol{\epsilon}, \mathbf{C}, \tau\right) + \boldsymbol{\epsilon}/\sigma_\tau\|^2$
11:      Calculate $\nabla_\theta L_{\text{Diff}}$ and update the parameters of $s_\theta^{\text{CW}}$
12: **end while**
13: **return** $s_\theta^{\text{CW}}$

---

Theorem 2 characterizes when replacing the terminal distribution $N(\widehat{\mu}_{X|C}, \widetilde{M}_{X|C})$ with $\widehat{Q}$ leads to a reduction in the KLD between $P_{X|C}$ and the terminal distribution. This reduction occurs when

- The estimator $\widehat{\mu}_{X|C}$ closely approximates the true conditional mean $\mu_{X|C}$.

- $\widehat{M}_{X|C}$ is a reliable estimator of $M_{X|C}$, with the eigenvalues of $M_{X|C}$ bounded away from both zero and infinity.

- The conditional covariance matrix $\Sigma_{X|C}$ has eigenvalues bounded away from zero and infinity, deviates from $M_{X|C}$, and is better approximated by a well-estimated $\widehat{\Sigma}_{X|C}$ than by $\widetilde{M}_{X|C}$. This deviation is formally measured by the Bregman divergence.

Setting $\widetilde{M}_{X|C} = M_{X|C} = I_{d_x}$ and excluding $\widehat{M}_{X|C}$ in Theorem 2 delineates the scenarios where our proposed replacement improves upon TMDM (Li et al., 2024). Similarly, taking $M_{X|C} = \sigma_{X|C}^2$,

---

**Algorithm 3** Sampling from a trained CW-Diff model

---

**Input**: Historical observation $\mathbf{C}$ in test set, diffusion schedule $\beta_\tau, \tau \in [0,1]$, a JMCE model $\mathtt{JMCE}(\cdot)$, a trained neural network $s_\theta^{\mathrm{CW}}$, an early stopping time $\tau_{\min}$.
**Output**: Samples approximate $P_{\mathbf{X}|\mathbf{C}}$.

1: Calculate $\widehat{\mu}_{\mathbf{X}|\mathbf{C}}, \widehat{L}_{1|\mathbf{C}}, \ldots, \widehat{L}_{T_f|\mathbf{C}} = \mathtt{JMCE}(\mathbf{C})$
2: Calculate $\widehat{\Sigma}_{\mathbf{X}_0|\mathbf{C}} = [\widehat{L}_{1|\mathbf{C}}\widehat{L}_{1|\mathbf{C}}^\top, \ldots, \widehat{L}_{T_f|\mathbf{C}}\widehat{L}_{T_f|\mathbf{C}}^\top]$
3: Calculate $\widehat{\Sigma}_{\mathbf{X}_0|\mathbf{C}}^{0.5} = [(\widehat{L}_{1|\mathbf{C}}\widehat{L}_{1|\mathbf{C}}^\top)^{0.5}, \ldots, (\widehat{L}_{T_f|\mathbf{C}}\widehat{L}_{T_f|\mathbf{C}}^\top)^{0.5}]$
4: Draw $\overleftarrow{\mathbf{X}}_1^{\mathrm{CW}} \sim \mathcal{N}(0, I_{d \times d \times T_f})$
5: Solve SDE $d\overleftarrow{\mathbf{X}}_\tau^{\mathrm{CW}} = \left[ -\frac{1}{2}\beta_\tau \overleftarrow{\mathbf{X}}_\tau^{\mathrm{CW}} - \beta_\tau s_\theta^{\mathrm{CW}}(\overleftarrow{\mathbf{X}}_\tau^{\mathrm{CW}}, \mathbf{C}, \tau) \right] d\tau + \sqrt{\beta_\tau} d\overleftarrow{\mathbf{W}}_\tau$ from $\tau = 1$ to $\tau = \tau_{\min}$, and get $\overleftarrow{\mathbf{X}}_{\tau_{\min}}^{\mathrm{CW}}$
6: **return** $\widehat{\Sigma}_{\mathbf{X}_0|\mathbf{C}}^{0.5} \circ \overleftarrow{\mathbf{X}}_{\tau_{\min}}^{\mathrm{CW}} + \widehat{\mu}_{\mathbf{X}|\mathbf{C}}$

---

**Algorithm 4** Train a Conditionally Whitened Flow Matching (CW-Flow)

---

**Input**: $(\mathbf{X}_0, \mathbf{C})$ in training set, a JMCE model $\mathtt{JMCE}(\cdot)$.
**Output**: A trained neural network $v_\psi^{\mathrm{CW}}$.

1: Calculate $\widehat{\mu}_{\mathbf{X}|\mathbf{C}}, \widehat{L}_{1|\mathbf{C}}, \ldots, \widehat{L}_{T_f|\mathbf{C}} = \mathtt{JMCE}(\mathbf{C})$
2: Calculate $\widehat{\Sigma}_{\mathbf{X}_0|\mathbf{C}} = [\widehat{L}_{1|\mathbf{C}}\widehat{L}_{1|\mathbf{C}}^\top, \ldots, \widehat{L}_{T_f|\mathbf{C}}\widehat{L}_{T_f|\mathbf{C}}^\top]$
3: Calculate $\widehat{\Sigma}_{\mathbf{X}_0|\mathbf{C}}^{0.5} = [(\widehat{L}_{1|\mathbf{C}}\widehat{L}_{1|\mathbf{C}}^\top)^{0.5}, \ldots, (\widehat{L}_{T_f|\mathbf{C}}\widehat{L}_{T_f|\mathbf{C}}^\top)^{0.5}]$
4: Initialize a neural network $v_\psi^{\mathrm{CW}}$
5: **while** not converge **do**
6:    Draw $\tau \sim U(0,1]$
7:    Draw $\epsilon^{\mathrm{CW}} \sim \mathcal{N}(0, I_{d \times d \times T_f})$
8:    Calculate $\epsilon^{\mathrm{CW}} = \widehat{\Sigma}_{\mathbf{X}_0|\mathbf{C}}^{0.5} \circ \epsilon^{\mathrm{CW}} + \widehat{\mu}_{\mathbf{X}|\mathbf{C}}$
9:    Calculate $L_{\mathrm{Flow}} = \|\epsilon^{\mathrm{CW}} - \mathbf{X}_0 - v_\psi^{\mathrm{CW}}(\mathbf{X}_0 + \tau(\epsilon^{\mathrm{CW}} - \mathbf{X}_0), \mathbf{C}, \tau)\|^2$
10:    Calculate $\nabla_\psi L_{\mathrm{Flow}}$ and update the parameters of $v_\psi^{\mathrm{CW}}$
11: **end while**
12: **return** $v_\psi^{\mathrm{CW}}$

---

**Algorithm 5** Sampling from a trained CW-Flow model

---

**Input**: Historical observation $\mathbf{C}$ in test set, a JMCE model $\mathtt{JMCE}(\cdot)$, a trained neural network $v_\psi^{\mathrm{CW}}$, an early stopping time $\tau_{\min}$.
**Output**: Samples approximate $P_{\mathbf{X}|\mathbf{C}}$.

1: Calculate $\widehat{\mu}_{\mathbf{X}|\mathbf{C}}, \widehat{L}_{1|\mathbf{C}}, \ldots, \widehat{L}_{T_f|\mathbf{C}} = \mathtt{JMCE}(\mathbf{C})$
2: Calculate $\widehat{\Sigma}_{\mathbf{X}_0|\mathbf{C}} = [\widehat{L}_{1|\mathbf{C}}\widehat{L}_{1|\mathbf{C}}^\top, \ldots, \widehat{L}_{T_f|\mathbf{C}}\widehat{L}_{T_f|\mathbf{C}}^\top]$
3: Calculate $\widehat{\Sigma}_{\mathbf{X}_0|\mathbf{C}}^{0.5} = [(\widehat{L}_{1|\mathbf{C}}\widehat{L}_{1|\mathbf{C}}^\top)^{0.5}, \ldots, (\widehat{L}_{T_f|\mathbf{C}}\widehat{L}_{T_f|\mathbf{C}}^\top)^{0.5}]$
4: Draw $\overleftarrow{\mathbf{X}}_1^{\mathrm{CW}} \sim \mathcal{N}(0, I_{d \times d \times T_f})$
5: Calculate $\overleftarrow{\mathbf{X}}_1^{\mathrm{CW}} = \widehat{\Sigma}_{\mathbf{X}_0|\mathbf{C}}^{0.5} \circ \overleftarrow{\mathbf{X}}_1^{\mathrm{CW}} + \widehat{\mu}_{\mathbf{X}|\mathbf{C}}$
6: Solve ODE $d\overleftarrow{\mathbf{X}}_\tau^{\mathrm{CW}} = -v_\psi^{\mathrm{CW}}(\overleftarrow{\mathbf{X}}_\tau^{\mathrm{CW}}, \mathbf{C}, \tau)d\tau$ from $\tau = 1$ to $\tau = \tau_{\min}$, and get $\overleftarrow{\mathbf{X}}_{\tau_{\min}}^{\mathrm{CW}}$
7: **return** $\overleftarrow{\mathbf{X}}_{\tau_{\min}}^{\mathrm{CW}}$

---

the matrix that contains only the main diagonal elements of $\Sigma_{X|C}$, and $\widetilde{M}_{X|C} = \widehat{M}_{X|C} = \widehat{\sigma}_{X|C}^2$, a positive-definite diagonal estimator of $\sigma_{X|C}^2$, identifies cases where our method provides advantages over NsDiff (Ye et al., 2025). In practice, $\Sigma_{X|C}$ rarely coincides with $I_{d_x}$ or $\sigma_{X|C}^2$, particularly in

time series data, where non-stationary dynamics (Li et al., 2024; Ye et al., 2025) and inter-variable dependencies (Yuan & Qiao, 2024) induce systematic departures from $I_{d_x}$ or $\sigma^2_{X|C}$.

## C.2 THE PROOF OF THEOREM 1

In this section, we demonstrate how to establish the sufficient conditions for $D_{\mathrm{KL}}(P_{X|C} \,\|\, \widehat{Q}) \leq D_{\mathrm{KL}}(P_{X|C} \,\|\, Q_0)$. This sufficient condition is fundamentally based on the following lemma.

**Lemma 1** *(Cardoso, 2003) Let $P_{X|C}$ be a conditional distribution of $X \in \mathbb{R}^{d_x}$ given $C$, with conditional mean $\mu_{X|C}$ and conditional covariance $\Sigma_{X|C}$. For any Gaussian distribution $Q = N(\mu, \Sigma)$, $D_{\mathrm{KL}}(P_{X|C} \,\|\, Q)$ is given by:*

$$D_{\mathrm{KL}}\left(P_{X|C} \,\|\, Q\right) = D_{\mathrm{KL}}\left(P_{X|C} \,\|\, N\left(\mu_{X|C}, \Sigma_{X|C}\right)\right) + D_{\mathrm{KL}}\left(N\left(\mu_{X|C}, \Sigma_{X|C}\right) \,\|\, Q\right). \tag{10}$$

This is a Pythagorean theorem for KLD. It tells us that the closest distribution to $P_{X|C}$ within the Gaussian family is $Q_* := N(\mu_{X|C}, \Sigma_{X|C})$. Note that $P_{X|C}$ is not necessarily Gaussian. This also lays the foundation for our subsequent theoretical analysis. With (10), we can rewrite $2\left[D_{\mathrm{KL}}(P_{X|C} \,\|\, \widehat{Q}) - D_{\mathrm{KL}}(P_{X|C} \,\|\, Q_0)\right]$ as:

$$2\left[D_{\mathrm{KL}}(P_{X|C} \,\|\, \widehat{Q}) - D_{\mathrm{KL}}(P_{X|C} \,\|\, Q_0)\right]$$

$$= 2\left[D_{\mathrm{KL}}(Q_* \,\|\, \widehat{Q}) - D_{\mathrm{KL}}(Q_* \,\|\, Q_0)\right]$$

$$= \left\| \widehat{\Sigma}_{X|C}^{-0.5}\left(\mu_{X|C} - \widehat{\mu}_{X|C}\right) \right\|_2^2 - \left\| \mu_{X|C} \right\|_2^2 \tag{a}$$

$$+ \log\left|\widehat{\Sigma}_{X|C}\right| + \mathrm{Tr}\left(\widehat{\Sigma}_{X|C}^{-1}\Sigma_{X|C}\right) - \mathrm{Tr}\left(\Sigma_{X|C}\right). \tag{b}$$

As a result, $2\left[D_{\mathrm{KL}}(P_{X|C} \,\|\, \widehat{Q}) - D_{\mathrm{KL}}(P_{X|C} \,\|\, Q_0)\right]$ is decomposed into two parts: (a) and (b). In the following, we bound (a) and (b) separately. First, for (a), we have:

$$\left\| \widehat{\Sigma}_{X|C}^{-0.5}\left(\mu_{X|C} - \widehat{\mu}_{X|C}\right) \right\|_2^2 - \left\| \mu_{X|C} \right\|_2^2$$

$$\leq \left\| \widehat{\Sigma}_{X|C}^{-0.5} \right\|_2^2 \left\| \mu_{X|C} - \widehat{\mu}_{X|C} \right\|_2^2 - \left\| \mu_{X|C} \right\|_2^2$$

$$\leq \left( \max_{i=1,\ldots,d_x} \{\widehat{\lambda}_{X|C,i}^{-0.5}\} \right)^2 \left\| \mu_{X|C} - \widehat{\mu}_{X|C} \right\|_2^2 - \left\| \mu_{X|C} \right\|_2^2$$

$$= \left( \min_{i=1,\ldots,d_x} \{\widehat{\lambda}_{X|C,i}\} \right)^{-1} \left\| \mu_{X|C} - \widehat{\mu}_{X|C} \right\|_2^2 - \left\| \mu_{X|C} \right\|_2^2.$$

Then, we derive an upper bound of (b):

$$\log\left|\widehat{\Sigma}_{X|C}\right| + \mathrm{Tr}\left(\widehat{\Sigma}_{X|C}^{-1}\Sigma_{X|C}\right) - \mathrm{Tr}\left(\Sigma_{X|C}\right)$$

$$= \log\left|\widehat{\Sigma}_{X|C}\right| + \mathrm{Tr}\left(I_{d_x}\right) - \mathrm{Tr}\left(I_{d_x}\right) + \mathrm{Tr}\left(\widehat{\Sigma}_{X|C}^{-1}\Sigma_{X|C}\right) - \mathrm{Tr}\left(\Sigma_{X|C}\right)$$

$$= \sum_{i=1}^{d_x}\left(1 + \log\widehat{\lambda}_{X|C,i}\right) - \mathrm{Tr}\left(\Sigma_{X|C}\right) + \mathrm{Tr}\left[\widehat{\Sigma}_{X|C}^{-1}\left(\Sigma_{X|C} - \widehat{\Sigma}_{X|C}\right)\right]$$

$$\leq \mathrm{Tr}\left(\widehat{\Sigma}_{X|C} - \Sigma_{X|C}\right) + \mathrm{Tr}\left[\widehat{\Sigma}_{X|C}^{-1}\left(\Sigma_{X|C} - \widehat{\Sigma}_{X|C}\right)\right]$$

$$\leq \mathrm{Tr}\left[I_{d_x}\left(\widehat{\Sigma}_{X|C} - \Sigma_{X|C}\right)\right] + \sum_{i=1}^{d_x}\widehat{\lambda}_{X|C,(d_x-i+1)}^{-1} \cdot \widetilde{s}_{(i)}$$

$$\leq \left\| I_{d_x} \right\|_F \cdot \left\| \widehat{\Sigma}_{X|C} - \Sigma_{X|C} \right\|_F + \max_{i=1,\ldots,d_x}\{\widehat{\lambda}_{X|C,i}^{-1}\}\sum_{i=1}^{d_x}\widetilde{s}_i$$

$$= \sqrt{d_x}\left\| \widehat{\Sigma}_{X|C} - \Sigma_{X|C} \right\|_F + \left(\min_{i=1,\ldots,d_x}\{\widehat{\lambda}_{X|C,i}\}\right)^{-1}\sum_{i=1}^{d_x}\widetilde{s}_i,$$

where the first inequality comes from the bound $1 + \log x \leq x$ for all $x > 0$, the second inequality applies Von Neumann's trace inequality (Mirsky, 1975), and the third inequality uses the inequality $\mathrm{Tr}\,(AB) \leq \|A\|_F \|B\|_F$, which holds for any multiplicable matrices $A$ and $B$.

Then, we derive:

$$2\left[ D_{\mathrm{KL}}(P_{X|C} \,\|\, \widehat{Q}) - D_{\mathrm{KL}}(P_{X|C} \,\|\, Q_0) \right] \tag{11}$$

$$\leq \left( \min_{i=1,\ldots,d_x} \{\widehat{\lambda}_{X|C,i}\} \right)^{-1} \left[ \|\mu_{X|C} - \widehat{\mu}_{X|C}\|_2^2 + \sum_{i=1}^{d_x} \widetilde{s}_i \right] + \sqrt{d_x} \left\| \Sigma_{X|C} - \widehat{\Sigma}_{X|C} \right\|_F - \|\mu_{X|C}\|_2^2,$$

which implies that as long as the right-hand side of (11) is non-positive (or equivalently, (3) is true), we can have:

$$D_{\mathrm{KL}}(P_{X|C} \,\|\, \widehat{Q}) \leq D_{\mathrm{KL}}(P_{X|C} \,\|\, Q_0).$$

$\square$

## C.3 THE PROOF OF THEOREM 2

By (10), we have

$$2\left[ D_{\mathrm{KL}}(P_{X|C} \,\|\, \widehat{Q}) - D_{\mathrm{KL}}(P_{X|C} \,\|\, N(\widehat{\mu}_{X|C}, \widetilde{M}_{X|C})) \right]$$

$$= 2\left[ D_{\mathrm{KL}}(Q_* \,\|\, \widehat{Q}) - D_{\mathrm{KL}}(Q_* \,\|\, N(\widehat{\mu}_{X|C}, \widetilde{M}_{X|C})) \right]$$

$$= \mathrm{Tr}\left( \widehat{\Sigma}_{X|C}^{-1} \Sigma_{X|C} - I_{d_x} \right) + (\mu_{X|C} - \widehat{\mu}_{X|C})^\top \widehat{\Sigma}_{X|C}^{-1} (\mu_{X|C} - \widehat{\mu}_{X|C})$$

$$\quad - \log\left| \widehat{\Sigma}_{X|C}^{-1} \Sigma_{X|C} \right| - \mathrm{Tr}\left( \left( \widetilde{M}_{X|C}^{-1} - M_{X|C}^{-1} \right) \Sigma_{X|C} \right)$$

$$\quad - \mathrm{Tr}\left( M_{X|C}^{-1} \Sigma_{X|C} - I_{d_x} \right) - (\mu_{X|C} - \widehat{\mu}_{X|C})^\top \widetilde{M}_{X|C}^{-1} (\mu_{X|C} - \widehat{\mu}_{X|C})$$

$$\quad + \log\left| \widetilde{M}_{X|C}^{-1} M_{X|C} \right| + \log\left| M_{X|C}^{-1} \Sigma_{X|C} \right|$$

$$\leq (\mu_{X|C} - \widehat{\mu}_{X|C})^\top \left( \widehat{\Sigma}_{X|C}^{-1} - \widetilde{M}_{X|C}^{-1} \right) (\mu_{X|C} - \widehat{\mu}_{X|C}) + 2B(\Sigma_{X|C}, \widehat{\Sigma}_{X|C})$$

$$\quad + \left\| \widetilde{M}_{X|C}^{-1} - M_{X|C}^{-1} \right\|_F \|\Sigma_{X|C} + M_{X|C}\|_F - 2B(\Sigma_{X|C}, M_{X|C})$$

$$\leq \|\mu_{X|C} - \widehat{\mu}_{X|C}\|_2^2 \left( \left\| \widehat{\Sigma}_{X|C}^{-1} - \Sigma_{X|C}^{-1} \right\|_2 + \left\| \Sigma_{X|C}^{-1} \right\|_2 + \left\| \widetilde{M}_{X|C}^{-1} \right\|_2 \right) + 2B(\Sigma_{X|C} \widehat{\Sigma}_{X|C})$$

$$\quad + d_x \left\| \widetilde{M}_{X|C}^{-1} - M_{X|C}^{-1} \right\|_2 \|\Sigma_{X|C} + M_{X|C}\|_2 - 2B(\Sigma_{X|C}, M_{X|C}),$$

where the first inequality follows from the matrix inequality $\log|M| \leq \mathrm{Tr}(M - I_{d_x})$, which holds when all eigenvalues of $M$ are positive real numbers. This implies that as long as (9) is true, we can have:

$$D_{\mathrm{KL}}(P_{X|C} \,\|\, \widehat{Q}) \leq D_{\mathrm{KL}}(P_{X|C} \,\|\, N(\widehat{\mu}_{X|C}, \widetilde{M}_{X|C})).$$

$\square$

## C.4 THE TERMINAL DISTRIBUTION OF (6)

In this section, we prove the terminal distribution of (6) is $\mathcal{N}(\widehat{\mu}_{\mathbf{X}|\mathbf{C}}, \widehat{\Sigma}_{\mathbf{X}_0|\mathbf{C}})$. First, let $\mathbf{Y}_\tau := \exp\{\int_0^\tau \beta_s ds/2\} \cdot (\mathbf{X}_\tau - \widehat{\mu}_{\mathbf{X}|\mathbf{C}})$, then we can derive:

$$d\mathbf{Y}_\tau = d(\mathbf{X}_\tau - \widehat{\mu}_{\mathbf{X}|\mathbf{C}}) \cdot e^{\int_0^\tau \beta_s ds/2} + (\mathbf{X}_\tau - \widehat{\mu}_{\mathbf{X}|\mathbf{C}}) \cdot de^{\int_0^\tau \beta_s ds/2}$$

$$= \left[ -\tfrac{1}{2}\beta_\tau (\mathbf{X}_\tau - \widehat{\mu}_{\mathbf{X}|\mathbf{C}})d\tau + \sqrt{\beta_\tau} \cdot \widehat{\Sigma}_{\mathbf{X}_0|\mathbf{C}}^{0.5} \circ d\mathbf{W}_\tau \right] \cdot e^{\int_0^\tau \beta_s ds/2}$$

$$\quad + (\mathbf{X}_\tau - \widehat{\mu}_{\mathbf{X}|\mathbf{C}}) \cdot e^{\int_0^\tau \beta_s ds/2} \cdot \tfrac{1}{2}\beta_\tau d\tau$$

$$= e^{\int_0^\tau \beta_s ds/2} \cdot \sqrt{\beta_\tau} \cdot \widehat{\Sigma}_{\mathbf{X}_0|\mathbf{C}}^{0.5} \circ d\mathbf{W}_\tau.$$

Integrate $d\mathbf{Y}_\tau$ from 0 to $\tau_1$ and we get:

$$\int_0^{\tau_1} d\mathbf{Y}_\tau = \mathbf{Y}_{\tau_1} - \mathbf{Y}_0 = \int_0^{\tau_1} e^{\int_0^\tau \beta_s ds/2} \cdot \sqrt{\beta_\tau} \cdot \widehat{\Sigma}_{\mathbf{X}_0|\mathbf{C}}^{0.5} \circ d\mathbf{W}_\tau.$$

Via the property of Ito integral, we can derive:

$$\mathbf{Y}_{\tau_1} - \mathbf{Y}_0 \sim \mathcal{N}\left(0, \int_0^{\tau_1} e^{\int_0^\tau \beta_s ds} \cdot \beta_\tau d\tau \cdot \widehat{\Sigma}_{\mathbf{X}_0|\mathbf{C}}\right),$$

or equivalently:

$$e^{\int_0^\tau \beta_s ds/2} \cdot (\mathbf{X}_\tau - \widehat{\mu}_{\mathbf{X}|\mathbf{C}}) \sim \mathcal{N}\left(\mathbf{X}_0 - \widehat{\mu}_{\mathbf{X}|\mathbf{C}}, (e^{\int_0^\tau \beta_s ds} - 1) \cdot \widehat{\Sigma}_{\mathbf{X}_0|\mathbf{C}}\right).$$

Finally, we can derive:

$$\mathbf{X}_\tau \sim \mathcal{N}\left(\widehat{\mu}_{\mathbf{X}|\mathbf{C}} + e^{-\int_0^\tau \beta_s ds/2}(\mathbf{X}_0 - \widehat{\mu}_{\mathbf{X}|\mathbf{C}}), (1 - e^{-\int_0^\tau \beta_s ds}) \cdot \widehat{\Sigma}_{\mathbf{X}_0|\mathbf{C}}\right).$$

Recall $\exp\{\int_0^\tau \beta_s ds\}$ becomes sufficiently large when $\tau \to 1$, then we can derive that the terminal distribution of $\mathbf{X}_\tau$ is $\mathcal{N}(\widehat{\mu}_{\mathbf{X}|\mathbf{C}}, \widehat{\Sigma}_{\mathbf{X}_0|\mathbf{C}})$.

## D   WHEN CAN INCORPORATING A PRIOR MODEL FAIL?

In this section, we discuss regimes in which Condition (3) may fail; in these regimes, incorporating a prior model may degrade performance. We also describe how we mitigate these risks.

First, if $\mu_{X|C} = 0$, then the right-hand side in (3), $\|\mu_{X|C}\|_2^2$, equals zero, while the left-hand side of (3) is dominated by estimation error. In this case, even small estimation errors can cause the inequality in (3) to fail. In practice, however, for non-stationary time series, $\mu_{X|C}$ often exhibits sharp variations and thus deviates from zero, so this scenario is unlikely to occur.

Second, When $\min_{i\in\{1,\dots,d_x\}} \widehat{\lambda}_{X|C,i}$ is very small, the factor $\left(\min_{i\in\{1,\dots,d_x\}} \widehat{\lambda}_{X|C,i}\right)^{-1}$ in (3) can blow up, so even modest deviations $\|\mu_{X|C} - \widehat{\mu}_{X|C}\|_2^2$ and $\|\Sigma_{X|C} - \widehat{\Sigma}_{X|C}\|_N$ may violate the condition. This motivates the explicit eigenvalue regularization in our JMCE loss in (4), which enforces strictly positive eigenvalues bounded away from zero.

Finally, Condition (3) explicitly involves the estimation errors $\|\mu_{X|C} - \widehat{\mu}_{X|C}\|_2^2$, $\|\Sigma_{X|C} - \widehat{\Sigma}_{X|C}\|_N$, and $\|\Sigma_{X|C} - \widehat{\Sigma}_{X|C}\|_F$. If $(\widehat{\mu}_{X|C}, \widehat{\Sigma}_{X|C})$ are not good estimators of $(\mu_{X|C}, \Sigma_{X|C})$, then these terms on the left-hand side of (3) become large and may easily exceed $\|\mu_{X|C}\|_2^2$ on the right-hand side. To mitigate this, we deliberately design JMCE as a joint conditional mean and covariance estimator whose loss directly mirrors the left-hand side of (3). The experiments in Section E.4 show that JMCE achieves small estimation error, providing empirical evidence that our estimator yields sufficiently accurate $(\widehat{\mu}_{X|C}, \widehat{\Sigma}_{X|C})$.

## E   EXTRA EXPERIMENTS

In this section, we first introduce the datasets and evaluation metrics, then report the performance of CW-Gen versus the Raw method on the ETTh2, ILI, Weather, and Solar Energy datasets in Tables 3–6. In addition, we present the ProbMSE and ProbMAE of our CW-Gen against the Raw models in Table 7 and 8. We also conduct ablation studies to show the effectiveness of our JMCE, as introduced in Section 3.2.

### E.1   DATASETS

We selected five widely used public real-world time series datasets for our experiments. ETT (Electricity Transformer Temperature) dataset (Zhou et al., 2021) contains hourly oil temperature and related external features (e.g., load, ambient temperature) collected from electricity transformers between July 2016 and July 2018. We use two subsets, (1) ETTh1 and (2) ETTh2, which cover

seven transformer-related factors. (3) ILI (Influenza-Like Illness): Collects the weekly proportion of patients with ILI among all patients, which is reported weekly by the Centers for Disease Control and Prevention of the United States from 2002 to 2021. (4) Weather: A meteorological time series dataset collected from 21 weather stations in Germany, containing meteorological variables such as temperature, humidity, and wind speed recorded every 10 minutes. (5) Solar Energy: Records solar power generation data from 137 photovoltaic plants in Alabama, sampled every 10 minutes during 2006. The basic statistical information of these datasets are summarized in Table 1.

### E.2 METRICS

We employ six metrics to evaluate probabilistic time series forecasting. Among them, CRPS (Matheson & Winkler, 1976) and QICE (Han et al., 2022) are widely used. Let $\mathbf{X}_{\text{gen},[k]} \in \mathbb{R}^{d \times T_f}$, $k = 1, \dots, K$ denote $K$ generated samples. Denote $\mathbf{X}_{\text{gen},[k]}^{i,t}$ and $\mathbf{X}_0^{i,t}$ as the $(i,t)$-th elements of $\mathbf{X}_{\text{gen},[k]}$ and $\mathbf{X}_0$, respectively, for $i = 1, \dots, d$ and $t = 1, \dots, T_f$. The defination of CRPS between the $K$ generated samples and $\mathbf{X}_0$ is given by:

$$\text{CRPS}(\{\mathbf{X}_{\text{gen},[k]}\}_{k=1}^K, \mathbf{X}_0) = \frac{1}{d \cdot T_f} \sum_{i=1}^d \sum_{t=1}^{T_f} \int_{\mathbb{R}} \left( \widehat{F}_{i,t}(z) - \mathbb{I}\{\mathbf{X}_0^{i,t} \le z\} \right)^2 dz,$$

where $\widehat{F}_{i,t}(z) := \frac{1}{K} \sum_{k=1}^K \mathbb{I}\{\mathbf{X}_{\text{gen},[k]}^{i,t} \le z\}$ and $\mathbb{I}\{\cdot\}$ is the indicator function.

To calculate QICE, we first construct $B$ equal quantile intervals from the generated samples (in our application, we choose $B = 10$). In the ideal case, each interval should contain exactly $1/B$ of the entries of $\mathbf{X}_0$. We then calculate the empirical frequency $r_b$ of $\mathbf{X}_0$'s entries falling into the $b$-th interval for $b = 1, \dots, B$. Finally, the QICE is calculated as:

$$\text{QICE} = \frac{1}{B} \sum_{b=1}^B \left| r_b - \frac{1}{B} \right|.$$

Correlation score (Ni et al., 2022) measures the discrepancy between the correlations among the $d$ dimensions of the generated and the true time series. The covariance between the $i$-th and $j$-th features of $\mathbf{X}_0$ is defined as:

$$\text{cov}_{i,j}(\mathbf{X}_0) = \frac{1}{T_f} \sum_{t=1}^{T_f} \mathbf{X}_0^{i,t} \mathbf{X}_0^{j,t} - \left( \frac{1}{T_f} \sum_{t=1}^{T_f} \mathbf{X}_0^{i,t} \right) \left( \frac{1}{T_f} \sum_{t=1}^{T_f} \mathbf{X}_0^{j,t} \right).$$

The Correlation score between $\mathbf{X}_0$ and $\mathbf{X}_{\text{gen},[k]}$ is defined as:

$$\text{Correlation score}(\mathbf{X}_{\text{gen},[k]}, \mathbf{X}_0) = \frac{1}{d^2} \sum_{i,j} \left| \frac{\text{cov}_{i,j}(\mathbf{X}_0)}{\text{cov}_{i,i}(\mathbf{X}_0)\text{cov}_{j,j}(\mathbf{X}_0)} - \frac{\text{cov}_{i,j}(\mathbf{X}_{\text{gen},[k]})}{\text{cov}_{i,i}(\mathbf{X}_{\text{gen},[k]})\text{cov}_{j,j}(\mathbf{X}_{\text{gen},[k]})} \right|.$$

The Probabilistic Correlation Score (ProbCorr) is defined as:

$$\text{ProbCorr}(\{\mathbf{X}_{\text{gen},[k]}\}_{k=1}^K, \mathbf{X}_0) = \frac{1}{K} \sum_{k=1}^K \text{Correlation score}(\mathbf{X}_{\text{gen},[k]}, \mathbf{X}_0).$$

ProbCorr measures the discrepancy between the correlation structure of each generated sample $\mathbf{X}_{\text{gen},[k]}$ and the ground-truth $\mathbf{X}_0$.

Nevertheless, it is important to recognize that CRPS, QICE, and ProbCorr do not effectively capture temporal dependencies between the generated and true sequences. To address this, a TS2Vec model (Yue et al., 2022) is trained on the real sequence $[\mathbf{C}, \mathbf{X}_0]$ and subsequently used to extract latent representations for $[\mathbf{C}, \mathbf{X}_0]$ and $[\mathbf{C}, \mathbf{X}_{\text{gen},[k]}]$, $k = 1, \dots, K$. The Fréchet Inception Distance (FID) computed between these representations is referred to as the conditional FID. Since TS2Vec employs a dedicated network architecture to jointly capture temporal patterns and feature correlations, conditional FID provides a more comprehensive assessment of generative quality.

In addition to probabilistic metrics, we also employ ProbMSE and ProbMAE to evaluate forecasting performance. ProbMSE and ProbMAE are used as point forecast metrics, and their definition is given by:

$$\text{ProbMSE}(\{\mathbf{X}_{\text{gen},[k]}\}_{k=1}^{K}, \mathbf{X}_0) = \frac{1}{d \cdot T_f} \sum_{i=1}^{d} \sum_{t=1}^{T_f} \left[ \frac{1}{K} \sum_{k=1}^{K} \left( \mathbf{X}_{\text{gen},[k]}^{i,t} \right) - \mathbf{X}_0^{i,t} \right]^2,$$

$$\text{ProbMAE}(\{\mathbf{X}_{\text{gen},[k]}\}_{k=1}^{K}, \mathbf{X}_0) = \frac{1}{d \cdot T_f} \sum_{i=1}^{d} \sum_{t=1}^{T_f} \left| \frac{1}{K} \sum_{k=1}^{K} \left( \mathbf{X}_{\text{gen},[k]}^{i,t} \right) - \mathbf{X}_0^{i,t} \right|.$$

Traditional MSE and MAE measure the discrepancy between the mean of the generated samples and the true time series. In contrast, ProbMSE and ProbMAE differ from these traditional metrics by taking into account the MSE and MAE between each individual generated sample and the true time series. Consequently, ProbMSE and ProbMAE provide a more stringent evaluation than standard MSE and MAE.

### E.3 CW-GEN ON MORE REAL DATASETS

Tables 3, 4, 5, and 6 present the results of different methods on the ETTh2, ILI, Weather, and Solar Energy datasets across four metrics (CRPS, QICE, ProbCorr, and Conditional FID). In time series forecasting tasks, ProbMSE and ProbMAE reflect the accuracy of point estimates. We report the evaluation results on these two metrics for the five real-world datasets in Tables 7 and 8, respectively.

From Table 3 to 8, we can observe that CW-Gen achieves advantages on the majority of probabilistic metrics, even on high-dimensional datasets such as Solar Energy. For point forecasting metrics, CW-Gen outperforms the baselines on all datasets except Solar Energy.

Table 3: Metrics for models trained on original ETTh2 (Raw) and conditionally whitened ETTh2 (CW). Each experiment is repeated by 10 times, and standard deviations are provided in brackets. The better results between Raw and CW are underlined. The win rates of every metric of Raw and CW-Gen models are also provided.

| Model | CRPS ($\downarrow$) | | QICE ($\downarrow$) | | ProbCorr ($\downarrow$) | | Conditional FID ($\downarrow$) | |
| (ETTh2) | Raw | CW | Raw | CW | Raw | CW | Raw | CW |
|---|---|---|---|---|---|---|---|---|
| TimeDiff | 2.543 | 0.395 | 12.769 | 6.584 | 0.753 | 0.327 | 211.67 | 4.495 |
| (2023) | (0.910) | (0.031) | (1.726) | (1.246) | (0.252) | (0.020) | (55.976) | (0.699) |
| SSSD | 0.754 | 0.458 | 14.698 | 6.637 | 0.525 | 0.417 | 187.29 | 14.780 |
| (2023) | (0.260) | (0.111) | (0.955) | (3.059) | (0.040) | (0.039) | (147.33) | (7.330) |
| Diffusion | 1.107 | 0.381 | 8.605 | 4.147 | 0.691 | 0.438 | 99.509 | 15.383 |
| -TS (2024) | (0.077) | (0.024) | (0.792) | (1.677) | (0.022) | (0.061) | (64.135) | (16.112) |
| TMDM | 0.421 | 0.377 | 4.500 | 3.945 | 0.378 | 0.313 | 9.528 | 4.107 |
| (2024) | (0.043) | (0.000) | (0.689) | (1.475) | (0.027) | (0.001) | (1.779) | (0.249) |
| NsDiff | 0.370 | 0.369 | 2.334 | 2.579 | 0.323 | 0.351 | 19.957 | 14.842 |
| (2025) | (0.027) | (0.014) | (0.040) | (0.345) | (0.026) | (0.018) | (5.029) | (2.783) |
| FlowTS | 1.534 | 0.824 | 12.147 | 11.744 | 0.650 | 0.498 | 80.540 | 10.640 |
| (2025) | (0.252) | (0.138) | (1.356) | (1.094) | (0.044) | (0.050) | (69.867) | (12.883) |
| Win rate | 0.0% | 100.0% | 16.7% | 83.3% | 16.7% | 83.3% | 0.0% | 100.0% |

### E.4 ABLATION STUDY FOR JMCE

In this subsection, we examine the impact of the two hyperparameters $w_{\text{Eigen}}$ and $\lambda_{\min}$ in (4) on estimation accuracy of JMCE. Beside, we also investigate the influence different backbones on JMCE. While the main text employs the Non-stationary Transformer (Liu et al., 2022), in this section we adopt FED Former (Zhou et al., 2022) and Informer (Zhou et al., 2021) as the backbones of JMCE and compare their performance.

Table 4: Metrics for models trained on original ILI (Raw) and conditionally whitened ILI (CW). Each experiment is repeated by 10 times, and standard deviations are provided in brackets. The better results between Raw and CW are underlined. The win rates of every metric of Raw and CW-Gen models are also provided.

| Model (ILI) | CRPS ($\downarrow$) | | QICE ($\downarrow$) | | ProbCorr ($\downarrow$) | | Conditional FID ($\downarrow$) | |
|---|---|---|---|---|---|---|---|---|
| | Raw | CW | Raw | CW | Raw | CW | Raw | CW |
| TimeDiff (2023) | 1.148 (0.134) | 1.046 (0.081) | 15.015 (0.430) | 13.597 (1.550) | 0.455 (0.016) | 0.399 (0.048) | 10.957 (3.483) | 6.845 (0.813) |
| SSSD (2023) | 1.038 (0.126) | 0.758 (0.110) | 15.063 (1.802) | 9.115 (2.030) | 0.374 (0.071) | 0.365 (0.038) | 6.416 (0.335) | 5.964 (1.895) |
| Diffusion -TS (2024) | 1.222 (0.271) | 0.769 (0.168) | 6.588 (2.479) | 8.883 (1.840) | 0.381 (0.038) | 0.373 (0.058) | 20.513 (27.582) | 5.969 (1.215) |
| TMDM (2024) | 0.796 (0.045) | 0.722 (0.025) | 6.706 (0.821) | 8.029 (2.734) | 0.365 (0.020) | 0.359 (0.000) | 22.693 (12.420) | 12.234 (18.767) |
| NsDiff (2025) | 0.738 (0.047) | 0.645 (0.059) | 5.930 (0.867) | 6.173 (0.970) | 0.352 (0.011) | 0.307 (0.058) | 73.379 (23.257) | 14.852 (3.843) |
| FlowTS (2025) | 0.997 (0.055) | 0.851 (0.068) | 9.771 (0.728) | 10.645 (0.778) | 0.413 (0.010) | 0.410 (0.021) | 7.689 (1.098) | 6.202 (0.536) |
| Win rate | 0.0% | 100.0% | 66.7% | 33.3% | 0.0% | 100.0% | 0.0% | 100.0% |

Table 5: Metrics for models trained on original Weather (Raw) and conditionally whitened Weather (CW). Each experiment is repeated by 10 times, and standard deviations are provided in brackets. The better results between Raw and CW are underlined. The win rates of every metric of Raw and CW-Gen models are also provided.

| Model (Weather) | CRPS ($\downarrow$) | | QICE ($\downarrow$) | | ProbCorr ($\downarrow$) | | Conditional FID ($\downarrow$) | |
|---|---|---|---|---|---|---|---|---|
| | Raw | CW | Raw | CW | Raw | CW | Raw | CW |
| TimeDiff (2023) | 0.531 (0.032) | 0.258 (0.014) | 8.530 (0.693) | 6.772 (1.869) | 0.362 (0.010) | 0.255 (0.006) | 9.673 (3.095) | 6.892 (1.183) |
| SSSD (2023) | 0.499 (0.145) | 0.530 (0.186) | 7.428 (0.790) | 4.121 (3.457) | 0.438 (0.012) | 0.411 (0.040) | 914.81 (260.690) | 330.31 (315.18) |
| Diffusion -TS (2024) | 0.495 (0.114) | 0.319 (0.033) | 3.957 (7.789) | 3.047 (1.049) | 0.503 (0.033) | 0.414 (0.047) | 278.60 (566.85) | 90.739 (59.186) |
| TMDM (2024) | 0.231 (0.003) | 0.254 (0.016) | 3.468 (0.412) | 3.127 (0.733) | 0.264 (0.008) | 0.247 (0.010) | 6.978 (0.836) | 5.941 (0.860) |
| NsDiff (2025) | 0.270 (0.003) | 0.262 (0.009) | 3.746 (0.201) | 3.536 (0.408) | 0.274 (0.007) | 0.266 (0.007) | 18.034 (0.887) | 9.870 (3.936) |
| FlowTS (2025) | 0.348 (0.043) | 0.244 (0.017) | 6.901 (1.616) | 6.598 (0.371) | 0.334 (0.035) | 0.262 (0.011) | 8.447 (1.540) | 6.948 (2.887) |
| Win rate | 33.3% | 66.7% | 0.0% | 100.0% | 0.0% | 100.0% | 0.0% | 100.0% |

We adopt $\mathcal{L}_2, \mathcal{L}_F, \mathcal{L}_{\text{SVD}}$ in (4) as the metrics. In addition, we also compute the left-hand side (LHS) of (3) as a metric, whose formulation is given by:

$$\text{LHS} = \big( \min_{i,t} \widehat{\lambda}_{\widehat{\Sigma}_{\mathbf{x}_0,t|\mathbf{c},i}} \big)^{-1} \cdot (\mathcal{L}_2 + \mathcal{L}_{\text{SVD}}) + \sqrt{d \cdot T_f} \mathcal{L}_F. \tag{12}$$

Tables 9 and 10 report JMCE results under varying $w_{\text{Eigen}}$ and $\lambda_{\min}$, respectively, and Table 21 presents CW-Gen results with separately trained models and different backbones.

Table 6: Metrics for models trained on original Solar Energy (Raw) and conditionally whitened Solar Energy (CW). Each experiment is repeated by 10 times, and standard deviations are provided in brackets. The better results between Raw and CW are underlined. The win rates of every metric of Raw and CW-Gen models are also provided.

| Model | CRPS ($\downarrow$) | | QICE ($\downarrow$) | | ProbCorr ($\downarrow$) | | Conditional FID ($\downarrow$) | |
|---|---|---|---|---|---|---|---|---|
| (Solar) | Raw | CW | Raw | CW | Raw | CW | Raw | CW |
| TimeDiff | 0.746 | 0.299 | 15.361 | 11.998 | 0.198 | 0.212 | 5.606 | 4.397 |
| (2023) | (0.017) | (0.016) | (0.617) | (1.383) | (0.004) | (0.007) | (0.357) | (0.232) |
| SSSD | 0.350 | 0.555 | 13.435 | 9.111 | 0.330 | 0.307 | 14.915 | 14.165 |
| (2023) | (0.042) | (0.135) | (1.966) | (0.675) | (0.020) | (0.081) | (0.622) | (4.571) |
| Diffusion | 0.349 | 0.289 | 2.857 | 2.843 | 0.229 | 0.221 | 5.796 | 5.007 |
| -TS (2024) | (0.030) | (0.026) | (1.326) | (1.186) | (0.008) | (0.008) | (0.665) | (0.902) |
| TMDM | 0.376 | 0.371 | 10.033 | 7.162 | 0.509 | 0.201 | 248.80 | 8.279 |
| (2024) | (0.004) | (0.018) | (0.076) | (0.214) | (0.008) | (0.012) | (16.384) | (2.528) |
| NsDiff | 0.304 | 0.359 | 6.861 | 4.251 | 0.366 | 0.239 | 106.83 | 7.417 |
| (2025) | (0.008) | (0.007) | (0.318) | (0.345) | (0.011) | (0.015) | (8.575) | (1.051) |
| FlowTS | 0.276 | 0.226 | 6.791 | 4.207 | 0.284 | 0.210 | 28.464 | 3.225 |
| (2025) | (0.029) | (0.018) | (0.687) | (1.162) | (0.025) | (0.011) | (5.609) | (0.157) |
| Win rate | 33.3% | 66.7% | 0.0% | 100.0% | 16.7% | 83.3% | 0.0% | 100.0% |

Table 7: ProbMSE for Raw and CW-Gen models. Each experiment is repeated by 10 times, and standard deviations are provided in brackets. The better results between Raw and CW are underlined. The win rates for all datasets are also provided.

| Model | Variant | ETTh1 | ETTh2 | ILI | Weather | Solar |
|---|---|---|---|---|---|---|
| TimeDiff | Raw | 1.366(0.080) | 0.793(0.983) | 3.803(2.062) | 0.803(0.062) | 0.800(0.020) |
| (2023) | CW | 0.756(0.135) | 0.496(0.064) | 2.913(0.303) | 0.267(0.014) | 0.264(0.022) |
| SSSD | Raw | 1.493(0.390) | 2.132(0.824) | 2.953(0.419) | 2.785(3.243) | 0.349(0.079) |
| (2023) | CW | 0.908(0.219) | 0.643(0.262) | 2.169(0.467) | 2.158(2.761) | 1.053(0.508) |
| Diffusion | Raw | 1.177(0.094) | 2.053(1.078) | 2.224(0.497) | 1.287(0.322) | 0.391(0.029) |
| -TS(2024) | CW | 0.717(0.094) | 0.503(0.071) | 2.788(0.658) | 0.345(0.085) | 0.326(0.045) |
| TMDM | Raw | 0.767(0.070) | 0.615(0.118) | 2.417(0.189) | 0.249(0.007) | 0.243(0.014) |
| (2024) | CW | 0.681(0.010) | 0.488(0.001) | 1.984(0.113) | 0.284(0.024) | 0.418(0.065) |
| NsDiff | Raw | 0.637(0.075) | 0.649(0.040) | 2.424(0.163) | 0.283(0.008) | 0.277(0.021) |
| (2025) | CW | 0.729(0.132) | 0.488(0.041) | 1.759(0.324) | 0.292(0.012) | 0.507(0.030) |
| FlowTS | Raw | 1.006(0.153) | 2.958(0.774) | 2.960(0.250) | 0.455(0.086) | 0.262(0.065) |
| (2025) | CW | 0.698(0.059) | 1.522(0.429) | 2.369(0.254) | 0.272(0.029) | 0.240(0.024) |
| Win Rate | | 83.33% | 83.33% | 100.0% | 66.67% | 50.00% |

Table 9 shows that as $w_{\text{Eigen}}$ increases, the smallest eigenvalue moves further away from zero, which aligns with the intended purpose of this parameter. Surprisingly, the estimation of both the conditional mean and the sliding-window covariance also becomes more accurate with larger $w_{\text{Eigen}}$.

Table 10 shows that as $\lambda_{\min}$ increases, the smallest eigenvalue moves further away from zero. A larger $\lambda_{\min}$ leads to poorer estimation of the sliding-window covariance, because the features of real-world time series are typically highly correlated and thus the sliding-window covariances have very small minimum eigenvalues. Penalizing the eigenvalues with a larger $\lambda_{\min}$ alters the structure of the estimation.

Table 21 shows that separately training the estimator of the conditional mean and that of the sliding-window covariance does not effectively control the smallest eigenvalue, although this training strat-

Table 8: ProbMAE for Raw and CW-Gen models. Each experiment is repeated by 10 times, and standard deviations are provided in brackets. The better results between Raw and CW are underlined. The win rates for all datasets are also provided.

| Model | Variant | ETTh1 | ETTh2 | ILI | Weather | Solar |
|---|---|---|---|---|---|---|
| TimeDiff | Raw | 0.899(0.040) | 0.480(0.055) | 1.025(0.436) | 0.670(0.034) | 0.800(0.021) |
| (2023) | CW | 0.581(0.025) | 0.489(0.033) | 1.121(0.076) | 0.307(0.009) | 0.331(0.019) |
| SSSD | Raw | 0.959(0.150) | 1.512(0.140) | 1.201(0.090) | 1.190(0.533) | 0.294(0.038) |
| (2023) | CW | 0.704(0.069) | 0.581(0.110) | 0.941(0.157) | 0.755(0.209) | 0.685(0.164) |
| Diffusion | Raw | 0.801(0.037) | 1.891(0.537) | 1.018(0.135) | 0.884(0.125) | 0.451(0.028) |
| -TS (2024) | CW | 0.594(0.031) | 0.508(0.045) | 1.178(0.222) | 0.381(0.066) | 0.390(0.039) |
| TMDM | Raw | 0.627(0.039) | 0.551(0.054) | 0.990(0.044) | 0.294(0.003) | 0.303(0.006) |
| (2024) | CW | 0.503(0.047) | 0.484(0.000) | 0.899(0.011) | 0.325(0.020) | 0.430(0.021) |
| NsDiff | Raw | 0.557(0.032) | 0.544(0.016) | 1.005(0.063) | 0.325(0.005) | 0.345(0.013) |
| (2025) | CW | 0.553(0.022) | 0.481(0.019) | 0.812(0.068) | 0.330(0.009) | 0.458(0.011) |
| FlowTS | Raw | 0.742(0.079) | 1.019(0.144) | 1.039(0.084) | 0.447(0.056) | 0.324(0.031) |
| (2025) | CW | 0.598(0.021) | 0.946(0.142) | 0.961(0.066) | 0.303(0.022) | 0.274(0.023) |
| Win Rate | | 100.00% | 83.33% | 100.0% | 66.67% | 50.00% |

egy offers a slight advantage in estimating the conditional mean. Different backbones also lead to different outcomes. FED-Former performs well in estimating the conditional mean but is slightly less effective in estimating the sliding-window covariance. In contrast, Informer achieves strong performance in covariance estimation and eigenvalue control, yet performs the worst in estimating the conditional mean.

In addition, we provide a comparison between the learning targets and the outputs of JMCE in Figure 4. As shown in the figure, JMCE is able to accurately predict both the future time series and most components of the sliding-window covariance on training and test sets. For some diagonal components of the sliding-window covariance (such as Cov Dim 8, 19, 26, and 28), JMCE intelligently enlarges these values, which helps prevent the minimum eigenvalue from becoming too small.

Table 9: Metrics for JMCE trained on ETTh1, with different $w_{\text{Eigen}}$. The $\lambda_{\min}$ is set to 0.1. Each experiment is repeated 10 times and standard deviations are provided in brackets. The defination of LHS can be found in (12).

| Model | $\mathcal{L}_2 (\downarrow)$ | $\mathcal{L}_F (\downarrow)$ | $\mathcal{L}_{\text{SVD}} (\downarrow)$ | $\left(\min_{t,i} \widehat{\lambda}_{\widetilde{\Sigma}_{\mathbf{X}_0,t\vert\mathbf{C}},i}\right)^{-1} (\downarrow)$ | LHS $(\downarrow)$ |
|---|---|---|---|---|---|
| JMCE | 0.702 | 0.198 | 0.493 | $2.2 \cdot 10^7$ | $2.4 \cdot 10^7$ |
| ($w_{\text{Eigen}} = 0$) | (0.006) | (0.000) | (0.000) | $(4.4 \cdot 10^{15})$ | $(5.5 \cdot 10^{15})$ |
| JMCE | 0.672 | 0.186 | 0.526 | 14.472 | 24.197 |
| ($w_{\text{Eigen}} = 10$) | (0.002) | (0.000) | (0.000) | (4.540) | (6.258) |
| JMCE | 0.683 | 0.189 | 0.534 | 12.996 | 22.819 |
| ($w_{\text{Eigen}} = 20$) | (0.005) | (0.000) | (0.000) | (1.438) | (4.422) |
| JMCE | 0.693 | 0.184 | 0.529 | 12.175 | 21.651 |
| ($w_{\text{Eigen}} = 40$) | (0.005) | (0.000) | (0.000) | (6.609) | (13.283) |
| JMCE | 0.726 | 0.180 | 0.528 | 11.487 | 21.037 |
| ($w_{\text{Eigen}} = 50$) | (0.001) | (0.000) | (0.000) | (3.286) | (5.085) |

## E.5 INFLUENCE OF DIFFERENT JMCEs ON CW-GEN

In this subsection, we investigate the impact of different JMCE models on CW-Gen. We first train the JMCE with different hyperparameters, JMCE with different backbones, or separately trained

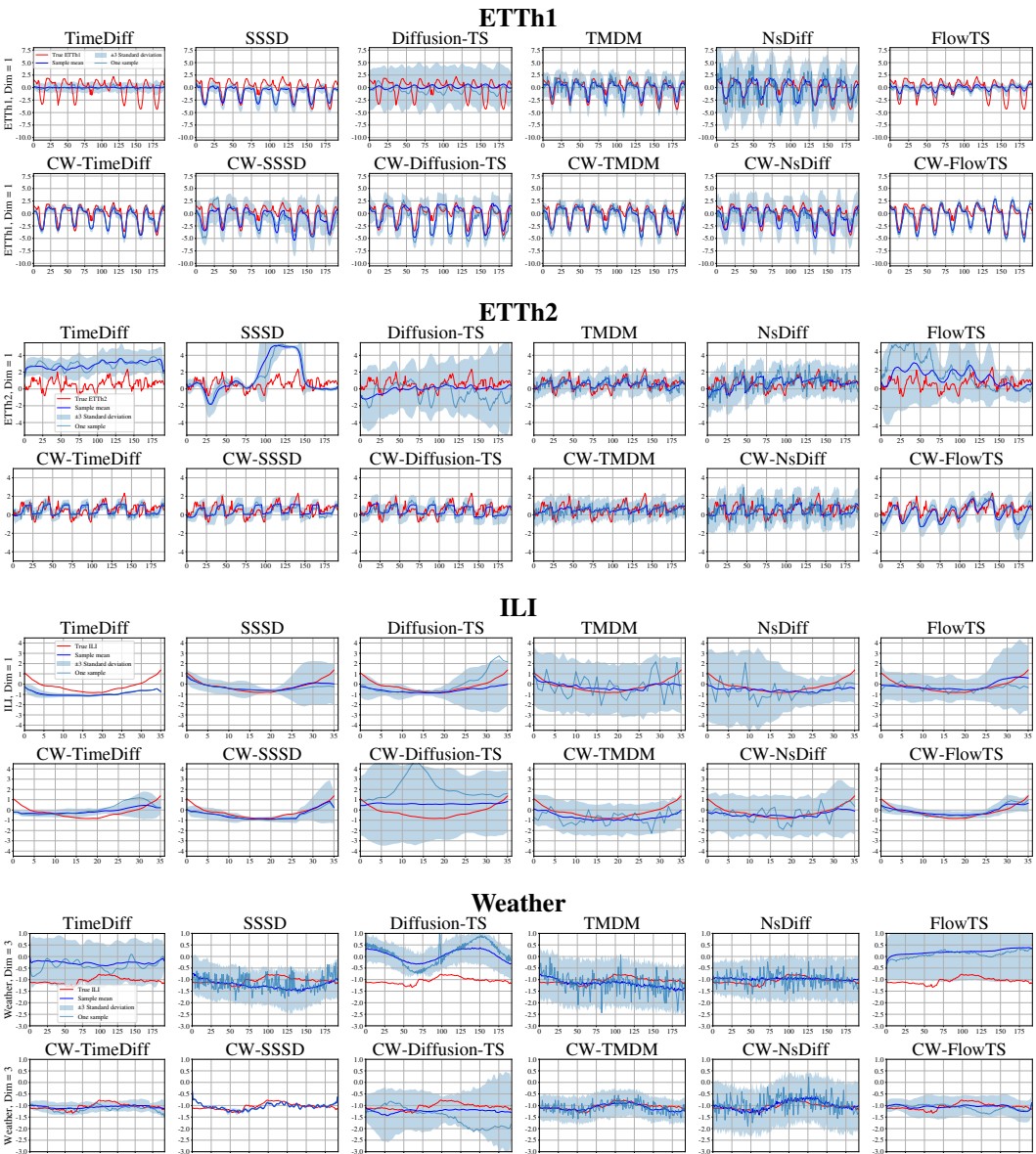

Figure 3: Comparison of all models on ETTh1, ETTh2, ILI and Weather.

mean and covariance models. Then, these models are severed as the prior for the CW-Gen models. The CW-Gen models are evaluated by the same metrics as in Section 5.

Table 12 and Table 13 indicate that the default parameters in our paper ($w_{\text{Eigen}} = 50, \lambda_{\min} = 0.1$) achieve slight advantages over other parameter combinations. Table 21 shows that CW-Gen models using different JMCEs exhibit slight variations, but all CW-Gen models with JMCE priors outperform those with separately trained prior models in most cases. Among the three backbones, the Non-stationary Transformer achieves the best performance on 13 metrics, while FED-Former achieves 8 and Informer achieves 1. Therefore, we adopt the Non-stationary Transformer as the backbone of JMCE.

### E.6 CONTRIBUTIONS OF INDIVIDUAL COMPONENTS OF THE PRIOR MODEL

In this section, we investigate how different components of JMCE contribute to the improvement. Specifically, we use only the conditional mean (Mean), only the diagonal elements of the condi-

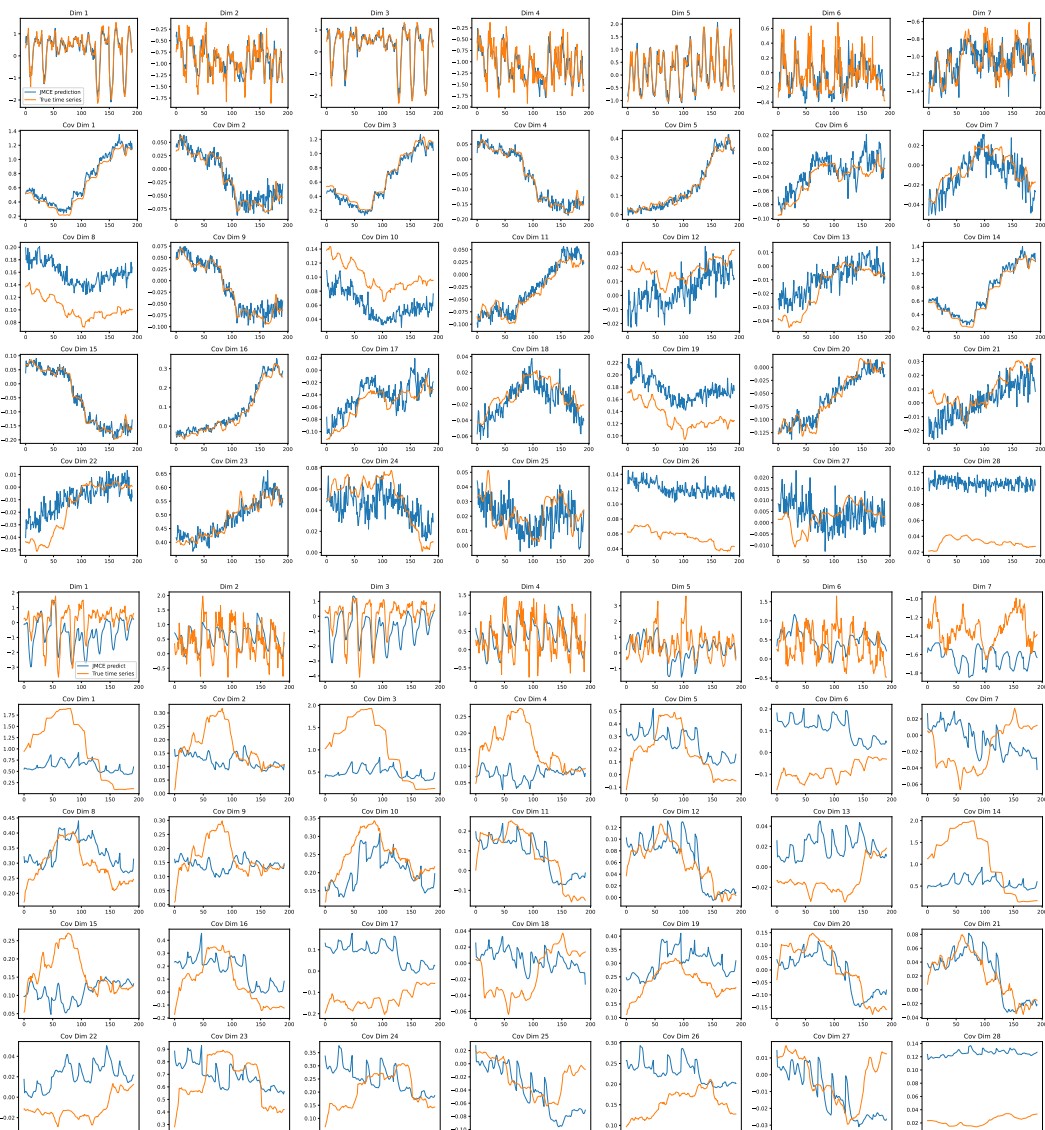

Figure 4: Comparison between the learning targets and the predictions of JMCE (top: training set, bottom: test set).

tional covariance (Var), only the full conditional covariance (Cov), as well as the combination of the conditional mean and the conditional variance (Mean & Var) as prior information, and then evaluate the performance of CW-Gen. Our default CW-Gen adopts the conditional mean together with the full conditional covariance (Mean & Cov) as the prior.

According to Table 14, we can generally conclude that using only the conditional mean for centering yields slightly inferior performance, compared to whitening using the conditional mean together with conditional variance or covariance. This indicates that learning the conditional variance or covariance of the time series provides beneficial prior information for the generative model. Moreover, if we compare the performance of CW-Gen when using the conditional mean together with the conditional variance versus using the conditional mean together with the conditional covariance, we observe that in most cases CW-Gen performs better with the full conditional covariance. This suggests that incorporating the full conditional covariance, rather than only the variance, provides a stronger and more informative prior for the generative model. In addition, using only the conditional variance or only the full conditional covariance as the prior degrades the performance of CW-Gen.

Table 10: Metrics for JMCE trained on ETTh1, with different $\lambda_{\min}$. The $w_{\text{Eigen}}$ is set to 50. Each experiment is repeated 10 times and standard deviations are provided in brackets. The defination of LHS can be found in (12).

| Model | $\mathcal{L}_2$ ($\downarrow$) | $\mathcal{L}_F$ ($\downarrow$) | $\mathcal{L}_{\text{SVD}}$ ($\downarrow$) | $\left(\min_{t,i} \widehat{\lambda}_{\widetilde{\Sigma}_{\mathbf{X}_0,t\mid\mathbf{C}},i}\right)^{-1}$ ($\downarrow$) | LHS ($\downarrow$) |
|---|---|---|---|---|---|
| JMCE ($\lambda_{\min} = 10^{-3}$) | 0.665 (0.002) | 0.193 (0.000) | 0.484 (0.000) | $2.3 \cdot 10^3$ ($1.2 \cdot 10^6$) | $2.7 \cdot 10^3$ ($1.6 \cdot 10^6$) |
| JMCE ($\lambda_{\min} = 10^{-2}$) | 0.662 (0.007) | 0.195 (0.000) | 0.493 (0.000) | 157.69 ($5.4 \cdot 10^3$) | 190.60 ($8.0 \cdot 10^3$) |
| JMCE ($\lambda_{\min} = 0.05$) | 0.680 (0.004) | 0.195 (0.000) | 0.509 (0.000) | 23.573 (3.423) | 35.213 (7.417) |
| JMCE ($\lambda_{\min} = 0.1$) | 0.726 (0.001) | 0.180 (0.000) | 0.528 (0.000) | 11.487 (3.286) | 21.037 (5.085) |

Table 11: Metrics for JMCE separately or jointly trained on ETTh1, with different backbones. The $\lambda_{\min}$ is set to 0.1 and $w_{\text{Eigen}}$ is set to 50. Each experiment is repeated 10 times and standard deviations are provided in brackets. The defination of LHS can be found in (12). NS, FED, and IN indicate that the backbone of JMCE is the Non-stationary Transformer, FED-Former, and Informer, respectively.

| Model | $\mathcal{L}_2$ ($\downarrow$) | $\mathcal{L}_F$ ($\downarrow$) | $\mathcal{L}_{\text{SVD}}$ ($\downarrow$) | $\left(\min_{t,i} \widehat{\lambda}_{\widetilde{\Sigma}_{\mathbf{X}_0,t\mid\mathbf{C}},i}\right)^{-1}$ ($\downarrow$) | LHS ($\downarrow$) |
|---|---|---|---|---|---|
| Separate (NS) | 0.721 (0.100) | 0.421 (0.164) | 0.797 (0.078) | 1196 (764) | 1824 (1184) |
| JMCE (NS) | 0.726 (0.001) | 0.180 (0.000) | 0.528 (0.000) | 11.487 (3.286) | 21.037 (5.085) |
| JMCE (FED) | 0.548 (0.026) | 0.387 (0.047) | 0.741 (0.030) | 43.381 (12.833) | 70.206 (16.962) |
| JMCE (IN) | 1.224 (0.039) | 0.260 (0.015) | 0.609 (0.016) | 10.481 (1.605) | 28.821 (2.956) |

However, the latter achieves a lower ProbCorr than the former, indicating that leveraging the full conditional covariance makes CW-Gen better capture the inter-variable dependencies.

### E.7 ABLATION STUDY OF THE LENGTH OF SLIDING WINDOW

In this section, we compare the effect of the length of the sliding window. In our main experiments, we set the length of sliding window as 95, following NsDiff (Ye et al., 2025). In ETTh1 dataset, we compare the performance of CW-Gen under four additional window lengths, namely 75, 85, 105, and 115. In ILI dataset, we compare two additional window lengths, namely 11 and 19.

From Table 15, we observe that on ETTh1, the sliding window length does not introduce substantial changes to the performance of CW-Gen. In contrast, Table 16 shows that the sliding window has a somewhat larger impact on the ILI dataset. This is likely because the dataset is relatively short, and changes in the sliding window length may alter the underlying dependence relationships.

### E.8 CW-GEN COMPARED WITH OTHER UNIVARIATE PRIOR METHODS

We discuss the similarities and differences between CW-Gen and other univariate generative models that incorporate prior information, like DSPD (Biloš et al., 2023) and TsFlow (Kollovieh et al., 2025).

DSPD leverages kernel functions such as $\exp(-\gamma|t_i - t_j|)$ and $\exp(-\gamma(t_i - t_j)^2), \gamma > 0$, to help the diffusion model better capture the temporal correlations restricted to the prediction window. How-

Table 12: Metrics for CW-Gen models with different $w_{\text{Eigen}}$ of JMCE. Each experiment is repeated by 10 times, and standard deviations are provided in brackets. The better results between Raw and CW are underlined. The win rates of every metric for different $w_{\text{Eigen}}$ are also provided.

| Model $w_{\text{Eigen}}$ | CRPS ($\downarrow$) | | QICE ($\downarrow$) | | ProbCorr ($\downarrow$) | | Conditional FID ($\downarrow$) | |
|---|---|---|---|---|---|---|---|---|
| | 40 | 50 | 40 | 50 | 40 | 50 | 40 | 50 |
| TimeDiff (2023) | 0.495 (0.038) | 0.505 (0.040) | 8.069 (2.310) | 8.821 (1.916) | 0.235 (0.035) | 0.243 (0.027) | 6.835 (7.952) | 6.788 (5.425) |
| SSSD (2023) | 0.510 (0.099) | 0.524 (0.085) | 4.935 (2.544) | 4.838 (1.921) | 0.239 (0.026) | 0.238 (0.024) | 7.438 (2.538) | 9.265 (5.003) |
| Diffusion -TS (2024) | 0.447 (0.014) | 0.445 (0.024) | 2.333 (0.831) | 2.963 (0.887) | 0.276 (0.027) | 0.266 (0.012) | 11.913 (9.911) | 7.686 (2.751) |
| TMDM (2024) | 0.443 (0.000) | 0.440 (0.001) | 4.131 (1.128) | 4.555 (0.855) | 0.209 (0.000) | 0.213 (0.001) | 3.554 (0.283) | 3.831 (0.431) |
| NsDiff (2025) | 0.422 (0.020) | 0.431 (0.029) | 1.264 (0.252) | 1.249 (0.228) | 0.200 (0.014) | 0.206 (0.010) | 8.160 (1.185) | 8.820 (1.541) |
| FlowTS (2025) | 0.491 (0.033) | 0.488 (0.020) | 9.014 (0.313) | 8.817 (0.460) | 0.261 (0.020) | 0.254 (0.021) | 5.030 (0.871) | 4.865 (0.563) |
| Win rate | 50.0% | 50.0% | 33.3% | 66.7% | 50.0% | 50.0% | 50.0% | 50.0% |

Table 13: Metrics for CW-Gen models with different $\lambda_{\text{min}}$ of JMCE. Each experiment is repeated by 10 times, and standard deviations are provided in brackets. The better results between Raw and CW are underlined. The win rates of every metric for different $\lambda_{\text{min}}$ are also provided.

| Model $\lambda_{\text{min}}$ | CRPS ($\downarrow$) | | QICE ($\downarrow$) | | ProbCorr ($\downarrow$) | | Conditional FID ($\downarrow$) | |
|---|---|---|---|---|---|---|---|---|
| | 0.05 | 0.1 | 0.05 | 0.1 | 0.05 | 0.1 | 0.05 | 0.1 |
| TimeDiff (2023) | 0.508 (0.035) | 0.505 (0.040) | 9.101 (1.753) | 8.821 (1.916) | 0.230 (0.027) | 0.243 (0.027) | 5.527 (3.326) | 6.788 (5.425) |
| SSSD (2023) | 0.530 (0.104) | 0.524 (0.085) | 5.166 (2.257) | 4.838 (1.921) | 0.246 (0.038) | 0.238 (0.024) | 9.637 (4.358) | 9.265 (5.003) |
| Diffusion -TS (2024) | 0.453 (0.024) | 0.445 (0.024) | 2.760 (1.093) | 2.963 (0.887) | 0.271 (0.031) | 0.266 (0.012) | 10.553 (5.914) | 7.686 (2.751) |
| TMDM (2024) | 0.446 (0.001) | 0.440 (0.001) | 4.260 (0.785) | 4.555 (0.855) | 0.216 (0.001) | 0.213 (0.001) | 3.702 (0.475) | 3.831 (0.431) |
| NsDiff (2025) | 0.416 (0.030) | 0.431 (0.029) | 1.369 (0.256) | 1.249 (0.228) | 0.199 (0.021) | 0.206 (0.010) | 8.477 (1.934) | 8.820 (1.541) |
| FlowTS (2025) | 0.494 (0.038) | 0.488 (0.020) | 8.969 (0.557) | 8.817 (0.460) | 0.257 (0.026) | 0.254 (0.021) | 5.131 (0.717) | 4.865 (0.563) |
| Win rate | 16.7% | 83.3% | 33.3% | 66.7% | 33.3% | 66.7% | 50.0% | 50.0% |

ever, such prior information does not incorporate the historical time series and conditional mean. Therefore, DSPD does not provide stronger guidance for forecasting then JMCE. Our JMCE can explicitly capture the correlations between variables by directly learning the sliding-window covariance on the prediction window. Moreover, the architecture of the Non-stationary Transformer enables JMCE to capture temporal correlations within the prediction window via masked self-attention. It also captures correlations between the observed series and the prediction window through cross-attention (Liu et al., 2022).

TsFlow employs Gaussian processes (GPs) to predict the conditional mean and variance within the prediction window. However, GPs rely heavily on the choice of kernel functions, and modeling non-stationary processes typically requires carefully designed kernels or kernels with varying length

Table 14: Metrics for models trained on ETTh1 conditionally whitened by different priors. Each experiment is repeated by 10 times, and standard deviations are provided in brackets. The best results are underlined and the second-best results are dashed-underlined.

| Model | Prior | CRPS | QICE | ProbCorr | Conditional FID |
|---|---|---|---|---|---|
| TimeDiff | Mean | 0.503(0.037) | 8.001(1.449) | 0.245(0.034) | 8.173(9.390) |
| (2023) | Var | 0.581(0.045) | 9.153(1.232) | 0.235(0.014) | 7.385(2.153) |
| | Cov | 0.624(0.053) | 9.133(1.557) | 0.258(0.049) | 25.858(50.988) |
| | Mean & Var | 0.512(0.032) | 9.883(2.368) | 0.246(0.023) | 6.749(7.313) |
| | Mean & Cov | 0.505(0.040) | 8.821(1.916) | 0.243(0.027) | 6.788(5.415) |
| SSSD | Mean | 0.554(0.116) | 7.175(2.386) | 0.242(0.028) | 6.431(2.537) |
| (2023) | Var | 0.566(0.080) | 5.738(1.911) | 0.349(0.043) | 18.689(11.482) |
| | Cov | 0.587(0.067) | 5.190(1.410) | 0.340(0.037) | 18.903(13.821) |
| | Mean & Var | 0.530(0.087) | 5.203(1.840) | 0.237(0.015) | 7.235(2.265) |
| | Mean & Cov | 0.524(0.085) | 4.838(1.921) | 0.238(0.024) | 9.265(5.003) |
| Diffusion | Mean | 0.468(0.035) | 2.544(0.897) | 0.301(0.027) | 8.535(2.208) |
| -TS (2024) | Var | 0.536(0.064) | 4.616(0.990) | 0.474(0.045) | 68.466(43.107) |
| | Cov | 0.549(0.032) | 5.360(1.176) | 0.502(0.046) | 95.769(50.600) |
| | Mean & Var | 0.465(0.027) | 2.539(1.182) | 0.288(0.026) | 8.558(3.331) |
| | Mean & Cov | 0.445(0.024) | 2.963(0.887) | 0.266(0.012) | 7.686(2.751) |
| TMDM | Mean | 0.446(0.000) | 3.760(1.201) | 0.215(0.000) | 4.687(1.447) |
| (2024) | Var | 0.644(0.002) | 7.372(1.263) | 0.291(0.000) | 14.107(9.783) |
| | Cov | 0.676(0.001) | 6.377(1.237) | 0.303(0.000) | 31.748(190.543) |
| | Mean & Var | 0.440(0.000) | 4.952(0.911) | 0.213(0.000) | 3.840(0.459) |
| | Mean & Cov | 0.440(0.001) | 4.555(0.855) | 0.213(0.001) | 3.831(0.431) |
| NsDiff | Mean | 0.455(0.023) | 4.647(0.500) | 0.238(0.006) | 71.683(9.262) |
| (2025) | Var | 0.623(0.011) | 4.132(0.450) | 0.209(0.011) | 23.738(3.654) |
| | Cov | 0.409(0.022) | 1.857(0.290) | 0.199(0.013) | 11.540(2.200) |
| | Mean & Var | 0.408(0.024) | 1.447(0.259) | 0.209(0.012) | 21.288(3.456) |
| | Mean & Cov | 0.431(0.029) | 1.249(0.228) | 0.206(0.010) | 8.820(1.541) |
| FlowTS | Mean | 0.477(0.024) | 8.778(0.798) | 0.231(0.010) | 4.850(0.485) |
| (2025) | Var | 0.653(0.021) | 8.328(0.701) | 0.316(0.010) | 10.459(2.206) |
| | Cov | 0.643(0.020) | 8.064(0.634) | 0.315(0.005) | 10.344(2.642) |
| | Mean & Var | 0.489(0.022) | 9.240(0.477) | 0.260(0.027) | 5.090(0.250) |
| | Mean & Cov | 0.488(0.743) | 8.817(0.460) | 0.254(0.021) | 4.865(0.563) |

scales, making the approach less straightforward in practice. Moreover, computing the GP mean and variance requires inverting a matrix whose size equals the length of the historical observations. In our setting, the GP requires inverting a matrix in $\mathbb{R}^{T_h \times T_h}$, which incurs a computational complexity of $\mathcal{O}(T_h^3)$ which is higher than our JMCE when $T_h > d$.

In addition, we apply DSPD and TsFlow to each individual dimension of ETTh1 and generate samples accordingly. We then aggregate the generated samples and evaluate them using the four metrics introduced in Appendix E.2. In Table 17, we report the performance of DSPD and TsFlow. Compared with CW-TimeDiff and CW-SSSD from the same year (2023), DSPD exhibits worse CRPS and a high QICE. ProbCorr of DSPD is also lower than both models, while its Conditional FID lies at an intermediate level. Besides, compared with CW-NsDiff and CW-FlowTS proposed in 2025, TsFlow shows worse CRPS, QICE, and ProbCorr, while its Conditional FID lies between the two. Their performance on the first dimension of ETTh1 is further illustrated in Figure 5. As observed, DSPD lacks the prior information provided by the conditional mean and therefore fails to effectively capture highly nonlinear patterns. On the other hand, TsFlow produces results that are less stable near the end of the prediction window, indicating that its effectiveness is limited in long-term forecasting settings.

Table 15: Metrics for models trained on ETTh1, with different length of sliding window. Each experiment is repeated by 10 times, and standard deviations are provided in brackets.

| Model (ETTh1) | Window | CRPS | QICE | ProbCorr | Conditional FID |
|---|---|---|---|---|---|
| TimeDiff (2023) | 75 | 0.477(0.030) | 8.039(2.242) | 0.232(0.025) | 4.858(1.156) |
| | 85 | 0.511(0.036) | 9.249(1.610) | 0.249(0.044) | 6.236(3.831) |
| | 95 | 0.505(0.040) | 8.821(1.916) | 0.243(0.027) | 6.788(5.415) |
| | 105 | 0.518(0.044) | 8.548(1.539) | 0.249(0.021) | 7.780(5.611) |
| | 115 | 0.502(0.043) | 7.621(1.925) | 0.251(0.037) | 4.750(1.318) |
| SSSD (2023) | 75 | 0.508(0.072) | 4.525(2.263) | 0.235(0.024) | 7.531(2.885) |
| | 85 | 0.520(0.070) | 4.484(2.086) | 0.235(0.021) | 8.236(2.754) |
| | 95 | 0.524(0.085) | 4.838(1.921) | 0.238(0.024) | 9.265(5.003) |
| | 105 | 0.540(0.140) | 4.383(2.391) | 0.248(0.018) | 7.858(2.501) |
| | 115 | 0.519(0.069) | 4.392(2.062) | 0.249(0.021) | 9.189(2.733) |
| Diffusion -TS (2024) | 75 | 0.425(0.012) | 2.439(0.767) | 0.248(0.023) | 10.527(11.100) |
| | 85 | 0.431(0.020) | 3.084(1.712) | 0.247(0.022) | 11.319(11.244) |
| | 95 | 0.445(0.024) | 2.963(0.887) | 0.266(0.012) | 7.686(2.751) |
| | 105 | 0.452(0.030) | 2.876(1.388) | 0.256(0.022) | 7.639(3.423) |
| | 115 | 0.450(0.024) | 2.473(1.110) | 0.268(0.029) | 9.613(5.999) |
| TMDM (2024) | 75 | 0.429(0.000) | 4.307(0.593) | 0.213(0.000) | 3.789(0.031) |
| | 85 | 0.435(0.000) | 4.398(0.760) | 0.212(0.000) | 3.622(0.059) |
| | 95 | 0.440(0.001) | 4.555(0.855) | 0.213(0.001) | 3.831(0.431) |
| | 105 | 0.464(0.001) | 4.655(1.288) | 0.231(0.000) | 4.036(0.168) |
| | 115 | 0.442(0.000) | 3.981(0.475) | 0.225(0.001) | 3.631(0.280) |
| NsDiff (2025) | 75 | 0.429(0.023) | 1.210(0.266) | 0.203(0.013) | 9.846(3.176) |
| | 85 | 0.423(0.021) | 1.193(0.203) | 0.203(0.009) | 9.025(1.011) |
| | 95 | 0.431(0.029) | 1.249(0.228) | 0.206(0.010) | 8.820(1.541) |
| | 105 | 0.422(0.020) | 1.281(0.212) | 0.207(0.011) | 9.707(1.354) |
| | 115 | 0.432(0.024) | 1.376(0.228) | 0.213(0.022) | 10.030(2.669) |
| FlowTS (2025) | 75 | 0.484(0.023) | 8.965(0.388) | 0.253(0.011) | 4.886(0.426) |
| | 85 | 0.476(0.015) | 8.865(0.340) | 0.251(0.015) | 5.016(0.637) |
| | 95 | 0.488(0.020) | 8.817(0.460) | 0.254(0.021) | 4.865(0.563) |
| | 105 | 0.483(0.024) | 8.894(0.704) | 0.256(0.010) | 4.920(0.364) |
| | 115 | 0.475(0.020) | 8.924(0.583) | 0.253(0.015) | 4.813(0.456) |

### E.9 ACCELERATING CW-GEN

Training a CW-Gen model consists of three steps: (1) training the JMCE model, (2) conditionally whitening the time series using the trained JMCE model, and (3) training the generative model by the conditionally whitened time series.

In step (1), the training algorithm of JMCE involves SVD and eigen-decomposition, both of which have a computational complexity of $O(d^3)$. Although these operations can be efficiently parallelized on GPUs, they still pose challenges when implementing the model on high-dimensional time-series datasets. Under high-dimensional cases, one possible approach is to omit the computation of $\mathcal{L}_{\text{SVD}}$ and the penalty term $\mathcal{R}_{\lambda_{\min}}$ in (4). In this case, to ensure that the minimum eigenvalue remains bounded away from zero, we can add $\lambda_{\min} \cdot I_d$ either to JMCE's output $\widehat{L}_{t|\mathbf{C}}$ or to $\widehat{\Sigma}_{\mathbf{X}_0,t|\mathbf{C}}$. Another possible approach is to only learn the diagonal part of the sliding-window covariance. However, a diagonal covariance matrix cannot approximate a general covariance matrix well in terms of the nuclear norm. Besides, diagonal covariance parameterizations lose the ability to control the minimum eigenvalue of the conditional covariance matrix, and therefore the theoretical foundations of JMCE no longer apply. Empirical evidence in Table 14 also indicates only using conditional variance leads to inferior performance, especially for ProbCorr.

Table 16: Metrics for models trained on ILI, with different length of sliding window. Each experiment is repeated by 10 times, and standard deviations are provided in brackets.

| Model (ILI) | Window | CRPS | QICE | ProbCorr | Conditional FID |
|---|---|---|---|---|---|
| TimeDiff (2023) | 11 | 0.668(0.980) | 8.341(2.465) | 0.314(0.027) | 5.715(1.571) |
| | 15 | 1.046(0.081) | 13.597(1.550) | 0.399(0.048) | 6.845(0.813) |
| | 19 | 0.786(0.120) | 10.722(5.149) | 0.345(0.064) | 5.162(0.701) |
| SSSD (2023) | 11 | 0.792(0.140) | 10.807(1.848) | 0.376(0.045) | 5.282(0.675) |
| | 15 | 0.758(0.110) | 9.115(2.030) | 0.365(0.038) | 5.964(1.895) |
| | 19 | 0.785(0.113) | 9.228(1.892) | 0.376(0.040) | 5.452(1.367) |
| Diffusion -TS (2024) | 11 | 0.873(0.150) | 4.230(1.403) | 0.336(0.037) | 9.619(2.963) |
| | 15 | 0.769(0.168) | 8.883(1.840) | 0.373(0.058) | 5.969(1.215) |
| | 19 | 0.894(0.100) | 5.504(1.291) | 0.327(0.013) | 8.325(2.539) |
| TMDM (2024) | 11 | 0.689(0.013) | 7.081(2.059) | 0.342(0.002) | 6.567(2.991) |
| | 15 | 0.722(0.025) | 8.029(2.734) | 0.359(0.000) | 12.234(18.767) |
| | 19 | 0.717(0.008) | 7.297(1.610) | 0.355(0.000) | 7.954(5.131) |
| NsDiff (2025) | 11 | 0.646(0.086) | 5.802(1.316) | 0.344(0.034) | 12.466(5.457) |
| | 15 | 0.645(0.059) | 6.173(0.970) | 0.307(0.058) | 14.852(3.843) |
| | 19 | 0.706(0.045) | 6.268(1.186) | 0.364(0.029) | 12.375(3.949) |
| FlowTS (2025) | 11 | 0.898(0.106) | 11.338(0.764) | 0.425(0.069) | 6.713(0.998) |
| | 15 | 0.851(0.068) | 10.645(0.778) | 0.410(0.021) | 6.202(0.536) |
| | 19 | 0.838(0.053) | 10.721(0.865) | 0.409(0.017) | 6.391(0.951) |

Table 17: Metrics for DSPD and TsFlow trained on ETTh1. Each experiment is repeated by 10 times, and standard deviations are provided in brackets.

| Model (ETTh1) | CRPS | QICE | ProbCorr | Conditional FID |
|---|---|---|---|---|
| DSPD (2023) | 0.741(0.090) | 11.032(0.816) | 0.288(0.039) | 10.828(9.544) |
| TsFlow (2025) | 0.568(0.040) | 7.968(1.018) | 0.257(0.026) | 18.596(12.677) |

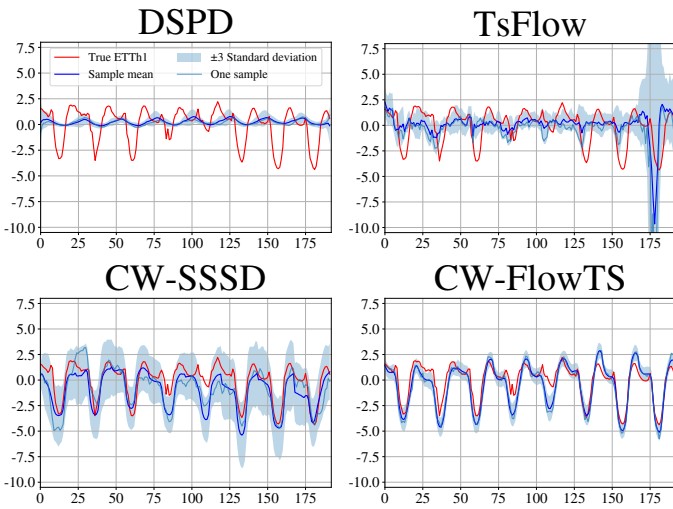

Figure 5: Comparison of DSPD, CW-SSSD, TsFlow and CW-FlowTS on the first dimension of ETTh1.

In step (2) and (3), we by default compute $\widehat{\Sigma}_{\mathbf{X}_0|\mathbf{C}}^k$, $k = \pm 0.5$ using eigen-decomposition. However, this step can in fact be avoided. Recall that the output of JMCE includes the conditional mean

$\widehat{\mu}_{\mathbf{X}|\mathbf{C}}$ and a lower-triangular matrix $\widehat{L}_{t|\mathbf{C}}$ for $t = 1, \ldots, T_f$. By directly computing the inverse of $\widehat{L}_{t|\mathbf{C}}$, we can whiten the time-series data without performing eigen-decomposition. We illustrate the idea by the following simple case. Suppose $X \in \mathbb{R}^d$ is a random variable with covariance $\mathrm{Cov}(X) = \Sigma = LL^\top$, where $L$ is a lower-triangular matrix. Then it is straightforward to verify that

$$\mathrm{Cov}(L^{-1}X) = L^{-1}\Sigma L^{-1\top} = L^{-1}LL^\top L^{\top^{-1}} = I_d.$$

Thus, we can replace $(\widehat{L}_{t|\mathbf{C}}\widehat{L}_{t|\mathbf{C}}^\top)^{-0.5}$ in line 3 of Algorithm 2 with $\widehat{L}_{t|\mathbf{C}}^{-1}$, and similarly replace $(\widehat{L}_{1|\mathbf{C}}\widehat{L}_{1|\mathbf{C}}^\top)^{0.5}$ in line 3 of Algorithm 3 with $\widehat{L}_{t|\mathbf{C}}$. This transforms the eigen-decomposition step in the original algorithm into computing the inverse of a lower-triangular matrix. Since the inverse of a lower-triangular matrix can be obtained efficiently using forward substitution, this modification yields a substantial speedup compared to performing eigen-decomposition (Strang, 2022).

Moreover, on Weather and Solar Energy datasets, we verify that this substitution substantially reduces the computational cost, with the exact reduction reported in Table 19.

### E.10 CW-GEN IN AN END2END FASHION

In the Section 3 and 4, our training pipeline first trains JMCE, then conditionally whitens the time series data, and finally trains the generative model. However, with the accelerated algorithm introduced in Appendix E.9, we are able to train JMCE and the generative model jointly in an end-to-end (E2E) fashion, which further improves training efficiency.

In Table 18, we compare the setting without prior information (Raw), CW-Gen trained in the default manner (CW), and CW-Gen trained in an E2E fashion (CW-E2E). The results show that CW-E2E generally improves both ProbCorr and Conditional FID, while its QICE is slightly inferior to that of CW. The algorithm of training CW-E2E can be found in Algorithm 6 and 7. We also carefully compared the training time of CW-Gen and CW-E2E in Table 19.

Table 18: Metrics for models trained on ETTh1, including those trained on raw data (Raw), the default CW-Gen pipeline (CW), and the end-to-end CW-Gen variant (CW-E2E). Each experiment is repeated 10 times, and standard deviations are provided in brackets. The best results are underlined and the second-best results are dashed-underlined.

| Model (ETTh1) | Variant | CRPS | QICE | ProbCorr | Conditional FID |
|---|---|---|---|---|---|
| TimeDiff (2023) | Raw | 0.787(0.051) | 9.038(0.946) | 0.320(0.012) | 19.008(6.088) |
| | CW | 0.505(0.040) | 8.821(1.916) | 0.243(0.027) | 6.788(5.425) |
| | CW-E2E | 0.514(0.039) | 9.189(1.189) | 0.218(0.016) | 4.305(0.547) |
| SSSD (2023) | Raw | 0.836(0.153) | 11.624(1.312) | 0.326(0.032) | 40.887(17.601) |
| | CW | 0.524(0.085) | 4.838(1.921) | 0.238(0.024) | 9.265(5.003) |
| | CW-E2E | 0.489(0.054) | 6.254(1.612) | 0.229(0.017) | 6.908(3.625) |
| Diffusion -TS (2024) | Raw | 0.626(0.027) | 3.002(0.838) | 0.401(0.017) | 81.563(60.905) |
| | CW | 0.445(0.024) | 2.963(0.887) | 0.266(0.012) | 7.686(2.751) |
| | CW-E2E | 0.474(0.031) | 5.536(1.514) | 0.271(0.014) | 5.105(1.126) |
| TMDM (2024) | Raw | 0.472(0.031) | 3.360(1.055) | 0.230(0.014) | 9.931(4.439) |
| | CW | 0.440(0.001) | 4.555(0.855) | 0.213(0.001) | 3.831(0.431) |
| | CW-E2E | 0.433(0.027) | 2.368(0.247) | 0.205(0.010) | 11.654(1.757) |
| NsDiff (2025) | Raw | 0.407(0.032) | 1.792(0.682) | 0.214(0.014) | 35.261(7.785) |
| | CW | 0.431(0.029) | 1.249(0.228) | 0.206(0.010) | 8.820(1.541) |
| | CW-E2E | 0.412(0.010) | 1.484(0.479) | 0.195(0.006) | 7.827(1.264) |
| FlowTS (2025) | Raw | 0.724(0.135) | 8.820(2.631) | 0.354(0.060) | 39.793(24.853) |
| | CW | 0.488(0.020) | 8.817(0.460) | 0.254(0.021) | 4.865(0.563) |
| | CW-E2E | 0.481(0.023) | 10.277(0.455) | 0.220(0.017) | 4.040(0.741) |

---

**Algorithm 6** Training JMCE and CW-Diff in an end-to-end fashion

---

**Input**: $(\mathbf{X}_0, \mathbf{C})$ in training set, hyperparameters $\lambda_{\min}, w_{\text{Eigen}}$, diffusion schedule $\beta_\tau, \tau \in [0, 1]$.
**Output**: A trained JMCE model $\text{JMCE}(\cdot)$ and a trained neural network $s_\theta^{\text{CW}}$.

1: Calculate sliding-window covariances $\widetilde{\Sigma}_{\mathbf{X}_0, 1}, \ldots, \widetilde{\Sigma}_{\mathbf{X}_0, T_f}$ of $\mathbf{X}_0$
2: Initialize a non-autoregressive model $\text{JMCE}(\cdot)$ and neural network of diffusion model $s_\theta^{\text{CW}}$
3: **while** not converge **do**
4:     Calculate $\widehat{\mu}_{\mathbf{X}|\mathbf{C}}, \widehat{L}_{1|\mathbf{C}}, \ldots, \widehat{L}_{T_f|\mathbf{C}} = \text{JMCE}(\mathbf{C})$
5:     **for** $t = 1, \ldots, T_f$ **do**
6:         Let $\widehat{\Sigma}_{\mathbf{X}_0, t|\mathbf{C}} = \widehat{L}_{t|\mathbf{C}} \widehat{L}_{t|\mathbf{C}}^\top$
7:         Perform eigen-decomposition of $\widehat{\Sigma}_{\mathbf{X}_0, t|\mathbf{C}}$ and obtain eigenvalues $\widehat{\lambda}_{\widehat{\Sigma}_{\mathbf{X}_0, t|\mathbf{C}}, i}, i = 1, \ldots, d$
8:         Perform singular value decomposition (SVD) of $\widetilde{\Sigma}_{\mathbf{X}_0, t} - \widehat{\Sigma}_{\mathbf{X}_0, t|\mathbf{C}}$ and obtain singular values $\widetilde{s}_{i,t}, i = 1, \ldots, d$
9:     **end for**
10:     Calculate $L_2 = \left\| \mathbf{X}_0 - \widehat{\mu}_{\mathbf{X}|\mathbf{C}} \right\|^2$, $L_F = \sum_{t=1}^{T_f} \left\| \widetilde{\Sigma}_{\mathbf{X}_0, t} - \widehat{\Sigma}_{\mathbf{X}_0, t|\mathbf{C}} \right\|_F$, $L_{\text{SVD}} = \sum_{t=1}^{T_f} \sum_{i=1}^d \widetilde{s}_{i,t}$,
11:     $R_{\lambda_{\min}} = \sum_{t=1}^{T_f} \sum_{i=1}^d \text{ReLU}(\lambda_{\min} - \widehat{\lambda}_{\widehat{\Sigma}_{\mathbf{X}_0, t|\mathbf{C}}, i})$
12:     Calculate $L_{\text{JMCE}} = L_2 + L_{\text{SVD}} + \lambda_{\min} \sqrt{d \cdot T_f} L_F + w_{\text{Eigen}} R_{\lambda_{\min}}$
13:     Calculate $\widehat{L}_{\mathbf{C}}^{-1} = [\widehat{L}_{1|\mathbf{C}}^{-1}, \ldots, \widehat{L}_{T_f|\mathbf{C}}^{-1}]$
14:     Calculate $\mathbf{X}_0^{\text{CW}} = \widehat{L}_{\mathbf{C}}^{-1} \circ (\mathbf{X}_0 - \widehat{\mu}_{\mathbf{X}|\mathbf{C}})$
15:     Draw $\tau \sim U(0, 1]$
16:     Draw $\epsilon \sim \mathcal{N}(0, I_{d \times d \times T_f})$
17:     Calculate $\alpha_\tau = \exp\left\{ -\int_0^\tau \beta_s ds/2 \right\}$ and $\sigma_\tau^2 = 1 - \alpha_\tau^2$
18:     Calculate $L_{\text{Diff}} = \| s_\theta^{\text{CW}} \left( \alpha_\tau \mathbf{X}_0^{\text{CW}} + \sigma_\tau \epsilon, \mathbf{C}, \tau \right) + \epsilon/\sigma_\tau \|^2$
19:     Calculate $L_{\text{E2E}} = L_{\text{JMCE}} + L_{\text{Diff}}$
20:     Calculate $\nabla L_{\text{E2E}}$ and update the parameters of $\text{JMCE}(\cdot)$ and $s_\theta^{\text{CW}}$
21: **end while**
22: **return** $\text{JMCE}(\cdot)$, $s_\theta^{\text{CW}}$

---

## F   IMPLEMENTATION DETAILS

The evaluation setup, including the history length, prediction horizon, sliding window covariance, and the basic configuration of the JMCE loss, has been described in Section 5. In this section, we provide the detailed training parameters and implementation specifics for the proposed JMCE and the baseline methods.

For our JMCE model, except for the Solar Energy dataset, the backbone is a Non-stationary Transformer with a model dimension of $d_{\text{model}} = 512$, 8 attention heads, 2 encoder layers, 1 decoder layer, a dropout rate of 0.1, and a feedforward layer dimension of 1024. For the Solar Energy dataset, $d_{\text{model}}$ is set to 128 and the number of encoder layers is increased to 3. Training is performed using the AdamW optimizer (Loshchilov & Hutter, 2019) with a learning rate of $1 \times 10^{-4}$, a weight decay of $5 \times 10^{-4}$, a batch size of 64, and 20 epochs. We select the model with the lowest loss over 20 epochs as the final model.

For the baseline methods: TimeDiff uses the default parameters in (Shen & Kwok, 2023). SSSD uses the default parameters of SSSD$^{\text{SA}}$ in (Alcaraz & Strodthoff, 2023). Diffusion-TS uses the parameters for ETTh in (Yuan & Qiao, 2024). TMDM and NsDiff follow (Li et al., 2024; Ye et al., 2025), with minor modifications to their own mean & variance estimators. FlowTS uses the parameters reported in (Hu et al., 2025). Except for TMDM and NsDiff, all other methods are trained using the AdamW optimizer (Loshchilov & Hutter, 2019) with a learning rate of $1 \times 10^{-3}$, a weight decay of $5 \times 10^{-4}$, a batch size of 128, and 20 epochs. We select the model with the lowest loss over 20 epochs as the final model.

All of the experiments are conducted on a single NVIDIA A6000, with a memory of 48GB.

---

**Algorithm 7** Training JMCE and CW-Flow in an end-to-end fashion

---

**Input**: $(\mathbf{X}_0, \mathbf{C})$ in training set, hyperparameters $\lambda_{\min}, w_{\text{Eigen}}$.
**Output**: A trained JMCE model $\text{JMCE}(\cdot)$ and a trained neural network $v_\psi^{\text{CW}}$.

1: Calculate sliding-window covariances $\widetilde{\Sigma}_{\mathbf{X}_0, 1}, \ldots, \widetilde{\Sigma}_{\mathbf{X}_0, T_f}$ of $\mathbf{X}_0$
2: Initialize a non-autoregressive model $\text{JMCE}(\cdot)$ and neural network of flow matching $v_\psi^{\text{CW}}$
3: **while** not converge **do**
4:     Calculate $\widehat{\mu}_{\mathbf{X}|\mathbf{C}}, \widehat{L}_{1|\mathbf{C}}, \ldots, \widehat{L}_{T_f|\mathbf{C}} = \text{JMCE}(\mathbf{C})$
5:     **for** $t = 1, \ldots, T_f$ **do**
6:         Let $\widehat{\Sigma}_{\mathbf{X}_0, t|\mathbf{C}} = \widehat{L}_{t|\mathbf{C}} \widehat{L}_{t|\mathbf{C}}^\top$
7:         Perform eigen-decomposition of $\widehat{\Sigma}_{\mathbf{X}_0, t|\mathbf{C}}$ and obtain eigenvalues $\widehat{\lambda}_{\widehat{\Sigma}_{\mathbf{X}_0, t|\mathbf{C}}, i}, i = 1, \ldots, d$
8:         Perform singular value decomposition (SVD) of $\widetilde{\Sigma}_{\mathbf{X}_0, t} - \widehat{\Sigma}_{\mathbf{X}_0, t|\mathbf{C}}$ and obtain singular values $\widetilde{s}_{i,t}, i = 1, \ldots, d$
9:     **end for**
10:    Calculate $L_2 = \left\| \mathbf{X}_0 - \widehat{\mu}_{\mathbf{X}|\mathbf{C}} \right\|^2, L_F = \sum_{t=1}^{T_f} \left\| \widetilde{\Sigma}_{\mathbf{X}_0, t} - \widehat{\Sigma}_{\mathbf{X}_0, t|\mathbf{C}} \right\|_F, L_{\text{SVD}} = \sum_{t=1}^{T_f} \sum_{i=1}^{d} \widetilde{s}_{i,t},$
11:    $R_{\lambda_{\min}} = \sum_{t=1}^{T_f} \sum_{i=1}^{d} \text{ReLU}(\lambda_{\min} - \widehat{\lambda}_{\widehat{\Sigma}_{\mathbf{X}_0, t|\mathbf{C}}, i})$
12:    Calculate $L_{\text{JMCE}} = L_2 + L_{\text{SVD}} + \lambda_{\min} \sqrt{d \cdot T_f} L_F + w_{\text{Eigen}} R_{\lambda_{\min}}$
13:    Let $\widehat{L}_{\mathbf{C}} = [\widehat{L}_{1|\mathbf{C}}, \ldots, \widehat{L}_{T_f|\mathbf{C}}]$
14:    Draw $\tau \sim U(0, 1)$
15:    Draw $\epsilon^{\text{CW}} \sim \mathcal{N}(0, I_{d \times d \times T_f})$
16:    Calculate $\epsilon^{\text{CW}} = \widehat{L}_{\mathbf{C}} \circ \epsilon^{\text{CW}} + \widehat{\mu}_{\mathbf{X}|\mathbf{C}}$
17:    Calculate $L_{\text{Flow}} = \| \epsilon^{\text{CW}} - \mathbf{X}_0 - v_\psi^{\text{CW}}(\mathbf{X}_0 + \tau(\epsilon^{\text{CW}} - \mathbf{X}_0), \mathbf{C}, \tau) \|^2$
18:    Calculate $L_{\text{E2E}} = L_{\text{JMCE}} + L_{\text{Flow}}$
19:    Calculate $\nabla L_{\text{E2E}}$ and update the parameters of $\text{JMCE}(\cdot)$ and $v_\psi^{\text{CW}}$
20: **end while**
21: **return** $\text{JMCE}(\cdot)$, $v_\psi^{\text{CW}}$

---

# G COMPUTATIONAL EFFICIENCY

In CW-Diff, our algorithm first trains a JMCE model and then applies conditional whitening to each batch in the training set. The whitened batches are subsequently fed into the diffusion model for training. This final stage requires essentially the same amount of time as a standard diffusion model; therefore, we refer readers to prior work for details on training and sampling times (Shen & Kwok, 2023; Alcaraz & Strodthoff, 2023; Yuan & Qiao, 2024; Li et al., 2024; Ye et al., 2025; Hu et al., 2025). In CW-Flow, however, additional multiplications and additions on white noise are performed in each epoch, leading to extra computational overhead, as shown in line 8 of Algorithm 4. Consequently, the computation time we report includes the training time of JMCE, the time for conditionally whitening all batches, and the extra training time of CW-Flow.

The variability in computation time is negligible, so we report results from a single run. The training time of JMCE primarily depends on the dimensionality and length of the dataset, while the cost of conditional whitening is also affected by dimensionality. Moreover, since we use a highly parallel eigen-decomposition algorithm, the speed depends on the number of batches rather than the number of samples per batch. Table 19 summarizes the training times on ETTh1, ETTh2, ILI, Weather, and Solar Energy. Because ETTh1 and ETTh2 have identical dimensionality and length, their computational efficiency is the same.

For CW-Flow, the tensor operation in line 8 of Algorithm 4 must be performed in every epoch. Its computational complexity is $\mathcal{O}(d^2 T_f^2)$, which is not a negligible cost. Fortunately, with advances in modern hardware and code packages, this operation can be executed in a highly parallelized manner. The extra time is reported in Table 20.

Table 19: The dimensions, total length, the training time of JMCE (Train JMCE), the time of conditionally whiten all batches by eigen decomposition (CW eigen) and the time of conditionally whiten all batches by calculate the inverse of triangle matrix $\widehat{L}_{t|\mathbf{C}}$ (CW trig), the time of training a NsDiff model (Train NsDiff), and the time of training a CW-NsDiff in an E2E style (CW-NsDiff-E2E). All time are counted in second.

| Dataset | Dimension | Total length | Train JMCE | CW eigen | CW trig | Train NsDiff | CW-NsDiff-E2E |
|---|---|---|---|---|---|---|---|
| ETTh1 | 7 | 14,400 | 156 | 2.8 | 2.5 | 79.8 | 212.9 |
| ETTh2 | 7 | 14,400 | 156 | 2.8 | 2.5 | 79.8 | 212.9 |
| ILI | 7 | 966 | 58 | 0.6 | 0.5 | 7.1 | 157.2 |
| Weather | 21 | 52,696 | 780 | 14.2 | 11.4 | 482.92 | 1110 |
| Solar Energy | 137 | 52,560 | 24185 | 14460 | 52.7 | 684.45 | 8120 |

Overall, for datasets with low to medium dimensionality, CW-Gen remains highly efficient. However, for high-dimensional datasets (such as Solar Energy), CW-Gen becomes slower, since it requires performing numerous matrix eigen-decompositions, whose computational complexity is $\mathcal{O}(d^3)$.

Table 20: The dimensions, total length, the time of training a FlowTS model (FlowTS), and time of training a CW-FlowTS model (CW-FlowTS).

| Dataset | Dimension | Total length | FlowTS | CW-FlowTS |
|---|---|---|---|---|
| ETTh1 | 7 | 14,400 | 147 | 152 |
| ETTh2 | 7 | 14,400 | 147 | 152 |
| ILI | 7 | 966 | 11.4 | 12.2 |
| Weather | 21 | 52,696 | 720 | 740 |
| Solar Energy | 137 | 52,560 | 6580 | 10640 |

## H  THE USE OF LARGE LANGUAGE MODELS (LLMS)

In the process of preparation and writing of this paper, we used *ChatGPT 5.0* as the LLM tool for text polishing. The specific application scope includes optimizing the language expression of the abstract, introduction, experimental results, and discussion, improving the clarity and fluency of academic language, and adjusting the logical connection between sentences and paragraphs.

All content polished by the LLM has undergone strict review and manual editing to ensure the accuracy of academic concepts, the rigor of logical reasoning, and the originality of research conclusions. The authors bear full responsibility for the authenticity, integrity, and academic validity of the entire content of the article. The LLM tool was only used for auxiliary text optimization and did not participate in research ideas, experimental design, data analysis, or conclusion derivation, so it does not meet the authorship criteria.

Table 21: Metrics for CW-Gen models with different backbones of JMCE. Each experiment is repeated by 10 times, and standard deviations are provided in brackets. The better results are underlined. NS, FED, and IN indicate that the backbone of JMCE is the Non-stationary Transformer, FED-Former, and Informer, respectively. SEP indicate the mean estimator and covariance estimator are separately trained, whose backbones are Non-stationary Transformer.

| Model Backbone | CRPS (↓) | | | | QICE (↓) | | | | ProbCorr (↓) | | | | Conditional FID (↓) | | | |
|---|---|---|---|---|---|---|---|---|---|---|---|---|---|---|---|---|
| | SEP | IN | NS | FED | SEP | IN | NS | FED | SEP | IN | NS | FED | SEP | IN | NS | FED |
| TimeDiff (2023) | 0.566 (0.032) | 0.952 (0.067) | 0.505 (0.040) | 0.372 (0.011) | 13.171 (1.712) | 14.616 (0.434) | 8.821 (1.916) | 7.215 (0.859) | 0.277 (0.018) | 0.339 (0.020) | 0.243 (0.027) | 0.204 (0.014) | 4.469 (0.416) | 15.612 (6.549) | 6.788 (5.415) | 3.626 (0.281) |
| SSSD (2023) | 0.563 (0.098) | 0.714 (0.171) | 0.524 (0.085) | 0.543 (0.132) | 4.931 (1.953) | 8.120 (2.535) | 4.838 (1.921) | 4.948 (2.927) | 0.273 (0.014) | 0.304 (0.055) | 0.238 (0.030) | 0.299 (0.024) | 26.243 (15.228) | 21.523 (13.230) | 9.265 (5.003) | 22.970 (10.828) |
| Diffusion-TS (2024) | 0.460 (0.029) | 0.652 (0.054) | 0.445 (0.024) | 0.385 (0.015) | 3.474 (1.430) | 8.615 (1.441) | 2.963 (0.887) | 2.919 (0.888) | 0.279 (0.027) | 0.282 (0.016) | 0.266 (0.012) | 0.320 (0.037) | 13.828 (13.082) | 15.609 (4.751) | 7.686 (2.751) | 32.600 (29.270) |
| TMDM (2024) | 0.638 (0.023) | 0.527 (0.041) | 0.440 (0.001) | 0.607 (0.049) | 6.775 (0.737) | 2.820 (1.086) | 4.555 (0.855) | 2.740 (1.067) | 0.251 (0.011) | 0.237 (0.022) | 0.213 (0.001) | 0.279 (0.046) | 13.195 (4.345) | 12.139 (6.919) | 3.831 (0.431) | 31.392 (11.788) |
| NsDiff (2025) | 0.432 (0.014) | 0.640 (0.027) | 0.431 (0.029) | 0.355 (0.005) | 1.880 (0.272) | 7.653 (0.545) | 1.249 (0.228) | 1.432 (0.121) | 0.263 (0.017) | 0.267 (0.011) | 0.206 (0.010) | 0.225 (0.010) | 22.437 (7.024) | 15.080 (3.123) | 8.820 (1.541) | 18.389 (10.985) |
| FlowTS (2025) | 0.482 (0.017) | 0.491 (0.019) | 0.488 (0.020) | 0.583 (0.069) | 4.017 (0.780) | 5.220 (0.569) | 8.817 (0.460) | 4.088 (0.923) | 0.249 (0.016) | 0.233 (0.010) | 0.254 (0.021) | 0.255 (0.015) | 15.836 (3.045) | 8.686 (0.996) | 4.865 (0.563) | 32.906 (9.032) |
| Win rate | 16.7% | 0.0% | 33.3% | 50.0% | 16.7% | 0.0% | 33.3% | 50.0% | 0.0% | 16.7% | 66.7% | 16.7% | 0.0% | 0.0% | 83.3% | 16.7% |

