# OpenReview forum: "Conditionally Whitened Generative Models for Probabilistic Time Series Forecasting"
_ICLR.cc/2026/Conference — ICLR 2026 Poster_

### Official Review · Reviewer_wUr1 · 2025-10-20

**Soundness:** 3
**Presentation:** 3
**Contribution:** 4
**Rating:** 6
**Confidence:** 3

**Summary:**

The authors propose to use a separate model to predict sliding-window means and covariances and subsequently use those to whiten to prediction targets of generative forecasting models.
Their method improves forecasting results for a broad selection of generative forecasting models on several datasets.

**Strengths:**

1. Consistent improvements in experimental results
1. Clear graphical illustration of method

**Weaknesses:**

1. As far as I understand from the algorithms in the appendix, conditional whitening with a JMCE is a wrapper or pre/postprocessing around a generative forecasting model. While this can be interwoven with the diffusion/flow matching dynamics in Section 4, this does not seem essential. If I am correct, I think it would be an advantage to highlight the simplicity of the final method.

**Questions:**

1. Do you train the JMCE model jointly with the generative model?
1. Line 313: How can this be done more efficiently exactly?
1. Are you modifying the generative models themselves? Or do you train the models on the conditionally whitened data as they are?
1. Have you ablated the different components of your JMCE loss?

---

> ### Author Response · Authors · 2025-11-21
> **Rebuttal to W1**
>
> We thank the reviewer for this constructive suggestion and recognizing our paper to have 'consistent improvements' and to be 'clear graphical illustrated'. We address each weakness and questions below.
>
> Your understanding is correct. Training a CW-Gen model consists of three steps:
> (1) pre-training JMCE,
> (2) conditionally whitening the dataset, and
> (3) training the generative model.
> For CW-Diff, we additionally use the outputs of JMCE to invert the conditional whitening operation during sampling, whereas CW-Flow does not require this step. This separate training pipeline indeed provides simplicity in both implementation and conceptual design. Due to the limit of the ICLR submission format, we will elaborate it in future.

---

> ### Author Response · Authors · 2025-11-21
> **Rebuttal to Q1**
>
> We greatly appreciate your wise question. In our original submission, we train JMCE and generative model separately. However, motivated by your comment, we explored training JMCE and the generative model jointly in an end-to-end fashion. We have submitted a revised version of our paper. In Appendix D.10 of the revised submission, We successfully trained JMCE and generative models jointly. The results in Table 18 show that the end-to-end approach allows CW-Gen to better capture the correlations in the data and achieves lower CRPS and FID. Although the end-to-end training slightly decreases QICE, the overall performance remains superior to generative models that do not use any prior information.

---

> ### Author Response · Authors · 2025-11-21
> **Rebuttal to Q2**
>
> In CW-Flow, we directly replace the terminal distribution with the conditional mean and conditional covariance outputted by a trained JMCE. This removes the need to conditionally whiten the dataset. And after generating samples with CW-Flow, we no longer need to apply the inverse CW transformation. For these reasons, we describe this modification as making the algorithm more efficient.

---

> ### Author Response · Authors · 2025-11-21
> **Rebuttal to Q3**
>
> Thank you for your question; it raises a very important point. In fact, we didn't modify diffusion model and modified the flow matching model.
>
> For diffusion models, our forward process follows Eq. (6). However, the SDE in Eq. (6) is equivalent to the SDE in Eq. (7), and the SDE in Eq. (7) is identical to that of the original diffusion model. This means that the forward process of CW-Diff is the same as that of traditional diffusion models. The loss function and reverse process of CW-Diff are also identical to those of standard diffusion models. The key difference is that CW-Diff learns the conditional distribution of the whitened time series, and after sampling, an inverse whitening operation is applied.
>
>
>
> For flow matching, the original flow-matching framework uses a terminal distribution that is a Gaussian with zero mean and identity covariance. In contrast, CW-Flow also adopts a Gaussian terminal distribution, but its mean and covariance are given by the conditional mean and conditional sliding-window covariance estimated by JMCE. Moreover, CW-Flow is trained directly on the raw time-series data, and no additional transformation is required after generating samples with CW-Flow.

---

> ### Author Response · Authors · 2025-11-21
> **Rebuttal to Q4**
>
> Thank you for your question. We have indeed ablated the different components of our JMCE loss. We record $L_2$, $L_F$, $L_{\text{SVD}}$, as well as the ratio between 1 and the minimum eigen value in Tables 9-11. In the revised submission, we have included a visualization of JMCE in Fig. 4 of Appendix D.4.
>
>
>
> Finally, we thank you again for your review. Your questions and comments guided us to address the weaknesses and improve the paper greatly.

---

> > ### Comment · Reviewer_wUr1 · 2025-11-25
> >
> > Thank you for the clarifications. I will remain with my positive assessment.

---

> > > ### Author Response · Authors · 2025-11-27
> > > **Thank you!**
> > >
> > > Thank you for your response. We greatly appreciate your insightful review again. It has been truly helpful for our work.

---

> ### Author Response · Authors · 2025-11-29
> **To AC, PC, SAC: A summary of review and our rebuttal (wUr1)**
>
> Dear AC, PC, SAC,
>
> This is the summary of pWon's review and our rebuttal. Reviewer wUr1 gave our work a positive assessment, highlighting consistent improvements and clear illustrations. pWon's main questions centered on: (1) the pipeline of the CW-Gen, (2) how efficiency of CW-Flow is achieved, (3) whether we modify underlying generative models, and (4) ablations of JMCE’s loss. Our rebuttal directly addressed all these concerns with proper clarification and clear explanations summarized below.
>
> **Major concern 1: the pipeline of the CW-Gen**
>
> W1 and Q1: The pipeline of CW-Gen and the training method of JMCE. And can we jointly train JMCE and generative models?
>
> **Our rebuttal**
>
> We clarified the 3 steps of CW-Gen and emphasized its conceptual and implementation simplicity. We also explained how CW-Diff and CW-Flow differ in postprocessing. In addition, we introduced a new end-to-end training strategy (CW-E2E) in Appendix D.10, which jointly trains JMCE and generative models. Results in Table 18 show CW-E2E has better ProbCorr, lower CRPS and FID.
>
> **Major concern 2: the efficiency of CW-Flow**
>
> Q2: How the CW-Flow can be more efficiently exactly?
>
>
> **Our rebuttal**
>
> We clarified that CW-Flow needs **no** conditional whitening (CW) and its reverse operation, directly using JMCE’s mean & covariance as terminal distribution.
>
> **Major concern 3: the modification of underlying generative models**
>
> Q3: Did we changed the generative models, or just CW the data?
>
> **Our rebuttal**
>
> We provided a complete technical clarification. The diffusion models remain unchanged (same forward SDE, same reverse process, same loss). Flow matching uses a different terminal distribution (JMCE mean & covariance) but trains directly on raw data without further transforms.
>
> **Major concern 4: the ablation study of JMCE loss**
>
> Q4: Have we ablated the different components of your JMCE loss?
>
> **Our rebuttal**
>
> We have ablated the different components of our JMCE loss in Tables 9-11. In the revised submission, we have included a visualization of JMCE in Fig. 4 of Appendix D.4.
>
> **wUr1's final response**
>
> Reviewer wUr1 kept their positive score after our rebuttal.
>
> We greatly appreciate wUr1's insightful review and the contribution of the staff of ICLR.

---

### Official Review · Reviewer_pWon · 2025-10-28

**Soundness:** 3
**Presentation:** 3
**Contribution:** 2
**Rating:** 6
**Confidence:** 4

**Summary:**

The paper introduces conditionally whitened generative models that incorporate information in the noising process. The authors tailor diffusion and flow matching processes by including conditional information obtained via a mean and covariance estimator (termed JMCE). Their approach is theoretically motivated and empirically demonstrated to improve generative performance.

**Strengths:**

- The methodology is theoretically justified, and the authors show under which conditions the distance between the prior and conditional distribution is minimized.
- A simple mean and covariance estimator is introduced and well-motivated to parametrize the generative process.
- The proposed conditional whitening leads to empirical performance improvements.
- The framework includes diffusion and flow matching.

**Weaknesses:**

- The model requires a two-stage process now. First fit JMCE, then train the generative model.
- The derived diffusion process requires the inversion of the covariance matrix, resulting in a higher runtime complexity compared to standard diffusion models. A runtime comparison would aid the comparison.
- The limitations of the method should be discussed more thoroughly. Furthermore, I recommend separating the related work section from the introduction.

Minors:

- L260: Sentence incomplete

**Questions:**

- Can the model be trained in an end-to-end fashion or does it require a two-stage process?
- Did you try diagonal covariance parametrizations to reduce the runtime complexity?
- Can you elaborate more on the performance of the JMCE model itself? How does it compare in forecasting?

---

> ### Author Response · Authors · 2025-11-21
> **Rebuttal to W1 and Q1**
>
> We thank the reviewer for the detailed feedback and for recognizing that our work 'theoretically justified', 'well-motivated' and 'improving performance'. Below, we address the specific comments and outline the improvements we have made based on these suggestions.
>
> Thank you for your review. Your suggestion has been extremely valuable to our work. The current training procedure of CW-Gen is indeed two-stage. Inspired by your comment, we explored an end-to-end training strategy in Appendix D.10 of the revised submission, where JMCE and the generative model are trained jointly. The results in Table 18 show that the end-to-end approach allows CW-Gen to better capture the correlations in the data and achieves lower CRPS and FID. Although the end-to-end training slightly decreases QICE, the overall performance remains superior to generative models that do not use any prior information.

---

> ### Author Response · Authors · 2025-11-21
> **Rebuttal to W2**
>
> Thanks for the review. The runtime comparison is indeed important. In the revised submission, we add a thorough comparison in Appendix F, Table 19. Besides, we also discussed some accelerating way in Appendix D.9 and D.10.

---

> ### Author Response · Authors · 2025-11-21
> **Rebuttal to W3**
>
> Thank you for your suggestion. We plan to provide a more detailed discussion of the limitations of our method in future work, and we will also separate the related work section from the introduction. However, due to the strict page limit of the ICLR submission format, it is difficult for us to make these changes in the current version.

---

> ### Author Response · Authors · 2025-11-21
> **Rebuttal to W4**
>
> Thank you for your review. We appreciate the time you spent carefully reading our paper. The sentence you mentioned is not incomplete. It is intended to briefly recall the output of JMCE introduced in Section 3.2 and define a new notation. We will improve this sentence in the future.

---

> ### Author Response · Authors · 2025-11-21
> **Rebuttal to Q2**
>
> We first use JMCE to obtain the conditional covariance matrix and then apply its diagonal form in CW-GEN. Using only the diagonal conditional covariance (or conditional variances) can indeed substantially accelerate training. We have added a discussion of this point in Appendix D.9 of the revised submission. Furthermore, as shown in Table 14, using only the conditional variances as prior information still yields significant improvements for the generative model; please also refer to our rebuttal to Reviewer NWT6’s W3 for additional details.
>
>
>
> We did not consider diagonal covariance parameterizations in the JMCE because a diagonal covariance matrix cannot approximate a general covariance matrix well in terms of the nuclear norm. Diagonal covariance parameterizations lose the ability to control the minimum eigenvalue of the conditional covariance matrix, and therefore the theoretical foundations of JMCE no longer apply.

---

> ### Author Response · Authors · 2025-11-21
> **Rebuttal to Q3**
>
> Of course! We are happy to provide more information regarding the predictive performance of JMCE. In Appendix D.4, we include an ablation study for JMCE , and Tables 9-11 report detailed results about its behavior. The 'Separate' setting in Table 11 can be viewed as a baseline, and the experiments show that JMCE provides a more stable eatimation of conditional mean, estimates the sliding-window covariance more accurately and effectively controls the minimum eigenvalue. Moreover, we have added a visualization of JMCE in Fig. 4 of Appendix D.4.
>
>
> Finally, we would like to thank you again for your review. Your questions and comments have been invaluable in helping us refine the weaknesses and strengthen the paper.

---

> ### Comment · Reviewer_pWon · 2025-11-27
>
> I thank the authors for their thorough rebuttal and additional experiments. My concerns have been addressed, and I have increased my score accordingly. The paper is solid work, and I recommend acceptance.

---

> > ### Author Response · Authors · 2025-11-28
> > **Thank you!**
> >
> > Thank you very much for your response. We are grateful for your thoughtful and insightful review, which has provided meaningful guidance for our study.

---

> ### Author Response · Authors · 2025-11-29
> **To AC, PC, SAC: A summary of review and our rebuttal (pWon)**
>
> Dear AC, PC, SAC,
>
> This is the summary of pWon's review and our rebuttal. pWon provided a constructive review, recognizing our work as theoretically justified, well-motivated and improving performance. pWon also raised good questions regarding training strategy, runtime, diagonal approximations, and JMCE performance. Our rebuttal directly addressed each concern with new methods, new experiments, and new analyses, which led pWon to increase their rating from 6 to 8 before the leakage incident. Below is a concise summary for the AC.
>
> **Major concern 1: two-stage training, first JMCE then generative model**
>
> W1 and Q1: CW-Gen required a two-stage process, first training JMCE then training  generative model. Is it able to be trained in an end-to-end (E2E) fashion?
>
> **Our rebuttal**
>
> **New end-to-end training scheme CW-E2E (Appendix D.10).** The two-stage CW-Gen can indeed be trained in a E2E fashion. New empirical results (Table 18) show CW-E2E is competitive or better on some metrics (e.g., CRPS, ProbCorr, FID).
>
> **Major concern 2: higher runtime due to covariance inversion**
>
> W2: A detailed runtime analysis is needed.
>
> Q2: Diagonal covariance parametrizations can help reduce the runtime complexity.
>
> **Our rebuttal**
>
> We added a comprehensive runtime comparison (Appendix F, Table 19), including training a JMCE, conditional whitening (CW), and training a generative model. We also discussed the acceleration techniques in Appendix D.9 and D.10, including: inverting triangular-matrixs for CW, and training CW-Gen in an E2E fashion. The diagonal covariance parameterizations has some theoretical concerns. And the rebuttal to Reviewer NWT6’s W3 shows more evidence about the performance of diagonal covariance.
>
> **Major concern 3: Performance of JMCE itself**
>
> Q3: The performance of the JMCE itself and its forecasting ability should be further highlighted
>
> **Our rebuttal**
>
> We have a full JMCE ablation study in Appendix D.4, showing that compared with the baseline model, JMCE has higher mean accuracy, covariance accuracy, eigenvalue stability. We also added new visualization of JMCE outputs in Figure 4.
>
> **Major concern 4: Limitations and organization of the paper**
>
> W3: Limitations should be more thorough. The related work section should be separated from the the introduction.
>
> W4: An incomplete sentence in L260
>
> **Our rebuttal**
>
> We will re-write the involved parts according to the instruction. But now we have to strictly follow the page limit of the ICLR submission format. The sentence in L260 is not incomplete, and we will polish it.
>
>
> **pWon's final response**
>
> pWon raised the rating from 6 to 8 before the document leakage incident.
>
> We are very grateful for pWon's thoughtful and insightful review and for the effort of the staff of ICLR.

---

### Official Review · Reviewer_skbS · 2025-10-29

**Soundness:** 3
**Presentation:** 3
**Contribution:** 3
**Rating:** 6
**Confidence:** 4

**Summary:**

The paper proposes CW-Gen, which whitens data using learned conditional means and covariances from a Joint Mean-Covariance Estimator before training generative models. The idea to align the model priors with data statistics is sound and yields consistent empirical gains across datasets.

**Strengths:**

- Tackle a challenging, important and underrepresented aspect for time-series modelling: multivariate time-series.
- Clear motivation: Proposes a method that allows for a new prior, that is closer to the data-distribution.
- Well written and theoretically founded.
- Extensive evaluation, which show that the proposed prior actually improves state-of-the art models.

**Weaknesses:**

- Unclear connection of DKL reduction in Theorem 1 to actual practical guarantees: Can you (a) clarify the tightness of the bound; (b) give intuition / toy examples showing when the condition is achievable (or not); (c) explicitly discuss regimes where the condition can fail
- Unclear estimator quality: Covariance targets are noisy; stability and regularization not analyzed.
- The paper notes that CW requires eigen-decomposition per sample / per time step but then glosses over practical limitations. More precise complexity analysis and discussion of remedies (approximate eigen/svd, diagonal + low-rank approximation, block-diagonalization, randomized SVD or factor models) could benefit the paper.
- The paper argues joint mean+full covariance is important. But it’s not fully convincing which component drives gains. There are some ablations (backbone/wEigen) but I don’t see a simple controlled ablation that compares: (a) subtract mean only, (b) subtract mean and scale by diagonal variance (NsDiff style), (c) full covariance JMCE. Can you add a direct ablation and show where full covariance helps.

**Questions:**

- Can you be very explicit on how your proposed method compare to other (univariate) flow-based and diffusion-based methods that introduced flexible priors, e.g., [1,2]
- Can you add a short experiment showing numeric values of both sides of Equation (3) on training/validation examples so readers can see how far the estimators are from satisfying the bound in practice.
- Choice of sliding window length (95 for most datasets, 15 for ILI) needs justification. Why 95? How sensitive are results to this hyperparameter?


1. Modeling temporal data as continuous functions with stochastic process diffusion, ICLR 2023
2. Flow Matching with Gaussian Process Priors for Probabilistic Time Series Forecasting, ICLR 2025

---

> ### Author Response · Authors · 2025-11-21
> **Rebuttal to W1**
>
> We thank the reviewer for pointing out that the statement of Theorem 1 could be better connected to practice. Below we (a) clarify the tightness of the bound in (3), (b) provide intuition / toy examples illustrating when the condition is (or is not) achievable, and (c) explicitly discuss regimes where the condition may fail.
>
> (a) Our use of Theorem 1 is primarily to provide theoretical guidance for designing the loss function in JMCE and to show that the optimization objective in JMCE drives down the KLD between the terminal distribution and the true conditional distribution $P_{X|C}$. This KLD appears as a factor in an upper bound on the total variation distance between the conditional diffusion-generated distribution and the true conditional distribution (Chen et al., 2023). Establishing the tightness of this bound is beyond the scope of the present work. Indeed, to the best of our knowledge, existing studies that derive convergence rates for conditional diffusion models in total variation or related divergences also do not establish tightness or minimax lower bounds for these rates (Fu et al., 2024; Tian & Shen, 2025; Wang et al., 2025). Proving tightness or minimax lower bounds for diffusion-type conditional generative models is an interesting but technically demanding problem, which we view as an important direction for future research rather than a goal of this paper.
>
> (b) We have already provided intuition for when the condition in Theorem 1 is achievable (see the paragraph below Theorem 1). In particular, condition (3) is more likely to hold when both $\mu_{X|C}$ and $\Sigma_{X|C}$ can be accurately estimated and $\mu_{X|C}$ is appreciably different from zero, which is common in non-stationary time series. The regimes where the condition fails are discussed in our response to point (c) below. Here, we present a toy example to further illustrate when the condition is achievable. Suppose $d_x=1$ and $X|C\sim N(\mu_{X|C},\sigma^2_{X|C})$. Then $Q_0= N(0,1)$ and $\widehat{Q}= N(\widehat{\mu}\_{X|C}, \sigma^2\_{X|C})$. In this case, the nuclear and Frobenius norms reduce to absolute values, and condition (3) becomes $$\frac{(\mu\_{X|C}-\widehat{\mu}\_{X|C})^2+|\sigma^2\_{X|C}-\widehat{\sigma}^2\_{X|C}|}{\widehat{\sigma}^2\_{X|C}}+|\sigma^2\_{X|C}-\widehat{\sigma}^2\_{X|C}|\leq \mu\_{X|C}^2.$$ Intuitively, this inequality requires that the signal level $\mu\_{X|C}^2$ (how far the conditional mean is from zero) must dominate the estimation error $(\mu\_{X|C}-\widehat{\mu}\_{X|C})^2/\widehat{\sigma}^2\_{X|C}$,  $|\sigma^2\_{X|C}-\widehat{\sigma}^2\_{X|C}|/\widehat{\sigma}^2\_{X|C}$, and $|\sigma^2\_{X|C}-\widehat{\sigma}^2\_{X|C}|$. In non-stationary time series, $\mu\_{X|C}$ is typically non-negligible, and our JMCE module is trained exactly to drive $(\mu\_{X|C}-\widehat{\mu}\_{X|C})^2/\widehat{\sigma}^2\_{X|C}$, $|\sigma^2\_{X|C}-\widehat{\sigma}^2\_{X|C}|/\widehat{\sigma}^2\_{X|C}$ and $|\sigma^2\_{X|C}-\widehat{\sigma}^2\_{X|C}|$ to be small. In such cases, the inequality is easily satisfied.
>
> [1] Sitan Chen, Sinho Chewi, Jerry Li, Yuanzhi Li, Adil Salim, and Anru Zhang. Sampling is as easy as learning the score: theory for diffusion models with minimal data assumptions. In International Conference on Learning Representations, 2023.
>
> [2] Hengyu Fu, Zhuoran Yang, Mengdi Wang, and Minshuo Chen. Unveil conditional diffusion models with classifier-free guidance: A sharp statistical theory. arXiv 2403.11968, 2024.
>
> [3] Xinyu Tian and Xiaotong Shen. Conditional data synthesis augmentation. Journal of the American Statistical Association, pp. 1–22, 2025.
>
> [4] Rongzhen Wang, Yan Zhang, Chenyu Zheng, Chongxuan Li, and Wu Guoqiang. A theory for conditional generative modeling on multiple data sources. In International Conference on Machine Learning, 2025.

---

> ### Author Response · Authors · 2025-11-21
> **Rebuttal to W1 continue**
>
> (c) First, if $\mu\_{X|C}=0$, then the right-hand side in (3), $\| \mu\_{X|C} \|\_2^2$, equals zero, while the left-hand side of (3) is dominated by estimation error. In this case, even small estimation errors can cause the inequality in (3) to fail. In practice, however, for non-stationary time series, $\mu\_{X|C}$ often exhibits sharp variations and thus deviates from zero, so this scenario is unlikely to occur.
>
> Second, When $\min_{i \in \{ 1,\ldots,d_x \} } \widehat{\lambda}\_{X|C,i}$ is very small, the factor $\left( \min\_{i \in \{ 1,\ldots,d\_x \} } \widehat{\lambda}\_{X|C,i} \\right)^{-1}$ in (3) can blow up, so even modest deviations $\| \mu\_{X|C} - \widehat{\mu}\_{X|C} \|\_2^2$ and $\| \\Sigma\_{X|C} - \widehat{\Sigma}\_{X|C} \|\_N$ may violate the condition. This motivates the explicit eigenvalue regularization in our JMCE loss in (4), which enforces strictly positive eigenvalues bounded away from zero.
>
> Finally, Condition (3) explicitly involves the estimation errors $\| \mu\_{X|C} - \widehat{\mu}\_{X|C} \|\_2^2$, $\| \Sigma\_{X|C} - \widehat{\Sigma}\_{X|C} \|\_N$, and $\| \Sigma\_{X|C} - \widehat{\Sigma}\_{X|C} \|_F$. If $(\widehat{\mu}\_{X|C},\widehat{\Sigma}\_{X|C})$ are not good estimators of $(\mu\_{X|C},\Sigma\_{X|C})$, then these terms on the left-hand side of (3) become large and may easily exceed $\| \mu\_{X|C} \|\_2^2$ on the right-hand side. To mitigate this, we deliberately design JMCE as a joint conditional mean and covariance estimator whose loss directly mirrors the left-hand side of (3). The experiments in Section D.4 show that JMCE achieves small estimation error, providing empirical evidence that our estimator yields sufficiently accurate $(\widehat{\mu}\_{X|C},\widehat{\Sigma}\_{X|C})$.

---

> ### Author Response · Authors · 2025-11-21
> **Rebuttal to W2**
>
> Thank you for your review. Examining the behavior of JMCE when learning covariance matrices is indeed highly important.
>
> First, JMCE does not learn the true conditional covariance, as it cannot be directly computed. Instead, it is trained to learn the conditional covariance within a sliding window. The noise level of this covariance is substantially reduced due to the sliding window. To illustrate this phenomenon, we have added visualizations of JMCE's prediction targets and outputs in Figure 4 of Appendix D.4 in the revised submission.
>
> Second, in Tables 9-11, we reported the ratio between 1 and the minimum eigenvalue, which can be viewed as an evaluation of the regularization effect. We also reported its standard deviation which can be viewed as the stability. In addition, we study how the minimum eigenvalue $\lambda_{\min}$ and the weight of the regularization term $w_{\text{eigen}}$ influences JMCE. Consequently, we also reported the influence of these parameters on CW-Gen, with the corresponding results reported in Tables 12 and 13.

---

> ### Author Response · Authors · 2025-11-21
> **Rebuttal to W3**
>
> Thank you for your valuable comments. Computational complexity analysis is indeed essential for any algorithm. In the revised submission, we have updated Table 19 to report in detail the runtime of each component of our method. Thanks to the parallel computing capabilities of modern GPU frameworks, the eigen-decomposition performed per sample or per timestep can be processed in parallel, and therefore does not constitute a major source of additional computational cost. We have also implemented additional strategies to further improve computational efficiency while maintaining competitive performance. Please refer to our response to Reviewer NWT6, Weakness 1, for the detailed description of these techniques. We greatly appreciate your suggestions in this regard. Due to time constraints, we have not yet fully implemented the approach you proposed, but it will have significant influence on our future work.

---

> ### Author Response · Authors · 2025-11-21
> **Rebuttal to W4**
>
> Thank you for your suggestion. It is indeed very important to analyze the benefits contributed by each component of the prior model. In the revised submission, we have added the corresponding experiments in **Table 14 of the Appendix D.6**. In brief, we systematically evaluated the performance of CW-Gen under the following prior information configurations: conditional mean only (Mean), diagonal conditional covariance only (Var), full conditional covariance only (Cov), conditional mean combined with conditional variance (Mean \& Var), and conditional mean combined with full covariance (Mean \& Cov). Table 14 shows that the Mean \& Cov configuration performs best, followed by Mean \& Var, with Mean alone in third place.

---

> ### Author Response · Authors · 2025-11-21
> **Rebuttal to Q1**
>
> Thank you for your question. Comparing our method against univariate generative models that also incorporate prior information is indeed very important. In Appendix D.8 of the revised submission, we provide a detailed discussion of the similarities and differences between our CW-Gen and DSPD [1] and TsFlow [2].
>
> In brief, DSPD leverages kernel functions (e.g., $\exp(-\gamma |t_i - t_j|)$) to help diffusion models better capture temporal correlations within the prediction window. However, such prior information is not conditioned on the historical temporal context, resulting in insufficient prior strength to support forecasting tasks. Moreover, DSPD lacks conditional mean priors—an omission that negatively impacts its predictive performance, as illustrated in Figure 5 of the revised version.
>
> TsFlow models the conditional mean and variance within the prediction window using Gaussian processes, but Gaussian processes are limited by kernel choices. More importantly, computing the mean and variance of a Gaussian process requires inverting a matrix of size equal to the historical observation length (i.e., $\mathbb{R}^{T_h \times T_h}$), which poses computational challenges for long-range forecasting.
>
> Furthermore, we applied DSPD and TsFlow to each dimension of the ETTh1 dataset and evaluated their performance using the same four metrics as in the main paper. The results are presented in Table 17, and visualizations of the means and variances of the generated samples are provided in Figure 5. The results show that both types of univariate models perform slightly worse than the proposed CW-Gen model.

---

> ### Author Response · Authors · 2025-11-21
> **Rebuttal to Q2**
>
> Thank you for your review. We agree that understanding the both-hand side term of the theoretical equation is indeed crucial. However, for real-world time series datasets, the conditional mean and the conditional covariance cannot be explicitly obtained, and therefore the both-hand side cannot be computed explicitly either. To address this concern, we have added new visualizations of the ETTh1 time series in Figure 4 of the revised submission. As shown in the figure, ETTh1 exhibits highly non-stationary and periodic patterns, indicating that the conditional mean is strongly shifted away from zero. This further implies that the right-hand side, i.e., $\mu_{\mathbf{X}\mid \mathbf{C}}$, is significantly greater than zero in practice.

---

> ### Author Response · Authors · 2025-11-21
> **Rebuttal to Q3**
>
> Thank you for your review. The choice of the sliding window length is indeed something that needs to be justified. In short, our choice of 95 is inspired by NsDiff. Due to implementation convenience, we set the length to 95 instead. Similar to the ETTh1 dataset (95/192), the window length on ILI also covers roughly half of the prediction window (15/32).
>
> We have added an ablation study of the length of the sliding window (written in lowercase as requested) in Appendix D.7 of the revised submission. The effects of different window lengths on CW-Gen can be found in Tables 15 and 16. Overall, changing the sliding window length has little impact on ETTh1, because the window is sufficiently long and modifying its length does not significantly affect the resulting sliding-window covariance. On the other hand, the variability introduced by changing the window length is slightly larger on the ILI dataset, but still remains within a reasonable range.
>
>
> Finally, we sincerely thank you again for your review. Your insightful questions and comments have helped us identify the shortcomings of our work and significantly improve the paper.

---

> > ### Comment · Reviewer_skbS · 2025-11-25
> >
> > I want to thank the authors for their extensive rebuttal. I have increased my score.

---

> > > ### Author Response · Authors · 2025-11-27
> > > **Thank you!**
> > >
> > > Thank you for your response. Again, we sincerely appreciate your thoughtful and insightful review, which provides important guidance for our work.

---

> ### Author Response · Authors · 2025-11-29
> **To AC, PC, SAC: A summary of review and our rebuttal (skbS)**
>
> Dear AC, PC, SAC,
>
> This is the summary of skbS's review and our rebuttal. Reviewer skbS gave a detailed and constructive review, recognizing our paper to be 'tackling a challenging task', 'clearly motivated', 'well written' and 'extensively evaluated'. skbS also raised good questions regarding theorem, numeric results of JMCE, computational speed of CW-Gen, the contribution of individual prior components, comparison with other pioneer (univariate) models, and the ablations of the length of sliding-window. Their feedback substantially improved the paper, and our rebuttal directly addressed every weakness with new theory clarifications, new experiments, and new analyses. skbS increased their rating from 6 to 8 before the leakage incident. Below is a detailed summary for the AC.
>
> **Major concern 1: practical relevance of Theorem 1**
>
> W1: The connection between the KL reduction in Theorem 1 and practical guarantees is unclear.
>
> **Our rebuttal**
>
> We added **substantial new theoretical clarification**, including:
>
> 1. A clear explanation of why Theorem 1 matters in practice (guides JMCE’s loss construction).
>
> 2. Discussion of the non-tightness of existing diffusion TV bounds and why tightness is beyond scope (with citations to Chen 2023; Fu 2024; Tian 2025).
>
> 3. Intuitive explanations and a 1D Gaussian toy example illustrating when Condition (3) is achievable in practice.
>
> 4. Discussion of failure regimes and an explicit explanation of how JMCE’s eigenvalue regularization and loss design are tailored to mitigate these issues.
>
> **Major concern 2: numeric results of JMCE and the both-hand sides of Eq. (3)**
>
> W2: Since the covariance targets are noisy, the estimator quality, stability and regularization should be analyzed.
>
> Q2: both-hand sides of Eq. (3) in training/validation sets should be provided.
>
> **Our rebuttal**
>
> First we clarify that our learning target is **sliding-window covariance** which is not noisy due to the sliding-window. New visualizations of JMCE's prediction targets and outputs of training/test sets are provided in Figure 4 of Appendix D.4. Alougth the true both-hand sides of Eq. (3) is not computable,  the Figure 4 shows non-stationarity time series has a larger right-hand side, which means our theorem 1 is valid in practice.
>
> We also provided the abalation study about the estimator quality, stability and regularization in Table 9-11.
>
> **Major concern 3: computational speed of CW-Gen**
>
> W3: CW-Gen requires extensive computation, while some matrix approximation methods can benefit the paper. In addition, more precise complexity analysis is needed.
>
> **Our rebuttal**
>
> We implemented multiple new efficiency strategies, including compute the inverse matrices of the lower-triangle matrices, and training JMCE and generative model together (CW-E2E). More detailed description can be found in our response to NWT6's W1 and W2. We also added a more precise complexity analysis in Table 19.
>
> **Major concern 4: the contribution of individual prior components**
>
> W4: Abalation study of individual prior components should be studyed.
>
> **Our rebuttal**
>
> We added a new ablation study to show the contribution of individual parts of the prior model (Appendix D.6, Table 14), covering: Mean only, Var only, Cov only, Mean & Var and Mean & Cov. More detalis can be found in our response to NWT6's W3.
>
> **Major concern 5: comparison with other pioneer (univarite) models**
>
> Q1: Cw-Gen's relation to pioneer univariate models with prior models (DSPD, TsFlow)
>
> **Our rebuttal**
>
> We added a new subsection (Appendix D.8, Table 17, Figure 5) to show detailed comparison between CW-Gen and DSPD/TsFlow. The new experiments show both DSPD and TsFlow underperform CW-Gen in the multivariate setting. And we clearly explained the advantages of CW-Gen.
>
>
> **Major concern 6: ablations of the length of sliding-window**
>
> Q3: The choice of sliding-window length needs justification. The sensitivity also needs justification.
>
> **Our rebuttal**
>
> Our choice of the sliding-window length is inspired by NsDiff. And we have added an ablation study of the length of the sliding-window in Appendix D.7, Table 15-16. The results shows sliding-window length only has quite limited impact.
>
> **skbS's final response**
>
> skbS raised the rating from 6 to 8 before the document leakage incident.
>
> Finally, we sincerely thank skbS's thoughtful and insightful review and the effort of the staff of ICLR.

---

### Official Review · Reviewer_Nwt6 · 2025-11-01

**Soundness:** 3
**Presentation:** 3
**Contribution:** 3
**Rating:** 6
**Confidence:** 2

**Summary:**

The paper introduces CW-Gen, including CW-Diff and CW-Flow. It proposes a novel JMCE to simultaneously estimate the conditional mean and sliding-window covariance of future time series, which guides the whitening process. The authors provide theoretical guarantees showing conditions under which their approach improves the generative model's sample quality by reducing the KL divergence between the conditional distribution and the model's terminal distribution. Experimental evaluations demonstrate improvements in probabilistic forecasting accuracy, capturing non-stationary dynamics and inter-variable correlations better than baselines.

**Strengths:**

1. The paper establishes conditions that justify why replacing the traditional terminal Gaussian distribution with one parameterized by estimated conditional mean and covariance improves sample quality.
2. The JMCE simultaneously learns accurate conditional means and sliding-window covariances with eigenvalue control to ensure stability and robustness—this nuanced approach effectively addresses non-stationarity and heteroscedasticity.
3. Experiments show the outperformed performance over baselines.
4. The algorithms and theoretical proofs are clearly detailed, and code for experiment reproduction is available.

**Weaknesses:**

1. The approach involves computationally expensive operations, particularly eigen-decomposition for whitening covariance matrices, which scales cubically with dimensionality. For very high-dimensional datasets, CW-Gen can become quite slow, limiting real-time deployment scenarios.
2. The framework's reliance on complex joint estimators and whitening transformations may impose a higher barrier to adoption compared to simpler baseline models.
3. The paper ablates hyperparameters but does not isolate the individual contributions of conditional mean estimation versus covariance estimation. Which component drives most improvements?

**Questions:**

See above

---

> ### Author Response · Authors · 2025-11-21
> **Rebuttal to W1**
>
> We sincerely thank the reviewer for recognizing our work to be "novel," "theoretically sound" and "improving performance." We greatly appreciate the constructive feedback. We address each weakness in detail below.
>
> Thank you for your insightful review. Training a CW-Gen model consists of three steps:
> (1) training the JMCE,
> (2) conditionally whitening the dataset, and
> (3) training the generative model.
> The first two steps involve computationally expensive eigen-decomposition and SVD operations. We acknowledge that they may affect the efficiency of CW-Gen on high-dimensional datasets; however, we have made the following efforts to significantly improve computational efficiency while maintaining competitive performance, thereby facilitating real-time deployment in high-dimensional scenarios.
>
>
> In step (2), the conditional whitening stage, we added Appendix D.9 in the revised submission. In short, suppose $X \in \mathbb{R}^d$ is a random variable with covariance $\mathrm{Cov}(X) = \Sigma = L L^\top$, where $L$ is a lower-triangular matrix. Then it is straightforward to verify that $\mathrm{Cov}(L^{-1} X) = L^{-1} L L^\top L^{\top^{-1}} = I_d$. Therefore, we can avoid eigen-decomposition and instead compute an inverse of a **lower-triangular matrix**. We also added some timing results in Appendix F. On the higher-dimensional dataset (Solar Energy), we are able to reduce the time of step (2) from 14,460 seconds to 52.7 seconds!
>
> For steps (1) and (3), we have proposed an end-to-end joint training scheme for JMCE and generative models (CW-E2E) in Appendix D.10 of the revised submission. On the Solar Energy dataset, the total training time of the CW-E2E model has been significantly reduced from 39,329 seconds to 8,120 seconds. We also present the practical performance of CW-E2E in Table 18. The results demonstrate that CW-E2E remains competitive across all metrics; in fact, it even outperforms the method proposed in the main paper in terms of CRPS, ProbCorr and Conditional FID.

---

> ### Author Response · Authors · 2025-11-21
> **Rebuttal to W2**
>
> We acknowledge that training CW-Gen introduces additional computational cost. However, Tables 2-8 demonstrate that even when paired with simpler baseline models, CW-Gen improves performance across the vast majority of metrics. Moreover, CW-Gen can be applied to high-dimensional datasets, such as the 137-dimensional Solar Energy dataset. In addition, as discussed in our rebuttal to W1, we provide several techniques to further accelerate computation.

---

> ### Author Response · Authors · 2025-11-21
> **Rebuttal to W3**
>
> Thank you for your review. We acknowledge that the original submission unintentionally omitted an evaluation of the independent contributions of different prior components. In the revised submission, we have added the corresponding experiments in **Table 14 of the Appendix D.6**. In brief, we systematically evaluated the performance of CW-Gen under the following prior information configurations: conditional mean only (Mean), diagonal conditional covariance only (Var), full conditional covariance only (Cov), conditional mean combined with conditional variance (Mean \& Var), and conditional mean combined with full covariance (Mean \& Cov). Based on the results in Table 14, we can conclude that the **conditional mean provides the largest individual improvement**. However, **combining the conditional mean with  the conditional variance/covariance yields even better performance**. The performance ranking is as follows: **the Mean & Cov configuration performs best**, followed by Mean \& Var, with Mean alone in third place. When Var or Cov is used individually, CW-Gen exhibits performance degradation; however, it is noteworthy that Cov consistently outperforms Var in the ProbCorr metric, indicating that the full covariance indeed helps the generative model capture inter-variable dependencies more effectively.
>
>
> Finally, we would like to once again thank you for your review. Your questions and comments have helped us address the weaknesses of our work and improve the overall quality of the paper.

---

> ### Comment · Reviewer_Nwt6 · 2025-11-25
>
> Thank you for your rebuttal, and I will keep my score.

---

> > ### Author Response · Authors · 2025-11-27
> > **Thank you!**
> >
> > Thank you very much for your response. We would like to express our gratitude again for your insightful review, which has been highly valuable for our work.

---

> ### Author Response · Authors · 2025-11-29
> **To AC, PC, SAC: A summary of review and our rebuttal (Nwt6)**
>
> Dear AC, PC, SAC,
>
> We understand that we are in a hard period, and the new ICLR policies have significantly increased your workload. To help minimize the time you need to spend, we have summarized each reviewer's feedback and our corresponding rebuttal below each review.
>
> We first would like to thank Reviewer Nwt6, who acknowledged our work as novel, theoretically sound, and performance-improving. Nwt6's concerns mainly focused on (1) computational efficiency, and (2) the contribution of individual prior components. Our rebuttal provided substantial new technical content, new experiments, and concrete improvements that directly addressed each concern. Below is a concise summary for the AC.
>
> **Major concern 1: computational cost**
>
> W1: Eigen-decomposition and conditional whitening (CW) may be too slow for high-dimensional datasets.
>
> W2: The algorithm may have a higher barrier compared to simpler baseline models.
>
> **Our rebuttal**
>
> 1. **New method to avoid eigen-decomposition entirely (Appendix D.9).** We show that whitening can be implemented via inverting a lower-triangular matrix, yielding a dramatic speed-up. On Solar Energy dataset (137-dim), the time of CW can be reduced from 14,460s to 52.7s.
> 2. **New end-to-end training scheme CW-E2E (Appendix D.10).** CW-E2E jointly trains JMCE and the generative model, eliminating redundant computations. Total time On Solar Energy dataset can be reduced from 39,329s to 8,120s. New empirical results (Table 18) show CW-E2E is competitive or better on some metrics (e.g., CRPS, ProbCorr, FID).
> 3. Indeed that training CW-Gen introduces additional computational cost. But compared with simpler baseline models, CW-Gen improves performance across the vast majority of metrics (Tables 2-8).
>
> **Major concern 2: unclear contributions of individual parts of the prior model**
>
> W3: Which component (mean, variance, covariance) contributes most?
>
> **Our rebuttal**
>
> We added a new ablation study to show the contribution of individual parts of the prior model (Appendix D.6, Table 14), covering: Mean only, Var only, Cov only, Mean & Var and Mean & Cov. The key findings are:
>
> 1. Conditional mean provides the largest single improvement.
> 2. Mean & Cov performs best overall, confirming that our CW-Gen algorithm is reasonable and effective.
> 3. Cov is better than Var in capturing inter-variable correlations (ProbCorr metric).
>
>
> **Nwt6's final response**
>
> Reviewer Nwt6 kept their positive score after our rebuttal.
>
> Again, we would like to express our sincere gratitude to Nwt6's insightful review, and to all organizers and staff.

---

### Meta-Review · Area_Chair_GNpi · 2026-01-06

**Summary:**

I recommend acceptance. This paper introduces a new way to incorporate prior information in diffusion/flow-style forecasters with a learned conditional prior from a whitened latent space. Across reviews, the common view is that the idea is clean, the paper is clearly written, and the empirical gains are consistent; the hesitation is almost entirely about practicality and attribution. Examples include the cost of per-sample covariance factorizations, how the two-stage pipeline should be understood (and whether it can be end-to-end), which part of the prior actually buys the improvement (mean vs variance vs full covariance), and how to interpret the KL-based theory in realistic settings. The rebuttal largely resolves the core technical concerns with concrete efficiency and runtime work, an end-to-end variant, and ablations that clarify the roles of mean and covariance. Two reviewers upgraded their ratings, while others positive. For the camera-ready version, I would recommend having an explicit scaling/complexity story, and a practical guide for when full covariance is worth it (and when to approximate), tighter positioning against flexible-prior diffusion/flow baselines; also to add a paragraph about estimator noise and failure modes.

**Reviewer Concerns:**

The authors largely addressed the main concerns during the rebuttal by adding 1/ an efficiency-oriented whitening implementation (avoiding eigendecomposition via triangular-factor inversion) and a runtime breakdown; 2/ an end-to-end joint training variant to reduce total pipeline cost; (iii) direct prior-component ablations showing that the conditional mean is the strongest single contributor while mean+covariance performs best overall; and (iv) clearer discussion of what the theorem does/does not guarantee, including toy intuition and failure regimes, plus sensitivity results for the sliding-window length.

**Reviewer Scores:**

Two reviewers explicitly increased their scores post-rebuttal; one kept their score; the remaining reviewer stayed positive.

---

### Decision · Program_Chairs · 2026-01-26

Accept (Poster)